# COPU: Recognizing Time Series' Heterogeneity In Stacked Neural Network

## Abstract

Neural networks (NNs) have been widely studied in complex fields due to their remarkable capacity for nonlinear modeling. However, in the realm of time series analysis, researches indicate that merely stacking NNs does not yield promising nonlinear modeling outputs and hinders model performance. Conventional NN architectures overemphasize homogeneous feature extraction, impeding the learning of diverse features and diminishing their nonlinear modeling capability. To address this gap, we propose the **C**ross-correlation Enhanced Approximated **O**rthogonal **P**rojection **U**nit (COPU) to quantify and augment the NN's nonlinear modeling capacity. COPU efficiently computes the local cross-correlation characteristics between features, amplifying heterogeneous components while compressing homogeneous ones. By reducing redundant information, COPU facilitates the learning of unique and independent features, thereby enhancing nonlinear modeling capability. Extensive experiments demonstrate that our method achieves superior performance across two real-world regression applications.

## 1 Introduction

A plethora of successful research endeavors featuring modular designs based on stacked structures has emerged in complex fields such as Computer Vision (CV) (Dosovitskiy et al., 2020) and Natural Language Processing (NLP) (Ouyang et al., 2022; Brown et al., 2020). The effectiveness of these designs is attributed to their stacked architecture (Ashish et al., 2017). However, cutting-edge deep learning research on time series analysis utilizes fewer stacked layers than that of CV and NLP, typically only 1 to 4 layers (Chong et al., 2023; Haixu et al., 2021; Tian et al., 2022). This phenomenon arises because base modules designed for CV and NLP reach their expressiveness ceiling quickly when applied in time series. For these methods to be effectively applied in this field, it is essential to recognize that time series has more ambiguous discriminative patterns than other forms like images and text (Alec et al., 2021). Such ambiguity hinders the model's ability to extract diverse features from the input, obstructing its capacity for nonlinear modeling. Specifically, it is relatively straightforward to distinguish images belonging to different categories or texts conveying various emotions. However, it is more challenging to discern the effect of two input sequences on the output of a system, particularly through a nonlinear dynamic system (Elad et al., 2018). Thus, from the perspective of input, the discriminative patterns among different time series are not only difficult to express mathematically but also inherently ambiguous. Figure 1 vividly illustrates this process using a simple kernel convolution.

Rank Ratio (RR) is introduced as a metric to gauge the confidence degree of a matrix's inverse in Shaoqi et al. (2024). We would also utilize it to measure the nonlinear modeling capability of NNs. In matrix analysis, the rank signifies the number of linearly independent vectors, a characteristic that embodies unique information not representable through linear combinations of others (Meyer, 2023). RR can be interpreted as the proportion of linear dependencies that have been transformed into independencies through NN. An RR value approaching 1 indicates that the extracted features contain a great amount of unique information. Such features empower NN to capture diverse and informative patterns, thereby enhancing accuracy. Conversely, an RR value approaching 0 suggests that the extracted features have much redundant information. Such features compel NN to focus on more homogeneous characteristics. When a shift occurs, e.g., from source to target (Ido et al., 2024), from training to testing (Olivia et al., 2022), or from offline to online (Yichen et al., 2024), the distribution of extracted features may undergo significant changes, increasing the risk of overfitting.

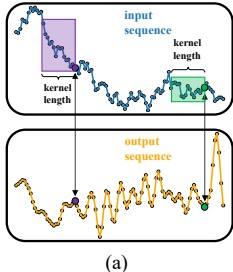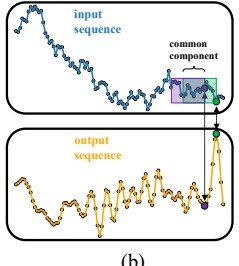

(a)            (b)

Figure 1: Input is cumulative random variables, kernel is a sine function. The input is convolved with the kernel to produce the output. (a) two independent and wide different segments of the input yield similar outputs; (b) two consecutive segments sharing numerous common components result in significantly different outputs.

Existing NN-based methods primarily focus on proposing innovative model structures while neglecting the analysis of RR (Tian et al., 2022; Yuqi et al., 2023). Specifically, most NN research centers on constructing local rules at the neuron level to mimic specific biomimetic or mathematical patterns, assuming that stacking these local rules will extract conducive global features, and validating this assumption experimentally (Kaiming et al., 2016; Ze et al., 2021). For instance, Albert et al. (2020); Gu et al. (2022); Albert et al. (2022) integrate the understanding of Fourier Recurrent Units and Legendre Memory Units to propose a state-space method for dynamic linear encoding with minimal information loss for long sequence data, empirically employing gating mechanisms to enhance nonlinear modeling capabilities (Alberta & Tria, 2024). Similarly, Nikita et al. (2020) and Haoyi et al. (2021) have modified attention mechanisms based on complex block partitioning and entropy sampling principles, introducing the Reformer and Informer for time series forecasting through multi-layer stacking. Haixu et al. (2021) have also proposed Autoformer, which leverages autocorrelation by rewriting attention mechanisms. Additionally, Zhiding et al. (2024) introduce a hierarchical Transformer-based transfer learning structure to capture temporal dependencies within sequences through stacked NNs. Huang et al. (2024)'s generative structure models scale-invariant temporal features by simulating evolutionary behaviors. Furthermore, Maximilian et al. (2024) have advanced LSTM architectures by integrating attention mechanisms, while Yuxuan et al. (2024) enhance the generalization ability of large language models across various time series analysis tasks by sharing encoders within each patch. These novel methods use stacked NNs to improve their expressive capability. The mainstream of conventional stacked network research involves empirically explore the feasibility of extracting features using base modules that exhibit outstanding performance under mean squared error or entropy loss in end-to-end tasks. However, it remains challenging to quantify and analyze whether these modules are qualified as feature extractors, and whether the features they extract are conducive to and consistent with the overall model. Furthermore, they struggle to monitor whether such extracted features are beneficial in optimization, much less guide directions for model improvement. An index that enables real-time monitoring of NN nonlinear modeling capability can significantly enhance interpretability and guide directions for model improvement (Shaoqi et al., 2024). RR fulfills this role and offers a general NN design principle centered on increasing RR. RR closely parallels the mesa-optimization problems proposed by Evan et al. (2021) and can be regarded as an analysis of prior alignment in deep NN optimization (Collin et al., 2023; Xu et al., 2024).

We propose the Cross-correlation Enhanced Perceptron (CEP) to achieve deep nonlinear modeling for time series data. CEP leverages the structured characteristics of sequential data, simultaneously aligning input features and measuring their correlations within a single step. This process enables the differentiation between redundant information and innovation, further suppressing homogeneous information among features while amplifying their differences. As a result, linear dependencies are transformed into independencies, facilitating the effective construction of nonlinear patterns. AOPU is the representation method that utilizes RR for the analysis of the approximation of natural gradient in online NN regression (Shaoqi et al., 2024). AOPU's gradient propagation of the dual parameters places high demands on RR. We introduce COPU by replacing the random gaussian matrix with CEP as the augmentation interface for AOPU, ensuring the accuracy of natural gradient calculations. This

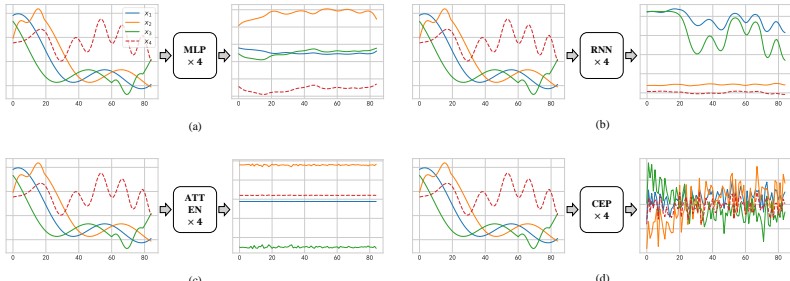

Figure 2: Visualizations of the outputs obtained through 4-layer-stacked initialized NN. In the input, $x_1$, $x_2$, and $x_3$ represent manifestations of the same pattern at different time lags, with additional innovations introduced at the end of $x_2$ and the beginning of $x_3$. $x_4$ is unrelated and independent. (a) visualization via multi-layer perceptron (MLP); (b) visualization via recurrent neural network (RNN); (c) visualization via attention (ATTEN); (d) visualization via CEP.

enhancement boosts the model's expressive power and improves its stability. The contributions of this paper are summarized as follows:

1. Performing a comprehensive exploration of NN's nonlinear modeling capability. RR is introduced as a metric to quantify the proportion of linear dependencies transformed into independence by the network, thereby reflecting the nonlinear capacity. Stacked NN does not bring high RR which explains the inapplicability of deep NN in time series. A potential relationship has been found between an increased RR and rapid convergence in training, highlighting the critical importance of studying RR.

2. Developing CEP framework to augment NN's nonlinear modeling capability in time series data. COPU amplifies heterogeneity and suppresses homogeneity among features, facilitating the learning of unique and independent information, thereby enhancing nonlinear modeling capability.

3. Developing COPU achieve efficient approximation to natural gradient and minimum variance estimation. As CEP success to maintain high RR in stacked structure during training, the precision compromises inherent in COPU have been moderated, resulting in more stable and superior performance.

## 2   CEP: CROSS-CORRELATION ENHANCED PERCEPTRON

In time series analysis, it is common to encounter multiple variables that embody a specific pattern at different time lags (Haixu et al., 2021). Figure 2 vividly illustrates this phenomenon. Each subplot's input comprises four variables: $x_1$, $x_2$, and $x_3$ all represent manifestations of the same pattern, while $x_4$ is independent. Such repetition of variables and their lag characteristics are prevalent in time series analysis, in particular, the industrial process (Qingqiang & Zhiqiang, 2021; Yan et al., 2024). Crucial differences between $x_1$, $x_2$, and $x_3$ emerge in the heterogeneous fluctuations observed at the beginning of $x_3$ and the end of $x_2$; these innovations could harbor key patterns vital for system identification.

Common studies employ architectures such as MLP, RNN, and ATTEN to extract features from those data for subsequent modeling, owing to these structures' excellent input-output mapping capabilities. **However, as feature extraction modules, the capacity to model nonlinearity and extract diverse features should be prioritized.** In this section, we leverage RR to compare the nonlinearity of various modules' output. Simultaneously, we document the performance of these modules when trained end-to-end, providing a wealth of valuable insights.

To address this challenge, we propose CEP as a nonlinear modeling solution designed to amplify heterogeneous differences among closely related features. CEP first tackles the issue of sequence alignment among features. Without accounting for time lags, the correlation coefficients between features expressing the same nonlinear pattern may be zero, as exemplified by sine and cosine functions. Subsequently, CEP quantifies the similarity characteristics of the aligned features. CEP

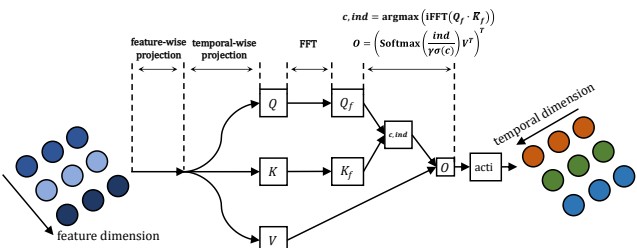

Figure 3: Schematic of CEP structure.

designs a dynamic weighting scheme to suppress the redundant extraction of homogenized information, thereby enhancing the model's focus on innovations. An easier alignment of two features indicates higher similarity and also implies sharing a greater amount of homogenized information so that they should be assigned lower weights and vice versa. We implement this approach by employing dictionary lookup (Ashish et al., 2017) to achieve feature mapping. CEP is defined as follows:

$$CEP: \quad c, ind = \text{CC}(Q, K), \quad Out = \text{acti}(\text{softmax}(\frac{ind}{\gamma\sigma(c)})V^T)^T \tag{1}$$

where $Q$, $K$ and $V$ represent the queries and key-value pairs derived from the inputs through feature-dimension mapping; $\gamma$ is a positive coefficient; $\sigma$ denotes the sigmoid function; and superscript $T$ signifies transposition. The component CC serves as a quantization module that gracefully accomplishes both sequence alignment and similarity measurement in a single step.

Specifically, it is known that the time-domain convolution between sequences and their frequency-domain product forms a Fourier transform pair.

$$Q \otimes K \xleftrightarrow{\mathcal{F}} Q_f \cdot K_f \tag{2}$$

where the subscript $f$ denotes frequency-domain components, and $\otimes$ represents the convolution operation. Subsequently, by employing complex conjugation, rapid forward scanning is efficiently achieved.

$$r_{QK} \xleftrightarrow{\mathcal{F}} Q_f \cdot \bar{K}_f \tag{3}$$

Specifically, $r_{QK}$ denotes the cross-correlation sequence between $Q$ and $K$ across various combinations of time lags, where the $\bar{K}_f$ signifies the conjugation of $K_f$. By extracting the maximum value and its corresponding index from this sequence, we obtain the output of the CC module.

$$c, ind = \text{argmax}(r_{QK}) \tag{4}$$

In the preceding analysis, larger values of $ind$ and smaller values of $c$ signify that the two variables share fewer common components; consequently, they should be assigned greater weights. This quantification is realized by computing the ratio $\frac{ind}{c}$. However, since $c$ may be less than zero, leading to outcomes that contradict our expectations, we employ the sigmoid function to transform $c$ into a positive value and regulate its range through the coefficient $\gamma$. $\sigma$ helps to preserve the magnitude relationships (in case of negative correlation) as well as avoid numerical errors caused by $c$ being zero. Ultimately, by leveraging the amplifying and compressing properties of the exponential function, we achieve efficient and dynamic nonlinear feature extraction expressed as follows,

$$Out = \text{acti}(\text{softmax}(\frac{ind}{\gamma\sigma(c)})V^T)^T \tag{5}$$

Notably, before generating queries and key-value pairs, the input undergoes linear projections along both feature and temporal dimensions, drawing upon the instruction from Ailing et al. (2023). We present the meticulous structure of CEP in Figure 3. It's worth noting that when the sequence length is $L$, the conventional method of scanning to find maximum values incurs a time complexity of $\mathcal{O}(L^2)$. By leveraging the time-frequency convolution theorem, CEP performs data scanning computations in the frequency domain, reducing the computational overhead to $L\log(L)$ (Tran et al., 2023).

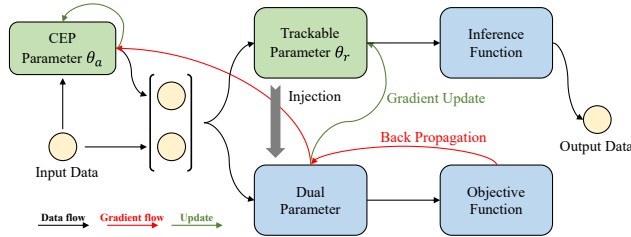

Figure 4: Schematic of COPU structure. The data flow shows the paths from input to output and loss. The gradient flow illustrates the optimization process. The CEP parameter $\theta_a$ serves as the feature extractor, while the trackable parameter $\theta_r$ functions as the feature-output mapper. $\theta_a$ is updated using standard GP technique (Gu et al., 2024), while $\theta_r$ is updated using the truncated gradient, as indicated by the update flow.

Figure 2 graphically showcases the advantages of CEP over other fundamental NN frameworks in time series feature modeling. An exemplary benchmarking method for nonlinear modeling should gracefully extract unique insights from diverse sources. As depicted in Figure 2, the simple experiment reveals that existing stacked frameworks are markedly inadequate in extracting diverse features from input. Not only is the distinctive information carried by $x_4$ overwhelmed, but the innovations embodied in $x_2$ and $x_3$ also fail to manifest. All NN structures other than CEP produce trivial, homogenized results, highlighting CEP's superiority in modeling diverse nonlinear characteristics.

## 3 COPU: Cross-correlation Enhanced Approximated Orthogonal Projection Unit

Constructing a dual parameter space to achieve a structural approximation of the natural gradient (James, 2020; Wu et al., 2023) has been proven to exert a significantly beneficial impact on model stability and convergence accuracy (Shaoqi et al., 2024).

$$\text{NGD:} \quad \theta^{(t+1)} = \theta^{(t)} - \eta \nabla_m \mathcal{L}(m) \tag{6}$$

where $\theta$ and $m$ signify the network's parameter and the corresponding dual parameter respectively; $\eta$ denotes the learning rate; $\mathcal{L}$ represents the loss; superscript $t$ denotes updating iteration. Specifically, NGD has been applied to the trackable parameter $\theta_r$ as shown in Figure 4. Shaoqi et al. (2024) has pointed out a potential and effective way of constructing the dual parameter space,

$$m = \tilde{x}\tilde{x}^T \theta_r \tag{7}$$

$$\tilde{x} = \text{concat}(x, \text{aug}(x)) \tag{8}$$

where $x \in \mathbb{R}^{d,b}$; $b$ signifies the mini-batch size and $d$ represents the feature dimensionality. Within this framework, we enhance modeling capabilities through data augmentation rather than network stacking. The dual parameters can be seen as a module specially designed to facilitate the optimization of input-output mapping, while the augmentation module focuses on diverse and informative feature extraction. However, this approach simultaneously introduces issues of numerical precision and stability during the computation of the loss function.

$$\mathcal{L} = \mathbb{E}[(y - (\tilde{x}^T \tilde{x})^{-1} \tilde{x}^T m)^2] \tag{9}$$

In this context, the invertibility of $(\tilde{x}^T \tilde{x})^{-1}$ cannot be always guaranteed; its solvability fundamentally hinges on whether the RR equals 1. By employing singular value decomposition, we can approximate this inverse when RR is less than 1; however, this approach introduces significant precision loss. As RR approaches 1, the approximation increasingly resembles the actual matrix inverse; conversely, as RR approaches 0, the approximation tends toward the matrix itself.

Therefore, it is imperative that the augmentation module possesses sufficiently robust nonlinear modeling capabilities to ensure that all samples $\tilde{x}$ within each mini-batch are linearly independent. We replace random gaussian matrix (RGM) with CEP for data augmentation, significantly enhancing the modeling capacity and stability of AOPU (Malik et al., 2023; Chen & Liu, 2018). COPU

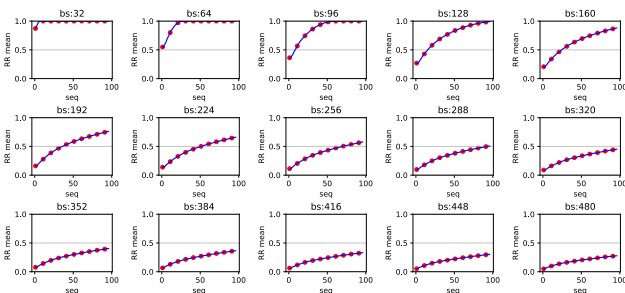

Figure 5: Curves of the mean of RR on sulfur recovery unit (SRU) under varying batch sizes (bs) and sequence length (seq) settings.

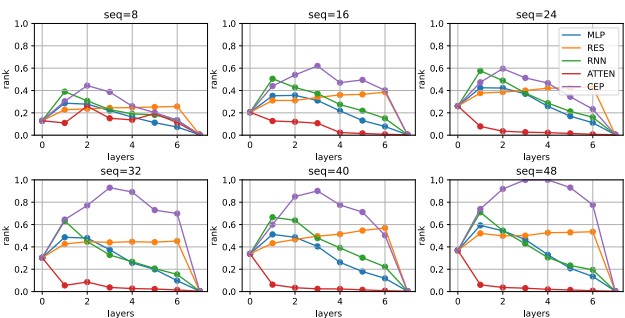

Figure 6: Curves of the mean of RR on SRU under varying sequence length and NN's structure settings. A 7-layer stacked end-to-end NN has been constructed for each structure. In each subplot, the horizontal axis represents the output of each layer from 0 to 7, while the vertical axis indicates the mean of RR distribution.

consists of two sets of parameters: the regression network parameters $\theta_r$ and the data augmentation network parameters $\theta_a$ (i.e., CEP), each optimized using different strategies as depicted in Figure 4. Specifically, we update $\theta_r$ using truncated gradients of dual parameters, while $\theta_a$ is updated by deep learning optimizers.

## 4 EXPERIMENT

In this section, we conduct a comprehensive series of experiments to qualitatively and quantitatively assess RR and COPU. We begin by elucidating the static differences among various methods in enhancing RR during the initialization phase. Subsequently, by tracking the evolution of RR across different models throughout the training process, we analyze their dynamic distinctions. By integrating these observations, we delve into the relationship between RR and model efficacy. Our experimental findings reveal a positive correlation between RR and model performance. These results are substantiated across two real-world datasets: SRU, and Debutanizer.

### 4.1 ANALYSIS OF RR ON INITIALIZATION

Figure 5, as researched by Shaoqi et al. (2024), illustrates the relationship among the RR distribution, batch size, and sequence length in SRU. The figure reveals that the phenomenon of multiple variables exhibiting specific patterns at different time lags, as previously mentioned, is indeed widespread. This is manifested by the RR often being significantly less than 1 within a mini-batch, indicating an excess of linearly correlated samples. As the sequence length increases, the sample window also expands. Consequently, samples become increasingly linearly uncorrelated, leading to an increase in RR. However, the RR remains quite low.

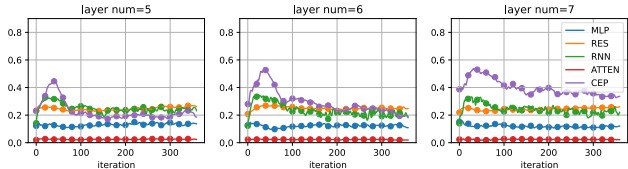

Figure 7: Curves of the mean of RR on SRU changes with training proceedings for different structures. Layer num indicates the number of stacked layers within each NN structure. The RR is calculated from the 4th layer's output.

Figure 6 presents schematic diagrams of the RR distributions from each layer's output under different sequence length settings. All NN frameworks are stacked with 7 layers, with the 7th layer being the output layer and the output dims set to 1.

From Figure 6, we can summarize several characteristics. Firstly, the outputs of CEP exhibit the highest RR across almost all layers, even though in certain cases RNN and residual connection (RES) may surpass it. Secondly, CEP is relatively sensitive to changes in sequence length. As the sequence length increases, the advantages of CEP in nonlinear modeling become more pronounced. This is attributed to CEP's utilization of the sequential characteristics of data; thus, the more apparent the data's sequential nature is, the better CEP performs. Furthermore, other algorithms, excluding CEP and RES, experience a significant decline in RR as layers are stacked, indicating they cannot effectively extract heterogeneous features from time series data. The RR of RES remains remarkably robust, demonstrating RES's superiority in extracting innovations. This offers a novel perspective that elucidates the underlying efficacy of residual connections in NNs beyond the kernel (Duvenaud et al., 2014) and gradient (Kaiming et al., 2016) explanations. Lastly, RNN has high RR when the number of stacked layers is small, explaining the widely recognized excellent performance of RNNs in time series analysis tasks.

## 4.2 ANALYSIS OF RR ON TRAINING

A larger RR indicates that the algorithm can focus on different unique information from the input, thereby aiding the model in learning more diverse and informative representations. Conversely, a lower RR forces the algorithm to concentrate on multiple copies of the same feature, increasing the risk of overfitting and resulting in poorer performance.

We closely monitor the dynamic shifts in RR distributions and performance metrics throughout the training processes of various foundational NN frameworks. By delving into these evolving patterns, we aim to unearth the subtle correlations and dependencies that underpin their performance. Figure 7 illustrates the dynamic evolution of RR distribution at output of the 4th layer over iterations. The batch size and sequence length are respectively set to 256 and 16, and different NN frameworks are stacked with varying numbers of layers. It can be observed that as the NN continues to learn, the RR of CEP, RNN, and RES initially exhibit an upward trend, reaching a peak within 10 to 20 iterations, after which they decline and stabilize. In contrast, the RR of MLP and ATTEN remain relatively unchanged during training, maintaining low values. Upon stabilization, CEP generally attains the highest RR, followed by RNN and RES, which are comparable and rank second, then MLP, and finally ATTEN. Figure 8 depicts the dynamic changes in the model's loss on the validation dataset. It reveals that increasing the number of stacked layers does not significantly enhance model performance; on the contrary, RNN, RES, and MLP exhibit substantial declines in stability, and ATTEN even shows performance deterioration.

In contrast, CEP sustains more robust iterative updates, with performance gradually improving as the number of layers increases. When the number of layers equals 2, CEP's performance is on par with other comparative methods; when the number of layers reaches 7, CEP surpasses all other NNs, both in stability and convergence accuracy.

Examining Figures 7 and 8 together uncovers intriguing characteristics. Notably, the interval during which the model's validation loss decreases rapidly closely coincides with the sharp rise in RR. The initial 20 iterations mark the rapid convergence phase for each model, during which most models exhibit a significant upward trend in RR. The observations suggest that in the early stages of training,

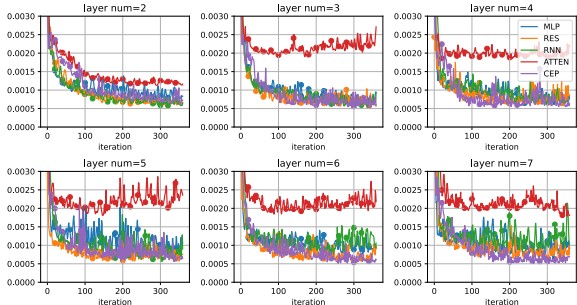

Figure 8: Curves of the val loss on SRU change with training proceedings for different structures. The curves serve as direct evaluation of the performance of each base module in end-to-end experiments. Different colors representing the different base modules are indicated in the legend. The output dimension of the network's final layer is 1. Layer num indicates the number of stacked layers within each NN structure.

NNs actively search across the wide parameter space, eagerly exploring various feature representation methods relevant to the task at hand. As training progresses and begins to stabilize, the RR settles down; however, the validation loss continues its steady decline. This indicates that the NNs are now fine-tuning the intricate mapping from latent variables to outputs, building upon the feature representation methods they previously discovered. Figure 7 and 8 imply that the window for effective learning, especially the acquisition of feature representation, is remarkably brief. Consequently, the ability of nonlinear models to extract diverse and rich information from inputs becomes critical.

### 4.3 STATISTICAL RESULT

To furnish a thorough comparison with COPU, we have selected MLP, RES, LSTM, Structured State Space (S4), Informer, and Autoformer, thereby encompassing quintessential approaches representative of fully connected networks, recurrent neural networks, and attention mechanisms. For models like MLP and RES that do not explicitly require inputs in sequence form, the inputs are flattened at the penultimate layer, and the outputs are generated through linear transformation. Regarding hyperparameter configurations, we fix the batch size at 64, the sequence length at 16, and the hidden dimension at 32. Minor fluctuations in these parameters are observed to exert minimal impact on model performance. The learning rate is set to 2E-4 for all NN models. The dual parameters' learning rate is set to 1E-1. Our experiments meticulously document performance across varying numbers of stacked layers. All NN configurations undergo 20 independent repetitions; the mean results are denoted by uppercase numerals on the left side of the table, while the standard deviations are indicated by lowercase subscripts on the right. The term E-x denotes $\times 10^{-x}$. For MAPE, the 1E-1 equals 10%. The high MAPE in Debutanizer is attributed to a certain segment where the butane content approaches zero.

**We wish to emphasize that COPU's performance can be effectively enhanced by stacking more layers, whereas the comparative methods are ill-suited for multi-layer stacking.** Table 1 shows that CEP is more suitable than conventional NN architectures for stacking to extract conducive abstract features in time series context. When the number of stacked layers increases from 2 to 5, COPU demonstrates significant performance enhancements, with improvements of 17.4% (6.94E-4→5.73E-4) and 9.4% (1.91E-2→1.73E-2) on the SRU and Debutanizer datasets, respectively. In contrast, in the comparative methods, increasing the number of stacked layers actually leads to poorer performance. For instance, S4's performance declines by 11.7% (2.13E-3→2.38E-3) and 19.2% (2.81E-2→3.35E-2); even residual connections fail to mitigate this phenomenon, as seen with RES, which experiences decreases of 17.4% (8.52E-4→1.00E-3) and 2.2% (2.30E-2→2.35E-2). Beyond its performance advantages, COPU also exhibits exceptional stability, evidenced by its standard deviation often being an order of magnitude smaller than those of other algorithms.

Naturally, COPU does show some degradation when the depth becomes excessive, but the outcome is still markedly superior than that of the comparative methods. For example, when the number of stacked layers increases from 5 to 7, RES declines by 17.0% (1.00E-3→1.17E-3) and 4.3% (2.35E-

Table 1: Comparative results of the various methods at different numbers of stacked layers on Debutanizer and SRU dataset with batch size set to 64 and sequence length set to 32.

| Model | | Dataset & Metric [†] | | | | | |
|---|---|---|---|---|---|---|---|
| | | Debutanizer | | | SRU | | |
| Stack num | Name | MSE | MAE | MAPE | MSE | MAE | MAPE |
| 2 Layers | Autoformer | $3.77\text{E-}2_{\pm7.77\text{E-}4}$ | $1.48\text{E-}1_{\pm1.45\text{E-}3}$ | $1.78\text{E+}2_{\pm1.15\text{E+}1}$ | $2.70\text{E-}3_{\pm2.13\text{E-}4}$ | $3.89\text{E-}2_{\pm1.44\text{E-}3}$ | $2.83\text{E-}1_{\pm2.06\text{E-}2}$ |
| | Informer | $3.12\text{E-}2_{\pm4.95\text{E-}3}$ | $1.35\text{E-}1_{\pm1.40\text{E-}2}$ | $1.74\text{E+}2_{\pm3.61\text{E+}1}$ | $1.19\text{E-}3_{\pm3.13\text{E-}4}$ | $2.36\text{E-}2_{\pm2.66\text{E-}3}$ | $1.68\text{E-}1_{\pm2.05\text{E-}2}$ |
| | MLP | $2.00\text{E-}2_{\pm2.68\text{E-}3}$ | $1.09\text{E-}1_{\pm6.75\text{E-}3}$ | $1.32\text{E+}2_{\pm1.35\text{E+}1}$ | $5.91\text{E-}4_{\pm7.06\text{E-}5}$ | $1.87\text{E-}2_{\pm8.71\text{E-}4}$ | $1.42\text{E-}1_{\pm8.22\text{E-}3}$ |
| | RES | $2.30\text{E-}2_{\pm4.38\text{E-}3}$ | $1.18\text{E-}1_{\pm1.27\text{E-}2}$ | $1.05\text{E+}2_{\pm2.81\text{E+}1}$ | $8.52\text{E-}4_{\pm1.48\text{E-}4}$ | $2.20\text{E-}2_{\pm2.13\text{E-}3}$ | $1.63\text{E-}1_{\pm2.10\text{E-}2}$ |
| | LSTM | $3.26\text{E-}2_{\pm3.18\text{E-}3}$ | $1.33\text{E-}1_{\pm7.27\text{E-}3}$ | $1.86\text{E+}2_{\pm2.09\text{E+}1}$ | $8.10\text{E-}4_{\pm9.94\text{E-}5}$ | $2.07\text{E-}2_{\pm9.88\text{E-}4}$ | $1.30\text{E-}1_{\pm7.05\text{E-}3}$ |
| | S4 | $2.81\text{E-}2_{\pm4.15\text{E-}3}$ | $1.29\text{E-}1_{\pm1.16\text{E-}2}$ | $1.58\text{E+}2_{\pm2.86\text{E+}1}$ | $2.13\text{E-}3_{\pm5.12\text{E-}4}$ | $3.44\text{E-}2_{\pm5.05\text{E-}3}$ | $2.30\text{E-}1_{\pm4.41\text{E-}2}$ |
| | **COPU** | $\mathbf{1.91\text{E-}2_{\pm4.03\text{E-}3}}$ | $\mathbf{1.08\text{E-}1_{\pm9.99\text{E-}3}}$ | $\mathbf{1.23\text{E+}2_{\pm2.11\text{E+}1}}$ | $\mathbf{6.94\text{E-}4_{\pm4.92\text{E-}5}}$ | $\mathbf{1.93\text{E-}2_{\pm6.37\text{E-}4}}$ | $\mathbf{1.57\text{E-}1_{\pm8.41\text{E-}3}}$ |
| 3 Layers | Autoformer | $3.67\text{E-}2_{\pm9.60\text{E-}4}$ | $1.46\text{E-}1_{\pm2.33\text{E-}3}$ | $1.80\text{E+}2_{\pm1.23\text{E+}1}$ | $2.60\text{E-}3_{\pm1.90\text{E-}4}$ | $3.82\text{E-}2_{\pm1.60\text{E-}3}$ | $2.75\text{E-}1_{\pm2.20\text{E-}2}$ |
| | Informer | $2.80\text{E-}2_{\pm3.13\text{E-}3}$ | $1.27\text{E-}1_{\pm8.85\text{E-}3}$ | $1.35\text{E+}2_{\pm3.42\text{E+}1}$ | $1.11\text{E-}3_{\pm1.94\text{E-}4}$ | $2.31\text{E-}2_{\pm1.38\text{E-}3}$ | $1.64\text{E-}1_{\pm1.66\text{E-}2}$ |
| | MLP | $2.05\text{E-}2_{\pm1.93\text{E-}3}$ | $1.12\text{E-}1_{\pm7.74\text{E-}3}$ | $1.27\text{E+}2_{\pm2.01\text{E+}1}$ | $6.09\text{E-}4_{\pm2.96\text{E-}5}$ | $1.91\text{E-}2_{\pm4.49\text{E-}4}$ | $1.43\text{E-}1_{\pm5.45\text{E-}3}$ |
| | RES | $2.36\text{E-}2_{\pm4.22\text{E-}3}$ | $1.18\text{E-}1_{\pm1.29\text{E-}2}$ | $1.10\text{E+}2_{\pm3.37\text{E+}1}$ | $9.78\text{E-}4_{\pm1.90\text{E-}4}$ | $2.31\text{E-}2_{\pm2.22\text{E-}3}$ | $1.74\text{E-}1_{\pm1.91\text{E-}2}$ |
| | LSTM | $3.08\text{E-}2_{\pm2.32\text{E-}3}$ | $1.31\text{E-}1_{\pm5.34\text{E-}3}$ | $1.52\text{E+}2_{\pm2.19\text{E+}1}$ | $9.13\text{E-}4_{\pm1.27\text{E-}4}$ | $2.11\text{E-}2_{\pm1.20\text{E-}3}$ | $1.33\text{E-}1_{\pm6.86\text{E-}3}$ |
| | S4 | $3.40\text{E-}2_{\pm3.64\text{E-}3}$ | $1.37\text{E-}1_{\pm9.03\text{E-}3}$ | $1.61\text{E+}2_{\pm2.08\text{E+}1}$ | $2.36\text{E-}3_{\pm8.51\text{E-}4}$ | $3.59\text{E-}2_{\pm8.09\text{E-}3}$ | $2.51\text{E-}1_{\pm6.61\text{E-}2}$ |
| | **COPU** | $\mathbf{1.87\text{E-}2_{\pm3.67\text{E-}3}}$ | $\mathbf{1.08\text{E-}1_{\pm1.14\text{E-}2}}$ | $\mathbf{1.24\text{E+}2_{\pm2.53\text{E+}1}}$ | $\mathbf{5.94\text{E-}4_{\pm4.39\text{E-}5}}$ | $\mathbf{1.83\text{E-}2_{\pm5.97\text{E-}4}}$ | $\mathbf{1.42\text{E-}1_{\pm7.78\text{E-}3}}$ |
| 4 Layers | Autoformer | $3.63\text{E-}2_{\pm1.80\text{E-}3}$ | $1.46\text{E-}1_{\pm2.91\text{E-}3}$ | $1.73\text{E+}2_{\pm1.50\text{E+}1}$ | $2.55\text{E-}3_{\pm1.78\text{E-}4}$ | $3.78\text{E-}2_{\pm1.12\text{E-}3}$ | $2.73\text{E-}1_{\pm1.26\text{E-}2}$ |
| | Informer | $2.89\text{E-}2_{\pm3.72\text{E-}3}$ | $1.30\text{E-}1_{\pm1.05\text{E-}2}$ | $1.22\text{E+}2_{\pm2.78\text{E+}1}$ | $1.15\text{E-}3_{\pm1.44\text{E-}4}$ | $2.37\text{E-}2_{\pm1.46\text{E-}3}$ | $1.69\text{E-}1_{\pm1.64\text{E-}2}$ |
| | MLP | $2.07\text{E-}2_{\pm3.73\text{E-}3}$ | $1.11\text{E-}1_{\pm9.31\text{E-}3}$ | $1.14\text{E+}2_{\pm1.78\text{E+}1}$ | $6.10\text{E-}4_{\pm4.83\text{E-}5}$ | $1.89\text{E-}2_{\pm9.14\text{E-}4}$ | $1.42\text{E-}1_{\pm9.17\text{E-}3}$ |
| | RES | $2.34\text{E-}2_{\pm5.59\text{E-}3}$ | $1.18\text{E-}1_{\pm1.50\text{E-}2}$ | $1.03\text{E+}2_{\pm4.59\text{E+}1}$ | $9.37\text{E-}4_{\pm1.28\text{E-}4}$ | $2.30\text{E-}2_{\pm1.80\text{E-}3}$ | $1.77\text{E-}1_{\pm1.86\text{E-}2}$ |
| | LSTM | $3.08\text{E-}2_{\pm2.42\text{E-}3}$ | $1.28\text{E-}1_{\pm5.27\text{E-}3}$ | $1.43\text{E+}2_{\pm4.21\text{E+}1}$ | $7.72\text{E-}4_{\pm1.15\text{E-}4}$ | $1.99\text{E-}2_{\pm1.13\text{E-}3}$ | $1.28\text{E-}1_{\pm5.69\text{E-}3}$ |
| | S4 | $3.31\text{E-}2_{\pm5.65\text{E-}3}$ | $1.35\text{E-}1_{\pm1.46\text{E-}2}$ | $1.37\text{E+}2_{\pm5.53\text{E+}1}$ | $2.25\text{E-}3_{\pm6.04\text{E-}4}$ | $3.53\text{E-}2_{\pm5.57\text{E-}3}$ | $2.44\text{E-}1_{\pm4.38\text{E-}2}$ |
| | **COPU** | $\mathbf{1.74\text{E-}2_{\pm2.87\text{E-}3}}$ | $\mathbf{1.04\text{E-}1_{\pm8.03\text{E-}3}}$ | $\mathbf{1.08\text{E+}2_{\pm1.94\text{E+}1}}$ | $\mathbf{5.94\text{E-}4_{\pm4.88\text{E-}5}}$ | $\mathbf{1.83\text{E-}2_{\pm4.66\text{E-}4}}$ | $\mathbf{1.46\text{E-}1_{\pm1.30\text{E-}2}}$ |
| 5 Layers | Autoformer | $3.57\text{E-}2_{\pm1.06\text{E-}3}$ | $1.46\text{E-}1_{\pm2.44\text{E-}3}$ | $1.66\text{E+}2_{\pm1.87\text{E+}1}$ | $2.60\text{E-}3_{\pm1.72\text{E-}4}$ | $3.80\text{E-}2_{\pm1.41\text{E-}3}$ | $2.76\text{E-}1_{\pm1.26\text{E-}2}$ |
| | Informer | $2.77\text{E-}2_{\pm3.65\text{E-}3}$ | $1.26\text{E-}1_{\pm1.31\text{E-}2}$ | $1.35\text{E+}2_{\pm2.97\text{E+}1}$ | $1.20\text{E-}3_{\pm1.24\text{E-}4}$ | $2.43\text{E-}2_{\pm9.70\text{E-}4}$ | $1.77\text{E-}1_{\pm7.96\text{E-}3}$ |
| | MLP | $2.11\text{E-}2_{\pm2.58\text{E-}3}$ | $1.13\text{E-}1_{\pm3.99\text{E-}3}$ | $1.23\text{E+}2_{\pm1.21\text{E+}1}$ | $5.98\text{E-}4_{\pm6.25\text{E-}5}$ | $1.83\text{E-}2_{\pm1.11\text{E-}3}$ | $1.40\text{E-}1_{\pm1.27\text{E-}2}$ |
| | RES | $2.35\text{E-}2_{\pm6.09\text{E-}3}$ | $1.19\text{E-}1_{\pm1.81\text{E-}2}$ | $1.12\text{E+}2_{\pm3.58\text{E+}1}$ | $1.00\text{E-}3_{\pm1.47\text{E-}4}$ | $2.31\text{E-}2_{\pm1.43\text{E-}3}$ | $1.78\text{E-}1_{\pm6.09\text{E-}3}$ |
| | LSTM | $2.78\text{E-}2_{\pm1.76\text{E-}3}$ | $1.26\text{E-}1_{\pm6.75\text{E-}3}$ | $1.26\text{E+}2_{\pm2.52\text{E+}1}$ | $8.23\text{E-}4_{\pm1.45\text{E-}4}$ | $2.12\text{E-}2_{\pm2.29\text{E-}3}$ | $1.36\text{E-}1_{\pm1.37\text{E-}2}$ |
| | S4 | $3.35\text{E-}2_{\pm4.99\text{E-}3}$ | $1.35\text{E-}1_{\pm1.41\text{E-}2}$ | $1.49\text{E+}2_{\pm4.28\text{E+}1}$ | $2.38\text{E-}3_{\pm6.17\text{E-}4}$ | $3.53\text{E-}2_{\pm5.49\text{E-}3}$ | $2.39\text{E-}1_{\pm3.96\text{E-}2}$ |
| | **COPU** | $\mathbf{1.73\text{E-}2_{\pm2.73\text{E-}3}}$ | $\mathbf{1.04\text{E-}1_{\pm8.80\text{E-}3}}$ | $\mathbf{1.24\text{E+}2_{\pm1.57\text{E+}1}}$ | $\mathbf{5.73\text{E-}4_{\pm4.85\text{E-}5}}$ | $\mathbf{1.81\text{E-}2_{\pm7.09\text{E-}4}}$ | $\mathbf{1.40\text{E-}1_{\pm9.26\text{E-}3}}$ |
| 6 Layers | Autoformer | $3.75\text{E-}2_{\pm7.95\text{E-}4}$ | $1.48\text{E-}1_{\pm1.78\text{E-}3}$ | $1.73\text{E+}2_{\pm3.19\text{E+}1}$ | $2.64\text{E-}3_{\pm1.63\text{E-}4}$ | $3.85\text{E-}2_{\pm1.33\text{E-}3}$ | $2.86\text{E-}1_{\pm1.70\text{E-}2}$ |
| | Informer | $2.55\text{E-}2_{\pm3.24\text{E-}3}$ | $1.18\text{E-}1_{\pm9.32\text{E-}3}$ | $1.02\text{E+}2_{\pm2.48\text{E+}1}$ | $1.67\text{E-}3_{\pm1.18\text{E-}4}$ | $2.50\text{E-}2_{\pm1.11\text{E-}3}$ | $1.85\text{E-}1_{\pm1.20\text{E-}2}$ |
| | MLP | $2.07\text{E-}2_{\pm1.65\text{E-}3}$ | $1.14\text{E-}1_{\pm5.25\text{E-}3}$ | $1.12\text{E+}2_{\pm2.19\text{E+}1}$ | $6.10\text{E-}4_{\pm9.68\text{E-}5}$ | $1.89\text{E-}2_{\pm1.23\text{E-}3}$ | $1.46\text{E-}1_{\pm1.24\text{E-}2}$ |
| | RES | $2.50\text{E-}2_{\pm6.72\text{E-}3}$ | $1.22\text{E-}1_{\pm1.40\text{E-}2}$ | $1.17\text{E+}2_{\pm4.43\text{E+}1}$ | $1.04\text{E-}3_{\pm2.29\text{E-}4}$ | $2.38\text{E-}2_{\pm2.70\text{E-}3}$ | $1.80\text{E-}1_{\pm2.78\text{E-}2}$ |
| | LSTM | $2.88\text{E-}2_{\pm3.49\text{E-}3}$ | $1.28\text{E-}1_{\pm7.30\text{E-}3}$ | $1.35\text{E+}2_{\pm1.52\text{E+}1}$ | $9.29\text{E-}4_{\pm1.83\text{E-}4}$ | $2.20\text{E-}2_{\pm2.88\text{E-}3}$ | $1.41\text{E-}1_{\pm1.88\text{E-}2}$ |
| | S4 | $3.16\text{E-}2_{\pm5.18\text{E-}3}$ | $1.33\text{E-}1_{\pm1.18\text{E-}2}$ | $1.45\text{E+}2_{\pm3.58\text{E+}1}$ | $2.76\text{E-}3_{\pm5.77\text{E-}4}$ | $3.84\text{E-}2_{\pm5.15\text{E-}3}$ | $2.73\text{E-}1_{\pm5.80\text{E-}2}$ |
| | **COPU** | $\mathbf{1.83\text{E-}2_{\pm3.06\text{E-}3}}$ | $\mathbf{1.07\text{E-}1_{\pm8.17\text{E-}3}}$ | $\mathbf{1.25\text{E+}2_{\pm1.98\text{E+}1}}$ | $\mathbf{5.82\text{E-}4_{\pm4.49\text{E-}5}}$ | $\mathbf{1.82\text{E-}2_{\pm6.49\text{E-}4}}$ | $\mathbf{1.44\text{E-}1_{\pm9.46\text{E-}3}}$ |
| 7 Layers | Autoformer | $4.11\text{E-}2_{\pm2.16\text{E-}3}$ | $1.55\text{E-}1_{\pm4.73\text{E-}3}$ | $1.83\text{E+}2_{\pm4.58\text{E+}1}$ | $2.89\text{E-}3_{\pm2.38\text{E-}4}$ | $3.95\text{E-}2_{\pm1.90\text{E-}3}$ | $3.08\text{E-}1_{\pm1.55\text{E-}2}$ |
| | Informer | $2.62\text{E-}2_{\pm2.83\text{E-}3}$ | $1.20\text{E-}1_{\pm9.69\text{E-}3}$ | $1.31\text{E+}2_{\pm1.80\text{E+}1}$ | $1.42\text{E-}3_{\pm1.87\text{E-}4}$ | $2.62\text{E-}2_{\pm1.77\text{E-}3}$ | $1.90\text{E-}1_{\pm1.37\text{E-}2}$ |
| | MLP | $2.11\text{E-}2_{\pm2.27\text{E-}3}$ | $1.14\text{E-}1_{\pm6.60\text{E-}3}$ | $1.06\text{E+}2_{\pm1.34\text{E+}1}$ | $6.26\text{E-}4_{\pm8.75\text{E-}5}$ | $1.90\text{E-}2_{\pm1.10\text{E-}3}$ | $1.41\text{E-}1_{\pm1.23\text{E-}2}$ |
| | RES | $2.45\text{E-}2_{\pm5.52\text{E-}3}$ | $1.17\text{E-}1_{\pm1.55\text{E-}2}$ | $1.15\text{E+}2_{\pm3.91\text{E+}1}$ | $1.17\text{E-}3_{\pm2.32\text{E-}4}$ | $2.54\text{E-}2_{\pm2.54\text{E-}3}$ | $1.93\text{E-}1_{\pm1.90\text{E-}2}$ |
| | LSTM | $2.91\text{E-}2_{\pm1.39\text{E-}3}$ | $1.29\text{E-}1_{\pm6.33\text{E-}3}$ | $1.21\text{E+}2_{\pm1.60\text{E+}1}$ | $9.75\text{E-}4_{\pm2.09\text{E-}4}$ | $2.23\text{E-}2_{\pm2.74\text{E-}3}$ | $1.43\text{E-}1_{\pm1.88\text{E-}2}$ |
| | S4 | $3.13\text{E-}2_{\pm5.19\text{E-}3}$ | $1.31\text{E-}1_{\pm1.45\text{E-}2}$ | $1.49\text{E+}2_{\pm5.94\text{E+}1}$ | $2.81\text{E-}3_{\pm8.66\text{E-}4}$ | $3.88\text{E-}2_{\pm7.12\text{E-}3}$ | $2.97\text{E-}1_{\pm9.24\text{E-}2}$ |
| | **COPU** | $\mathbf{1.85\text{E-}2_{\pm1.65\text{E-}3}}$ | $\mathbf{1.09\text{E-}1_{\pm4.76\text{E-}3}}$ | $\mathbf{1.24\text{E+}2_{\pm1.60\text{E+}1}}$ | $\mathbf{6.15\text{E-}4_{\pm6.78\text{E-}5}}$ | $\mathbf{1.85\text{E-}2_{\pm7.31\text{E-}4}}$ | $\mathbf{1.48\text{E-}1_{\pm1.31\text{E-}2}}$ |

[†] MSE, MAE, MAPE stand for mean squared error, mean absolute error, and mean absolute percentage error.

2→2.45E-2) on the SRU and Debutanizer datasets, respectively; LSTM decreases by 18.5% (8.23E-4→9.75E-4) and 4.7% (2.78E-2→2.91E-2), whereas COPU only drops by 7.3% (5.73E-4→6.15E-4) and 6.9% (1.73E-2→1.85E-2). The statistical result demonstrates that CEP has propelled progress and deepened the understanding of time series analysis.

## 4.4 ABLATION STUDY

This section utilizes detailed ablation experiments to validate the critical importance and indispensability of CEP to COPU. The study by Shaoqi et al. (2024) (see Figure 5) indicates that when the sequence length remains unchanged and the batch size increases, RR often declines, leading to a performance downturn in AOPU. CEP mitigates this deficiency.

**We wish to emphasize that CEP makes COPU much more robust to the change of RR while other NN structures do not. This can be attributed to CEP's ability to maintain high RR during training. The fact that the second-robust structure is RES (see Figure 6) validates this proposition.** Table 2 shows that COPU-CEP experiences the least performance decline as the batch size progressively increases, whereas AOPU-RGM is most adversely affected. When the batch

Table 2: Ablation results of various augmentation methods at different batch size settings on Debutanizer and SRU dataset with sequence length set to 32 and number of stacked layers set to 7.

| Model | | Dataset & Metric | | | | | |
|---|---|---|---|---|---|---|---|
| Batch Size | Name† | Debutanizer | | | SRU | | |
| | | MSE | MAE | MAPE | MSE | MAE | MAPE |
| 32 | AOPU-RGM | $1.75\text{E-}2_{\pm7.06\text{E-}4}$ | $9.96\text{E-}2_{\pm2.33\text{E-}3}$ | $1.50\text{E+}2_{\pm5.88\text{E+}0}$ | $8.28\text{E-}4_{\pm1.38\text{E-}5}$ | $1.96\text{E-}2_{\pm1.09\text{E-}4}$ | $2.00\text{E-}1_{\pm1.90\text{E-}3}$ |
| | COPU-MLP | $2.18\text{E-}2_{\pm2.60\text{E-}3}$ | $1.12\text{E-}1_{\pm6.38\text{E-}3}$ | $1.26\text{E+}2_{\pm1.02\text{E+}1}$ | $6.65\text{E-}4_{\pm5.18\text{E-}5}$ | $1.88\text{E-}2_{\pm5.38\text{E-}4}$ | $1.58\text{E-}1_{\pm1.11\text{E-}2}$ |
| | COPU-RES | $2.21\text{E-}2_{\pm4.60\text{E-}3}$ | $1.18\text{E-}1_{\pm1.18\text{E-}2}$ | $9.41\text{E+}1_{\pm2.01\text{E+}1}$ | $6.62\text{E-}4_{\pm7.91\text{E-}5}$ | $1.95\text{E-}2_{\pm1.37\text{E-}3}$ | $1.52\text{E-}1_{\pm1.60\text{E-}2}$ |
| | COPU-RNN | $2.02\text{E-}2_{\pm2.42\text{E-}3}$ | $1.08\text{E-}1_{\pm5.74\text{E-}3}$ | $1.32\text{E+}2_{\pm1.55\text{E+}1}$ | $7.15\text{E-}4_{\pm5.46\text{E-}5}$ | $1.88\text{E-}2_{\pm5.42\text{E-}4}$ | $1.71\text{E-}1_{\pm8.84\text{E-}3}$ |
| | COPU-ATTEN | $2.02\text{E-}2_{\pm2.63\text{E-}3}$ | $1.10\text{E-}1_{\pm6.33\text{E-}3}$ | $1.37\text{E+}2_{\pm2.08\text{E+}1}$ | $8.59\text{E-}4_{\pm5.81\text{E-}5}$ | $2.00\text{E-}2_{\pm5.34\text{E-}4}$ | $2.00\text{E-}1_{\pm1.06\text{E-}2}$ |
| | **COPU-CEP** | $\mathbf{1.77\text{E-}2_{\pm1.93\text{E-}3}}$ | $\mathbf{1.05\text{E-}1_{\pm5.72\text{E-}3}}$ | $\mathbf{1.24\text{E+}2_{\pm1.82\text{E+}1}}$ | $\mathbf{6.22\text{E-}4_{\pm5.51\text{E-}5}}$ | $\mathbf{1.86\text{E-}2_{\pm6.47\text{E-}4}}$ | $\mathbf{1.48\text{E-}1_{\pm1.00\text{E-}2}}$ |
| 64 | AOPU-RGM | $2.18\text{E-}2_{\pm3.36\text{E-}3}$ | $1.10\text{E-}1_{\pm8.06\text{E-}3}$ | $1.71\text{E+}2_{\pm1.47\text{E+}1}$ | $8.11\text{E-}4_{\pm1.30\text{E-}5}$ | $1.97\text{E-}2_{\pm1.85\text{E-}4}$ | $1.92\text{E-}1_{\pm5.85\text{E-}3}$ |
| | COPU-MLP | $2.30\text{E-}2_{\pm3.60\text{E-}3}$ | $1.15\text{E-}1_{\pm8.88\text{E-}3}$ | $1.21\text{E+}2_{\pm1.71\text{E+}1}$ | $5.74\text{E-}4_{\pm3.36\text{E-}5}$ | $1.76\text{E-}2_{\pm4.06\text{E-}4}$ | $1.43\text{E-}1_{\pm8.06\text{E-}3}$ |
| | COPU-RES | $2.60\text{E-}2_{\pm9.25\text{E-}3}$ | $1.22\text{E-}1_{\pm1.79\text{E-}2}$ | $1.14\text{E+}2_{\pm3.57\text{E+}1}$ | $6.90\text{E-}4_{\pm7.59\text{E-}5}$ | $1.99\text{E-}2_{\pm1.17\text{E-}3}$ | $1.58\text{E-}1_{\pm1.09\text{E-}2}$ |
| | COPU-RNN | $2.14\text{E-}2_{\pm3.38\text{E-}3}$ | $1.12\text{E-}1_{\pm6.72\text{E-}3}$ | $1.42\text{E+}2_{\pm1.19\text{E+}1}$ | $6.20\text{E-}4_{\pm2.98\text{E-}5}$ | $1.80\text{E-}2_{\pm3.06\text{E-}4}$ | $1.54\text{E-}1_{\pm7.45\text{E-}3}$ |
| | COPU-ATTEN | $2.10\text{E-}2_{\pm3.15\text{E-}3}$ | $1.13\text{E-}1_{\pm8.80\text{E-}3}$ | $1.36\text{E+}2_{\pm3.22\text{E+}1}$ | $8.55\text{E-}4_{\pm3.48\text{E-}5}$ | $2.00\text{E-}2_{\pm4.14\text{E-}4}$ | $2.00\text{E-}1_{\pm7.33\text{E-}3}$ |
| | **COPU-CEP** | $\mathbf{1.75\text{E-}2_{\pm1.63\text{E-}3}}$ | $\mathbf{1.05\text{E-}1_{\pm5.21\text{E-}3}}$ | $\mathbf{1.18\text{E+}2_{\pm1.84\text{E+}1}}$ | $\mathbf{5.94\text{E-}4_{\pm5.13\text{E-}5}}$ | $\mathbf{1.84\text{E-}2_{\pm7.14\text{E-}4}}$ | $\mathbf{1.41\text{E-}1_{\pm8.58\text{E-}3}}$ |
| 128 | AOPU-RGM | $3.44\text{E-}2_{\pm4.28\text{E-}3}$ | $1.38\text{E-}1_{\pm8.05\text{E-}3}$ | $1.77\text{E+}2_{\pm2.26\text{E+}1}$ | $9.30\text{E-}4_{\pm4.62\text{E-}5}$ | $2.33\text{E-}2_{\pm8.21\text{E-}4}$ | $1.76\text{E-}1_{\pm6.90\text{E-}3}$ |
| | COPU-MLP | $2.94\text{E-}2_{\pm4.97\text{E-}3}$ | $1.30\text{E-}1_{\pm8.92\text{E-}3}$ | $1.43\text{E+}2_{\pm2.40\text{E+}1}$ | $5.84\text{E-}4_{\pm3.52\text{E-}5}$ | $1.80\text{E-}2_{\pm5.86\text{E-}4}$ | $1.33\text{E-}1_{\pm6.10\text{E-}3}$ |
| | COPU-RES | $2.50\text{E-}2_{\pm4.32\text{E-}3}$ | $1.24\text{E-}1_{\pm8.08\text{E-}3}$ | $1.10\text{E+}2_{\pm3.35\text{E+}1}$ | $7.60\text{E-}4_{\pm6.64\text{E-}5}$ | $2.09\text{E-}2_{\pm1.27\text{E-}3}$ | $1.62\text{E-}1_{\pm1.26\text{E-}2}$ |
| | COPU-RNN | $2.64\text{E-}2_{\pm5.15\text{E-}3}$ | $1.24\text{E-}1_{\pm1.03\text{E-}2}$ | $1.41\text{E+}2_{\pm2.44\text{E+}1}$ | $6.48\text{E-}4_{\pm3.18\text{E-}5}$ | $1.87\text{E-}2_{\pm5.45\text{E-}4}$ | $1.37\text{E-}1_{\pm5.20\text{E-}3}$ |
| | COPU-ATTEN | $2.58\text{E-}2_{\pm6.13\text{E-}3}$ | $1.23\text{E-}1_{\pm1.59\text{E-}2}$ | $1.40\text{E+}2_{\pm3.40\text{E+}1}$ | $8.79\text{E-}4_{\pm4.81\text{E-}5}$ | $2.23\text{E-}2_{\pm6.68\text{E-}4}$ | $1.66\text{E-}1_{\pm6.26\text{E-}3}$ |
| | **COPU-CEP** | $\mathbf{1.76\text{E-}2_{\pm2.75\text{E-}3}}$ | $\mathbf{1.06\text{E-}1_{\pm8.20\text{E-}3}}$ | $\mathbf{1.05\text{E+}2_{\pm2.29\text{E+}1}}$ | $\mathbf{6.54\text{E-}4_{\pm6.12\text{E-}5}}$ | $\mathbf{1.91\text{E-}2_{\pm5.60\text{E-}4}}$ | $\mathbf{1.50\text{E-}1_{\pm1.01\text{E-}2}}$ |
| 256 | AOPU-RGM | $3.72\text{E-}2_{\pm5.08\text{E-}3}$ | $1.47\text{E-}1_{\pm8.10\text{E-}3}$ | $1.71\text{E+}2_{\pm2.03\text{E+}1}$ | $3.07\text{E-}3_{\pm6.53\text{E-}5}$ | $4.06\text{E-}2_{\pm4.74\text{E-}4}$ | $3.63\text{E-}1_{\pm5.71\text{E-}3}$ |
| | COPU-MLP | $3.92\text{E-}2_{\pm6.62\text{E-}3}$ | $1.48\text{E-}1_{\pm1.16\text{E-}2}$ | $1.70\text{E+}2_{\pm3.15\text{E+}1}$ | $3.31\text{E-}3_{\pm2.86\text{E-}4}$ | $4.19\text{E-}2_{\pm1.79\text{E-}3}$ | $3.74\text{E-}1_{\pm1.90\text{E-}2}$ |
| | COPU-RES | $3.10\text{E-}2_{\pm8.83\text{E-}3}$ | $1.31\text{E-}1_{\pm1.23\text{E-}2}$ | $1.30\text{E+}2_{\pm3.87\text{E+}1}$ | $9.77\text{E-}4_{\pm1.17\text{E-}4}$ | $2.29\text{E-}2_{\pm1.60\text{E-}3}$ | $1.61\text{E-}1_{\pm1.46\text{E-}2}$ |
| | COPU-RNN | $3.78\text{E-}2_{\pm8.21\text{E-}3}$ | $1.46\text{E-}1_{\pm1.38\text{E-}2}$ | $1.73\text{E+}2_{\pm2.57\text{E+}1}$ | $2.51\text{E-}3_{\pm1.63\text{E-}4}$ | $3.70\text{E-}2_{\pm1.78\text{E-}3}$ | $3.15\text{E-}1_{\pm2.18\text{E-}2}$ |
| | COPU-ATTEN | $4.01\text{E-}2_{\pm7.08\text{E-}3}$ | $1.51\text{E-}1_{\pm1.04\text{E-}2}$ | $1.69\text{E+}2_{\pm3.19\text{E+}1}$ | $2.50\text{E-}3_{\pm1.26\text{E-}4}$ | $3.72\text{E-}2_{\pm1.65\text{E-}3}$ | $3.14\text{E-}1_{\pm1.58\text{E-}2}$ |
| | **COPU-CEP** | $\mathbf{2.30\text{E-}2_{\pm4.82\text{E-}3}}$ | $\mathbf{1.21\text{E-}1_{\pm1.22\text{E-}2}}$ | $\mathbf{1.40\text{E+}2_{\pm3.46\text{E+}1}}$ | $\mathbf{9.74\text{E-}4_{\pm2.58\text{E-}4}}$ | $\mathbf{2.25\text{E-}2_{\pm1.72\text{E-}3}}$ | $\mathbf{1.69\text{E-}1_{\pm1.81\text{E-}2}}$ |

† The term COPU-MLP refers to the COPU model augmented with a Multilayer Perceptron as its augmentation model. AOPU utilizes an RGM as its augmentation model.

size equals 32, RR is relatively high, and there is no significant performance difference between AOPU and COPU. However, as the batch size rises to 256, the RR in the data markedly decreases; AOPU is impacted, with its MSE loss increasing by 270.8% (8.28E-4→3.07E-3) and 112.6%(1.75E-2→3.72E-2). In contrast, COPU-CEP exhibits minimal performance fluctuation owing to its ability to maintain RR at a high level.

Compared to COPU-CEP, variants like COPU-RNN, COPU-MLP, COPU-ATTEN, and COPU-RES perform worse and cannot adapt to changes in RR when the batch size increases. The underlying reason is that their NN architectures have not deeply explored the structural characteristics of sequential data. As the training proceeds, their RR stays relatively low thereby causing much precision compromises in objective loss and gradient propagation.

## 5 CONCLUSION

This paper proposes a new NN component termed CEP that has strong nonlinear representation performance in time series analysis and serves as the foundation for developing the COPU framework. An inherent distinction between time series data and image or text data lies in the ambiguity of its discriminative patterns: two wide different sequences may exert similar influences on system outputs, while two nearly identical sequences can produce different outcomes. Consequently, constructing effective latent features from such inputs is exceedingly intricate. Simply applying methods from CV or NLP to time series analysis fails to yield satisfactory results, as empirical studies have demonstrated both qualitatively and quantitatively. By focusing on and leveraging sequential characteristics, CEP effectively extracts innovations from similar features, thereby exhibiting enhanced nonlinear modeling prowess. Integrating CEP with AOPU, we propose COPU; CEP augments RR capabilities while resolving the computational precision and expressive power issues inherent in AOPU. Experimental results reveal that COPU significantly outperforms comparative methods, and ablation studies further underscore the critical importance and indispensability of CEP. Although CEP is not compatible with other modality data due to its specialization, such focused research can more effectively propel disciplinary development and advance the field, while also illuminating the potential of research on domain-wise characterization rather than general solutions.

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

## A    ADDITIONAL FIGURE

In this section, we unveil a wealth of additional experimental results, primarily extending the analyses presented in Figures 6, 7, and 8. For example, Figure 7 is actually the lower-half segment of Figure 12. Figure 9, 10, 11, 12, 13, and 14 show the RR result of different layer's output. The core of this expansion lies in revealing further patterns of RR variation across outputs from different layers, as well as replicating all experiments on the Debutanizer process.

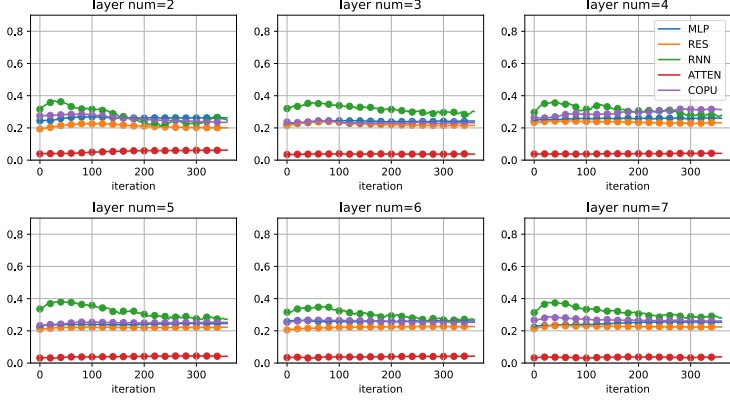

Figure 9:    Curves of the mean of RR on SRU changes with training proceedings for different structures. Layer num indicates the number of stacked layers within each NN structure. The RR is calculated from the 1st layer's output.

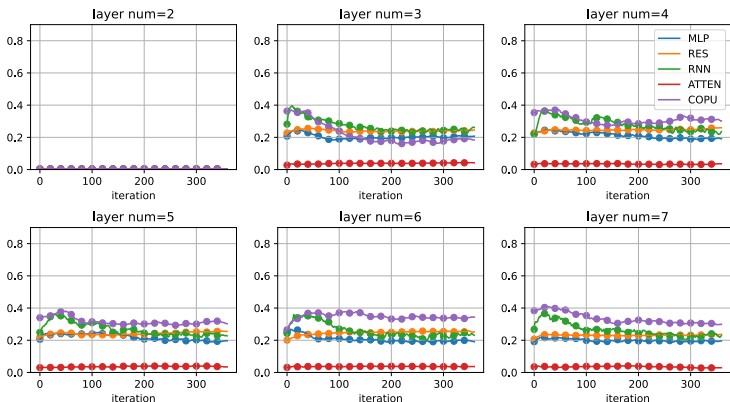

Figure 10:   Curves of the mean of RR on SRU changes with training proceedings for different structures. Layer num indicates the number of stacked layers within each NN structure. The RR is calculated from the 2nd layer's output.

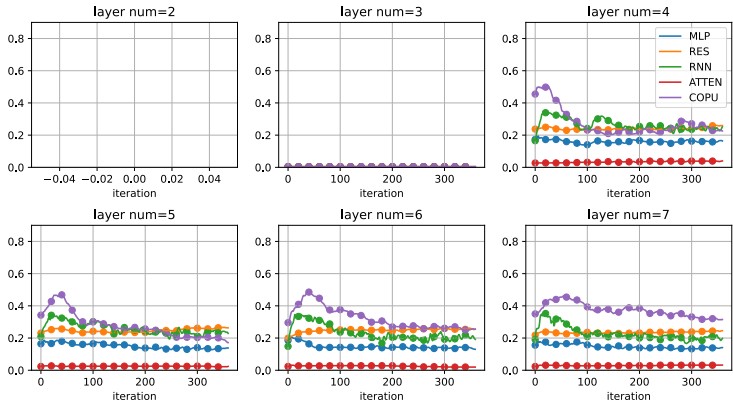

Figure 11:   Curves of the mean of RR on SRU changes with training proceedings for different structures. Layer num indicates the number of stacked layers within each NN structure. The RR is calculated from the 3th layer's output.

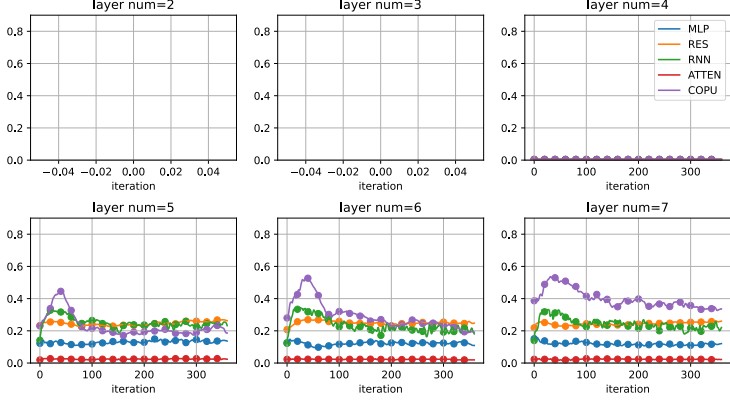

Figure 12:   Curves of the mean of RR on SRU changes with training proceedings for different structures. Layer num indicates the number of stacked layers within each NN structure. The RR is calculated from the 4th layer's output.

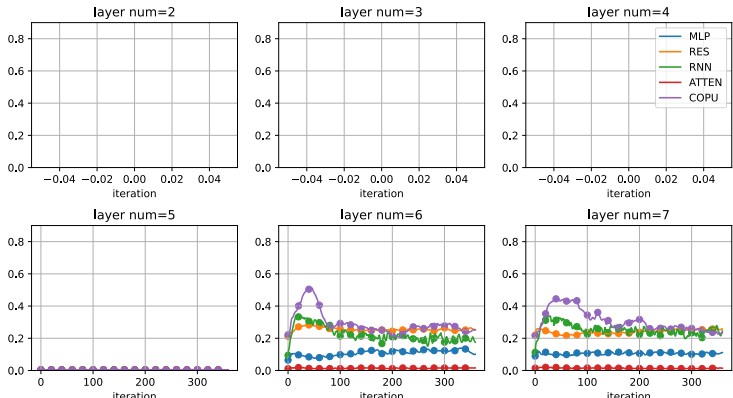

Figure 13: Curves of the mean of RR on SRU changes with training proceedings for different structures. Layer num indicates the number of stacked layers within each NN structure. The RR is calculated from the 5th layer's output.

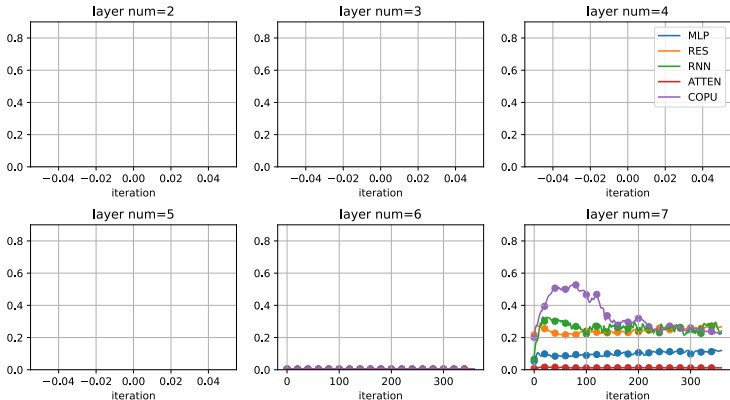

Figure 14: Curves of the mean of RR on SRU changes with training proceedings for different structures. Layer num indicates the number of stacked layers within each NN structure. The RR is calculated from the 6th layer's output.

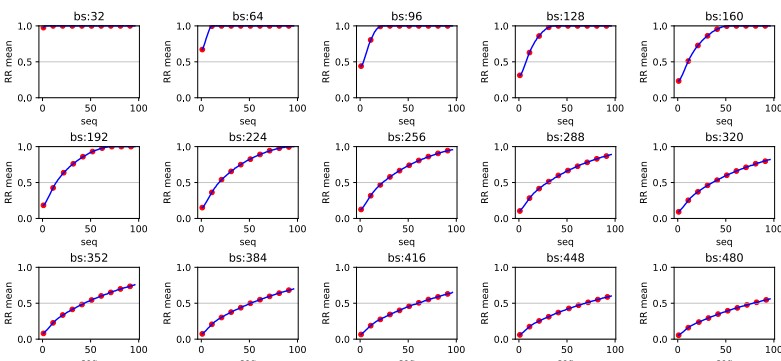

Figure 15: Curves of the mean of RR on Debutanizer under varying batch sizes (bs) and sequence length (seq) settings. In each subplot, the horizontal axis represents sequence length, while the vertical axis indicates the mean of RR. The batch size increases from left to right and from top to bottom.

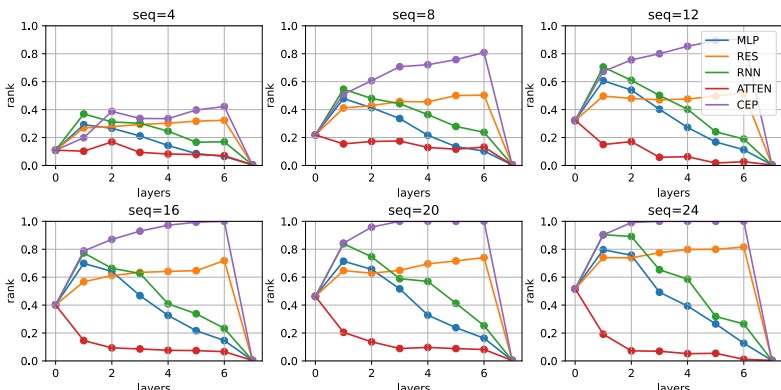

Figure 16: Curves of the mean of RR on Debutanizer under varying sequence length and NN's structure settings. A 7-layer stacked end-to-end NN has been constructed for each structure. In each subplot, the horizontal axis represents the output of each layer from 0 to 7, while the vertical axis indicates the mean of RR distribution.

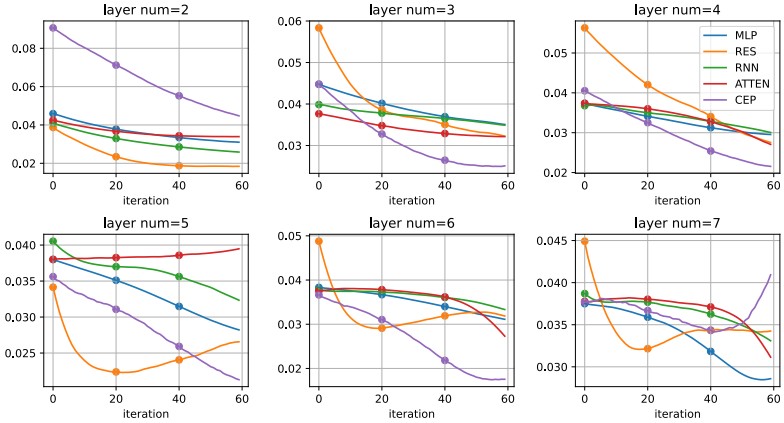

Figure 17: Curves of the val loss on Debutanizer change with training proceedings for different structures. Layer num indicates the number of stacked layers within each NN structure.

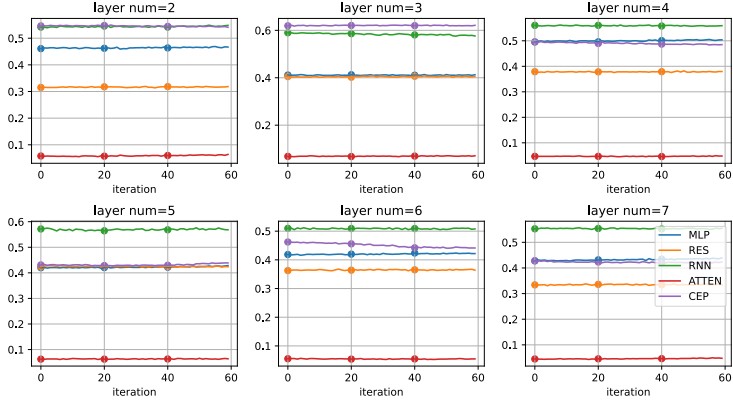

Figure 18: Curves of the mean of RR on Debutanizer changes with training proceedings for different structures. Layer num indicates the number of stacked layers within each NN structure. The RR is calculated from the 1st layer's output.

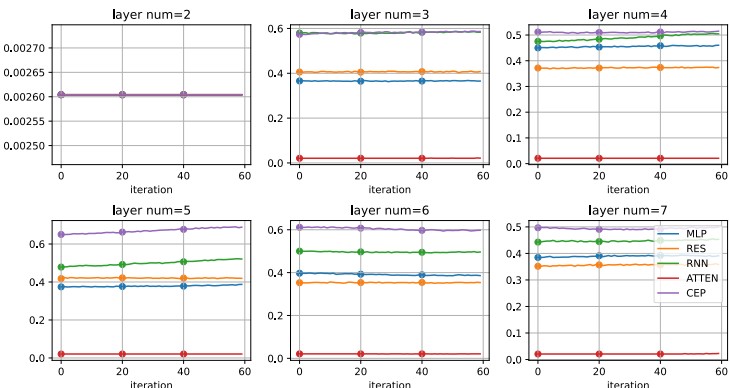

Figure 19: Curves of the mean of RR on Debutanizer changes with training proceedings for different structures. Layer num indicates the number of stacked layers within each NN structure. The RR is calculated from the 2nd layer's output.

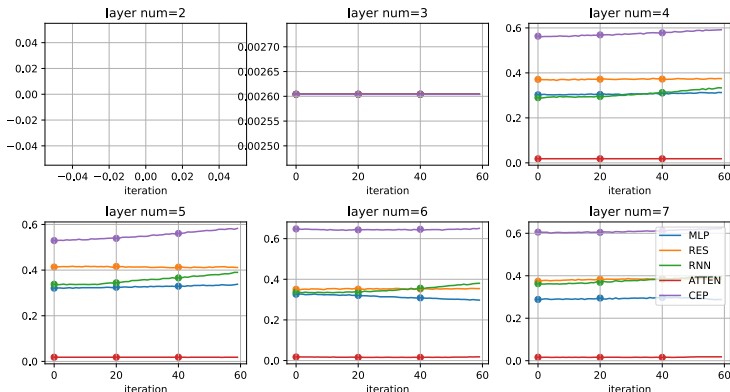

Figure 20: Curves of the mean of RR on Debutanizer changes with training proceedings for different structures. Layer num indicates the number of stacked layers within each NN structure. The RR is calculated from the 3rd layer's output.

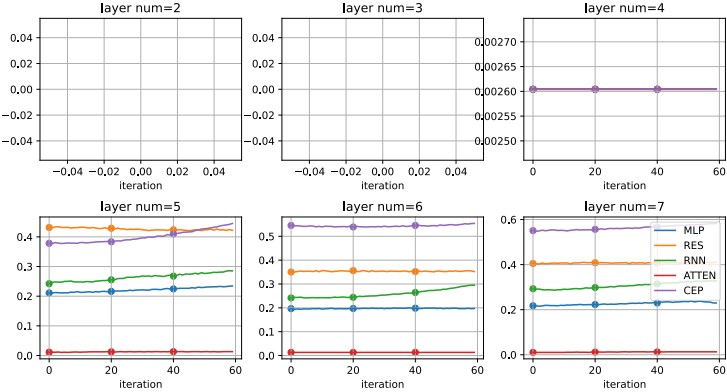

Figure 21: Curves of the mean of RR on Debutanizer changes with training proceedings for different structures. Layer num indicates the number of stacked layers within each NN structure. The RR is calculated from the 4th layer's output.

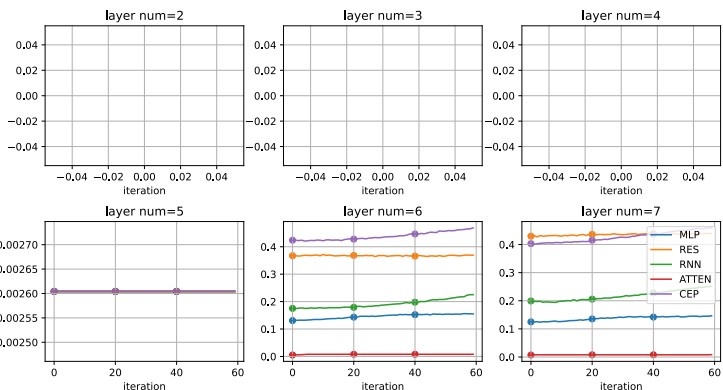

Figure 22: Curves of the mean of RR on Debutanizer changes with training proceedings for different structures. Layer num indicates the number of stacked layers within each NN structure. The RR is calculated from the 5th layer's output.

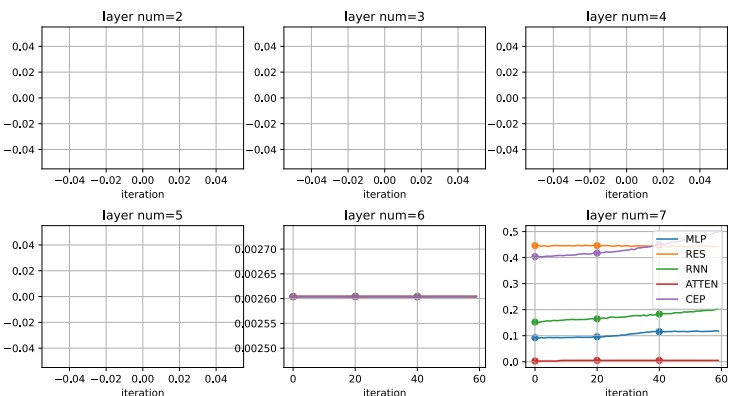

Figure 23: Curves of the mean of RR on Debutanizer changes with training proceedings for different structures. Layer num indicates the number of stacked layers within each NN structure. The RR is calculated from the 6th layer's output.

## B  ADDITIONAL RESULT

In the main text, we primarily analyze the characteristics of CEP using SRU as a case study. This section supplements our findings with experimental results on Debutanizer to enhance the paper's completeness and credibility. We aim to emphasize two points: first, compared to SRU, Debutanizer is more informative and presents a greater challenge for model fitting; second, CEP is not introduced as an end-to-end NN. These points explain why CEP's performance on Debutanizer does not fully align with the previous analyses. Most importantly, CEP successfully compensates for the AOPU's heavy reliance on RR, enabling COPU to be applicable across a wider range of batch sizes and sequence lengths.

Figure 15 illustrates how the RR distribution of Debutanizer varies with changes in batch size and sequence length. We observe that Debutanizer's RR is significantly higher than that of SRU under equivalent settings. This indicates that Debutanizer inherently contains more heterogeneous information and is informative even without CEP. Figure 16 further validates this point; when the sequence length is 8, the RR at each NN layer is notably higher than the results shown in Figure 6. The small step size for sequence length in Figure 16 is intentionally chosen to prevent the Debutanizer's RR from rapidly converging to 1. That would obscure presenting CEP's advantage in enhancing RR. Debutanizer is considerably more difficult to fit than the SRU, as reflected not only in the MAPE in Tables 1 and 2 but also demonstrated in numerous studies (Luigi et al., 2007).

CEP functions as an RR enhancement module rather than an end-to-end regression network. Figure 17 shows the dynamic progression of validation loss for each NN, revealing that CEP's output can be suboptimal, even bad performing. This can be attributed to CEP's excessive focus on amplifying heterogeneous information, thereby neglecting the modeling of input-output relationships. **We would like to point out that CEP is introduced as a feature extraction module instead of an end-to-end input-output mapping module. The nonlinear modeling capability is prioritized in this context.** Figures 18–23 demonstrate that CEP still effectively enhances RR, allowing COPU to maintain minimal performance loss with larger batch sizes (i.e., smaller RR) as shown in Table 2.

## C DATASET DESCRIPTION

Table 3: Variable Description

| Debutanizer | | | SRU | | |
|---|---|---|---|---|---|
| Process Variables | Unit | Description | Process Variables | Unit | Description |
| $U_1$ | $^\circ C$ | Top temperature | $U_1$ | $m^3 \cdot h^{-1}$ | Gas flow MEA_GAS |
| $U_2$ | $kg \cdot cm^{-2}$ | Top pressure | $U_2$ | $m^3 \cdot h^{-1}$ | Air flow AIR_MEA |
| $U_3$ | $m^3 \cdot h^{-1}$ | Reflux flow | $U_3$ | $m^3 \cdot h^{-1}$ | Secondary air flow AIR_MEA_2 |
| $U_4$ | $m^3 \cdot h^{-1}$ | Flow to next process | $U_4$ | $m^3 \cdot h^{-1}$ | Gas flow in SWS zone |
| $U_5$ | $^\circ C$ | $6^{th}$ temperature | $U_5$ | $m^3 \cdot h^{-1}$ | Air flow in SWS zone |
| $U_6$ | $^\circ C$ | Bottom temperature A | | | |
| $U_7$ | $^\circ C$ | Bottom temperature B | | | |

The Debutanizer column is part of a desulfuring and naphtha splitter plant (Luigi et al., 2007). It is required to maximize the C5 (stabilized gasoline) content in the Debutanizer overheads(LP gas splitter feed) and minimize the C4 (butane) content in the Debutanizer bottoms (Naphtha splitter feed). However, the butane content is not directly measured on the bottom flow, but on the overheads of the downstream deisopentanizer column by the gas chromatograph resulting in a large measuring delay, which is the reason soft sensor steps in Zhichao et al. (2023). The dataset comprises 2,394 records, each featuring 7 relevant sensor measurements. Detailed information about inputs can be found in Table 3.

The sulfur recovery unit (SRU) removes environmental pollutants from acid gas streams before they are released into the atmosphere (Luigi et al., 2007). On-line analyzers are used to measure the concentration of both hydrogen sulfide and sulfur dioxide in the tail gas of each sulfur line. Hydrogen sulfide and sulfur dioxide frequently cause damage to sensors, which often have to be removed for maintenance. Soft sensors are introduced to address this issue (Yuan et al., 2020). The dataset comprises 10,080 records, each featuring 5 relevant sensor measurements. Detailed information about inputs can also be found in Table 3.

For both datasets, we allocate the initial 80% of samples to form the training set. From the remaining data, the next 10% constitutes the validation set, while the final 10% is designated as the test set.

## D ADDITIONAL ANALYSIS

To further validate the universality of the RR issue and the superiority of COPU, we conduct experiments on several public time-series datasets. The experimental results corroborate our previous conjecture, further emphasizing the critical importance of understanding the modality characteristics of time-series.

It is important to note that the prior experimental results on SRU and Debutanizer presented unnormalized outcomes, i.e., all metrics are calculated on the original scales of the datasets. In contrast, the results metrics discussed here are normalized, with the output on the scale of a standard normal distribution. Additionally, the datasets utilized here are commonly employed in contemporary research for testing prediction tasks, but in this paper, they are used to evaluate regression tasks. Therefore, hyperparameters such as label length and predict length are not applicable.

Figures 24 and 25 illustrate that the RR issue indeed exists across various application scenarios in the field of time series. As the batch size increases, all datasets exhibit a significant decrease in RR; conversely, as the sequence length increases, they consistently display a gradual increase in RR.

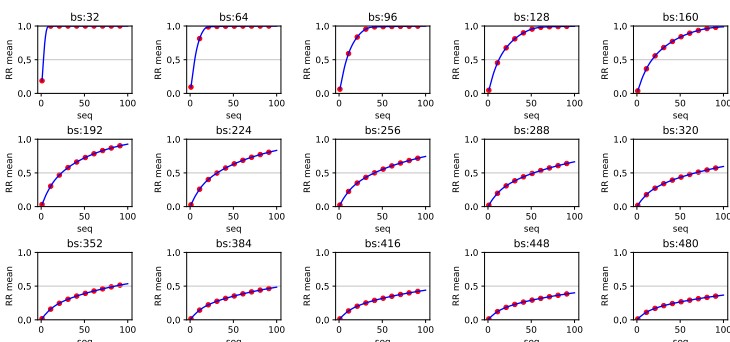

Figure 24: Curves of the mean of RR on ETTm2 under varying bs and seq settings.

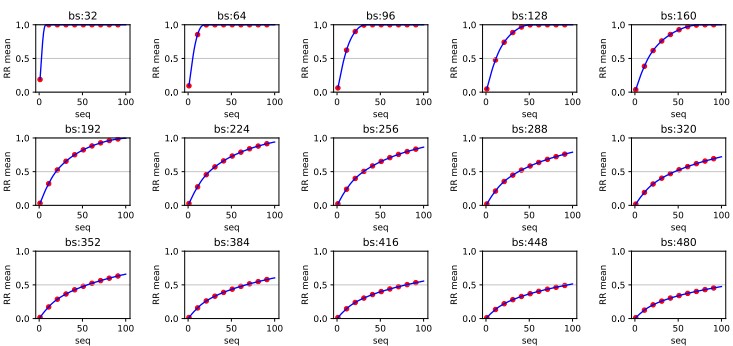

Figure 25: Curves of the mean of RR on ETTh2 under varying bs and seq settings.

Taking the ETTm2 dataset in Figure 24 as an example, when the sequence length is set to 50 and the batch size is 256, the RR is approximately 0.5. This indicates that, on average, only 128 samples within a mini-batch are independent and effective. This phenomenon aligns with the case when the batch size is set to 128, resulting in an RR of approximately 1.

**Remark**: *In fact, when the sequence length is set to 50 and the batch size is 256, the RR is approximately 0.552. This indicates that, on average, only 55.2% samples within a mini-batch are independent and effective samples. Conversely, when the batch size is 128, the RR increases to approximately 0.951, yielding 122 effective samples. The reason RR does not strictly adhere to proportional transformations is that it is a locally rather than globally determined characteristic within a mini batch.*

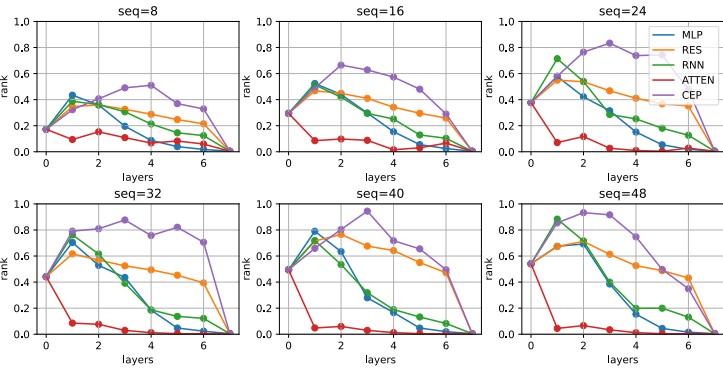

Figure 26: Curves of the mean of RR on ETTm2 under varying seq and NN's settings.

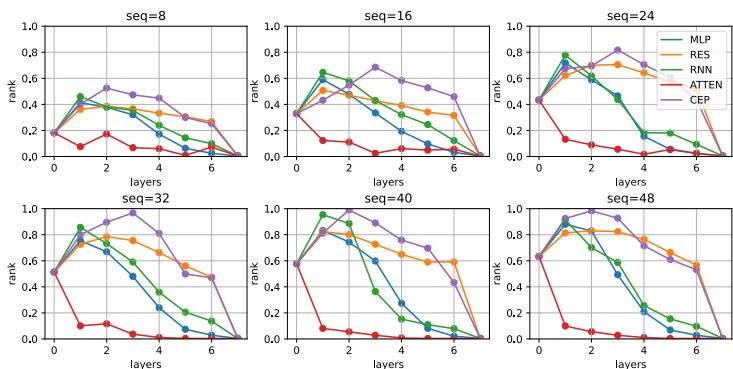

Figure 27: Curves of the mean of RR on ETTh2 under varying seq and NN's settings.

The issue of RR highlights the significant difference between time-series data and CV/NLP modalities, that the time-series data are less informative. The pronounced decrease in RR as batch size increases indicates that the unique information contained within samples diminishes, making them more easily represented linearly by other samples. This phenomenon has actually been implicitly recognized in the field of industrial system identification. For example, when applying Ordinary Least Squares (OLS), there is a stringent requirement for the number of samples to vastly exceed the dimensionality of features, precisely because it is challenging to ensure the invertibility of $xx^T$. Similarly, Partial Least Squares (PLS) is proposed as an identification tool to refine heterogeneous information within samples while suppressing homogeneous ones.

It is this characteristic of time-series data that renders the trivial application of models designed for CV/NLP modalities inappropriate. Modules such as Attention aims to focus on features of interest from vast and diverse information. However, in the context of time series, they might extract multiple copies of the same pattern from large amounts of homogeneous information. Such modules lack the ability to regularize the diversity of the extracted features. Figure 26 and 27 and Table 1 to 4 illustrate this point both qualitatively and quantitatively. We believe that a deeper understanding of RR can help transcend the limitations of existing models and ultimately lead to the design of network architectures specifically tailored for time-series analysis.

**Remark**: *The issue of RR further reveals a potential question. In the domains of CV/NLP, increasing the batch size can reduce gradient variance, thereby aiding the model in optimizing toward a global optimum. However, in the field of time series, expanding the batch size may not change the number of effective samples, which may provoke a series of questions and considerations.*

The impact of label consistency between redundant and effective samples presents an intriguing question that remains unclear and merits further investigation. Redundant samples refer to the samples constituting the difference between the batch size and the number of effective samples. Essentially, these are linear combinations of effective samples, prompting us to ponder whether their labels satisfy such linear relationships. If their labels do satisfy these linear relationships, which essentially implies embodying a linear model prior, will this cause networks to tend toward degenerating into linear models? Conversely, if their labels do not satisfy such linear relationships, will this encourage networks to learn the nonlinear dynamic characteristics inherent in the data, or will it lead to conflicts resulting in unstable updates? We believe that these reflections based on the RR issue can deepen our understanding of applying NNs in the realm of time-series analysis.

The ability of CEP to enhance RR suggests its potentially superior nonlinear modeling capabilities. RR itself represents the proportion of linearly independent samples within a mini-batch; its variation with batch size reflects the modality's informativeness, while its variation with the number of stacked layers indicates the network's nonlinear modeling capacity. The essence of nonlinearity is transforming linearly dependent (but not identical) elements into linearly independent ones, i.e., $\text{acti}(a + b) \neq \text{acti}(a) + \text{acti}(b)$. Therefore, an increase in RR implies that the NN transforms originally linearly dependent components into independent ones, implicitly measuring its nonlinear modeling ability. By utilizing CEP, COPU can maintain a high RR even when stacking multiple network layers, thereby achieving better performance, as illustrated in Figures 26 and 27. Another

explicit application of CEP in COPU is that, when computing Equation 9, a higher RR leads to greater approximation accuracy, enhancing the robustness of the network's performance, as shown in Table 2.

Experimental results demonstrate that COPU exhibits the best performance. This is particularly evident as the number of stacked network layers increases; other comparative methods experience significant performance degradation, whereas COPU remains robust and even improves. This phenomenon is most pronounced on the ETTh2 dataset. When the stack number is 2 or 3, COPU is not even the best-performing model. However, as the stack number increases to 4 and 5, the performance of other models declines sharply, making COPU the optimal choice. When the stack number reaches 6 and 7, COPU not only holds a significant advantage but also achieves the best performance across all configurations.

Table 4: Comparative results of the various methods at different numbers of stacked layers on ETTm2 and ETTh2 datasets.

| Model | | Dataset & Metric [†] | | | | | |
| --- | --- | --- | --- | --- | --- | --- | --- |
| Stack num | Name | ETTm2 | | | ETTh2 | | |
| | | MSE | MAE | MAPE | MSE | MAE | MAPE |
| 2 Layers | Autoformer | $1.02E+0_{\pm4.97E-2}$ | $8.76E-1_{\pm2.85E-2}$ | $9.94E-1_{\pm5.15E-2}$ | $8.15E-1_{\pm2.98E-2}$ | $7.76E-1_{\pm2.07E-2}$ | $9.72E-1_{\pm3.58E-2}$ |
| | Informer | $3.25E-1_{\pm6.92E-2}$ | $4.90E-1_{\pm5.55E-2}$ | $1.26E+0_{\pm1.89E-1}$ | $2.21E-1_{\pm9.39E-3}$ | $3.81E-1_{\pm1.67E-2}$ | $1.25E+0_{\pm2.02E-1}$ |
| | MLP | $2.47E-1_{\pm8.78E-3}$ | $4.16E-1_{\pm1.28E-2}$ | $1.07E+0_{\pm7.93E-2}$ | $1.58E-1_{\pm2.04E-2}$ | $3.09E-1_{\pm2.01E-2}$ | $9.62E-1_{\pm5.75E-2}$ |
| | RES | $2.52E-1_{\pm1.45E-2}$ | $4.10E-1_{\pm2.34E-2}$ | $1.00E+0_{\pm1.40E-1}$ | $1.62E-1_{\pm4.16E-3}$ | $3.21E-1_{\pm4.62E-3}$ | $9.20E-1_{\pm7.72E-2}$ |
| | LSTM | $2.69E-1_{\pm1.36E-2}$ | $4.31E-1_{\pm1.19E-2}$ | $1.12E+0_{\pm8.06E-2}$ | $1.98E-1_{\pm2.48E-2}$ | $3.55E-1_{\pm3.33E-2}$ | $9.13E-1_{\pm1.30E-1}$ |
| | S4 | $3.66E-1_{\pm7.22E-2}$ | $5.06E-1_{\pm5.58E-2}$ | $1.16E+0_{\pm2.94E-1}$ | $2.63E-1_{\pm5.65E-2}$ | $4.08E-1_{\pm4.40E-2}$ | $1.11E+0_{\pm2.22E-1}$ |
| | **COPU** | $\mathbf{2.35E\text{-}1}_{\pm\mathbf{2.77E\text{-}2}}$ | $\mathbf{3.96E\text{-}1}_{\pm\mathbf{2.21E\text{-}2}}$ | $\mathbf{9.89E\text{-}1}_{\pm\mathbf{3.94E\text{-}2}}$ | $\mathbf{1.78E\text{-}1}_{\pm\mathbf{2.03E\text{-}2}}$ | $\mathbf{3.38E\text{-}1}_{\pm\mathbf{2.09E\text{-}2}}$ | $\mathbf{9.30E\text{-}1}_{\pm\mathbf{3.57E\text{-}2}}$ |
| 3 Layers | Autoformer | $1.01E+0_{\pm7.72E-2}$ | $8.74E-1_{\pm4.19E-2}$ | $1.00E+0_{\pm7.35E-2}$ | $8.21E-1_{\pm8.41E-2}$ | $7.86E-1_{\pm5.13E-2}$ | $9.78E-1_{\pm6.94E-2}$ |
| | Informer | $3.25E-1_{\pm3.31E-2}$ | $4.75E-1_{\pm1.87E-2}$ | $1.15E+0_{\pm2.21E-1}$ | $2.51E-1_{\pm1.88E-2}$ | $4.03E-1_{\pm1.23E-2}$ | $1.10E+0_{\pm2.59E-1}$ |
| | MLP | $2.56E-1_{\pm1.75E-2}$ | $4.29E-1_{\pm1.49E-2}$ | $1.13E+0_{\pm6.35E-2}$ | $1.61E-1_{\pm1.11E-2}$ | $3.09E-1_{\pm1.32E-2}$ | $9.82E-1_{\pm4.94E-2}$ |
| | RES | $2.78E-1_{\pm5.21E-2}$ | $4.34E-1_{\pm3.92E-2}$ | $1.05E+0_{\pm1.40E-1}$ | $1.68E-1_{\pm3.68E-2}$ | $3.23E-1_{\pm3.71E-2}$ | $9.29E-1_{\pm1.23E-1}$ |
| | LSTM | $2.77E-1_{\pm5.31E-3}$ | $4.39E-1_{\pm1.31E-2}$ | $1.13E+0_{\pm8.64E-2}$ | $2.19E-1_{\pm3.80E-2}$ | $3.73E-1_{\pm3.82E-2}$ | $1.00E+0_{\pm1.93E-1}$ |
| | S4 | $3.72E-1_{\pm5.85E-2}$ | $5.02E-1_{\pm4.62E-2}$ | $1.09E+0_{\pm2.02E-1}$ | $2.62E-1_{\pm3.65E-2}$ | $4.11E-1_{\pm3.84E-2}$ | $1.09E+0_{\pm2.68E-1}$ |
| | **COPU** | $\mathbf{2.43E\text{-}1}_{\pm\mathbf{1.50E\text{-}2}}$ | $\mathbf{4.03E\text{-}1}_{\pm\mathbf{9.58E\text{-}3}}$ | $\mathbf{9.72E\text{-}1}_{\pm\mathbf{8.51E\text{-}2}}$ | $\mathbf{1.65E\text{-}1}_{\pm\mathbf{6.71E\text{-}3}}$ | $\mathbf{3.23E\text{-}1}_{\pm\mathbf{8.77E\text{-}3}}$ | $\mathbf{9.77E\text{-}1}_{\pm\mathbf{4.98E\text{-}2}}$ |
| 4 Layers | Autoformer | $9.98E-1_{\pm4.20E-2}$ | $8.65E-1_{\pm2.31E-2}$ | $9.92E-1_{\pm3.97E-2}$ | $7.66E-1_{\pm1.46E-1}$ | $7.44E-1_{\pm9.63E-2}$ | $9.30E-1_{\pm9.18E-2}$ |
| | Informer | $2.94E-1_{\pm4.40E-2}$ | $4.56E-1_{\pm3.72E-2}$ | $1.18E+0_{\pm2.06E-1}$ | $2.49E-1_{\pm2.59E-2}$ | $4.03E-1_{\pm2.04E-2}$ | $1.14E+0_{\pm2.37E-1}$ |
| | MLP | $2.66E-1_{\pm1.54E-2}$ | $4.41E-1_{\pm1.65E-2}$ | $1.12E+0_{\pm1.45E-1}$ | $1.79E-1_{\pm1.53E-2}$ | $3.42E-1_{\pm1.48E-2}$ | $1.06E+0_{\pm5.67E-2}$ |
| | RES | $2.95E-1_{\pm3.75E-2}$ | $4.47E-1_{\pm2.81E-2}$ | $1.09E+0_{\pm1.49E-1}$ | $1.99E-1_{\pm1.12E-2}$ | $3.59E-1_{\pm1.50E-2}$ | $9.43E-1_{\pm4.72E-2}$ |
| | LSTM | $3.03E-1_{\pm3.80E-2}$ | $4.58E-1_{\pm3.70E-2}$ | $1.15E+0_{\pm1.48E-1}$ | $2.22E-1_{\pm2.32E-2}$ | $3.72E-1_{\pm2.10E-2}$ | $1.07E+0_{\pm2.00E-1}$ |
| | S4 | $3.39E-1_{\pm2.66E-2}$ | $4.87E-1_{\pm2.91E-2}$ | $1.08E+0_{\pm1.16E-1}$ | $2.79E-1_{\pm9.43E-2}$ | $4.20E-1_{\pm6.73E-2}$ | $1.08E+0_{\pm2.35E-1}$ |
| | **COPU** | $\mathbf{2.49E\text{-}1}_{\pm\mathbf{3.10E\text{-}2}}$ | $\mathbf{4.07E\text{-}1}_{\pm\mathbf{2.62E\text{-}2}}$ | $\mathbf{9.87E\text{-}1}_{\pm\mathbf{9.83E\text{-}2}}$ | $\mathbf{1.60E\text{-}1}_{\pm\mathbf{5.45E\text{-}3}}$ | $\mathbf{3.18E\text{-}1}_{\pm\mathbf{6.50E\text{-}3}}$ | $\mathbf{9.53E\text{-}1}_{\pm\mathbf{4.62E\text{-}2}}$ |
| 5 Layers | Autoformer | $9.70E-1_{\pm1.02E-1}$ | $8.42E-1_{\pm5.62E-2}$ | $9.96E-1_{\pm7.14E-2}$ | $7.99E-1_{\pm1.44E-1}$ | $7.64E-1_{\pm1.03E-1}$ | $1.00E+0_{\pm4.08E-2}$ |
| | Informer | $2.65E-1_{\pm6.00E-2}$ | $4.30E-1_{\pm4.60E-2}$ | $1.06E+0_{\pm1.21E-1}$ | $2.19E-1_{\pm3.25E-2}$ | $3.76E-1_{\pm2.71E-2}$ | $1.05E+0_{\pm9.81E-2}$ |
| | MLP | $2.70E-1_{\pm1.04E-2}$ | $4.41E-1_{\pm9.31E-3}$ | $1.16E+0_{\pm7.23E-2}$ | $1.92E-1_{\pm1.13E-2}$ | $3.54E-1_{\pm1.14E-2}$ | $1.15E+0_{\pm6.36E-2}$ |
| | RES | $2.90E-1_{\pm4.14E-2}$ | $4.40E-1_{\pm3.41E-2}$ | $1.03E+0_{\pm9.82E-2}$ | $2.05E-1_{\pm1.10E-2}$ | $3.62E-1_{\pm8.70E-3}$ | $9.27E-1_{\pm1.18E-1}$ |
| | LSTM | $2.63E-1_{\pm1.63E-2}$ | $4.26E-1_{\pm2.17E-2}$ | $1.03E+0_{\pm1.12E-1}$ | $2.27E-1_{\pm2.02E-2}$ | $3.74E-1_{\pm8.07E-3}$ | $1.09E+0_{\pm1.38E-1}$ |
| | S4 | $3.91E-1_{\pm1.10E-1}$ | $5.25E-1_{\pm6.25E-2}$ | $1.30E+0_{\pm3.22E-1}$ | $2.62E-1_{\pm3.54E-2}$ | $4.02E-1_{\pm2.42E-2}$ | $1.00E+0_{\pm1.16E-1}$ |
| | **COPU** | $\mathbf{2.47E\text{-}1}_{\pm\mathbf{1.14E\text{-}2}}$ | $\mathbf{4.05E\text{-}1}_{\pm\mathbf{6.67E\text{-}3}}$ | $\mathbf{9.84E\text{-}1}_{\pm\mathbf{5.54E\text{-}2}}$ | $\mathbf{1.63E\text{-}1}_{\pm\mathbf{1.24E\text{-}2}}$ | $\mathbf{3.22E\text{-}1}_{\pm\mathbf{1.35E\text{-}2}}$ | $\mathbf{9.48E\text{-}1}_{\pm\mathbf{4.65E\text{-}2}}$ |
| 6 Layers | Autoformer | $1.05E+0_{\pm4.77E-2}$ | $8.87E-1_{\pm3.06E-2}$ | $1.05E+0_{\pm5.85E-2}$ | $8.81E-1_{\pm4.13E-2}$ | $8.09E-1_{\pm3.06E-2}$ | $1.05E+0_{\pm7.16E-2}$ |
| | Informer | $2.62E-1_{\pm4.07E-2}$ | $4.25E-1_{\pm3.46E-2}$ | $1.04E+0_{\pm1.74E-1}$ | $2.48E-1_{\pm4.74E-2}$ | $4.01E-1_{\pm3.50E-2}$ | $1.17E+0_{\pm1.92E-1}$ |
| | MLP | $2.78E-1_{\pm3.05E-2}$ | $4.40E-1_{\pm2.58E-2}$ | $1.10E+0_{\pm1.03E-1}$ | $1.88E-1_{\pm1.25E-2}$ | $3.52E-1_{\pm1.12E-2}$ | $1.15E+0_{\pm3.38E-2}$ |
| | RES | $3.01E-1_{\pm2.92E-2}$ | $4.49E-1_{\pm2.10E-2}$ | $1.08E+0_{\pm9.98E-2}$ | $2.12E-1_{\pm3.12E-2}$ | $3.68E-1_{\pm2.78E-2}$ | $8.48E-1_{\pm9.62E-2}$ |
| | LSTM | $2.99E-1_{\pm5.82E-2}$ | $4.50E-1_{\pm4.79E-2}$ | $1.06E+0_{\pm1.63E-1}$ | $2.39E-1_{\pm1.67E-2}$ | $3.84E-1_{\pm1.30E-2}$ | $1.10E+0_{\pm1.64E-1}$ |
| | S4 | $4.24E-1_{\pm9.37E-2}$ | $5.38E-1_{\pm7.09E-2}$ | $1.12E+0_{\pm3.13E-1}$ | $2.73E-1_{\pm3.33E-2}$ | $4.15E-1_{\pm2.33E-2}$ | $1.14E+0_{\pm2.74E-1}$ |
| | **COPU** | $\mathbf{2.47E\text{-}1}_{\pm\mathbf{1.46E\text{-}2}}$ | $\mathbf{4.05E\text{-}1}_{\pm\mathbf{1.21E\text{-}2}}$ | $\mathbf{1.00E+0}_{\pm\mathbf{9.61E\text{-}2}}$ | $\mathbf{1.59E\text{-}1}_{\pm\mathbf{6.70E\text{-}3}}$ | $\mathbf{3.17E\text{-}1}_{\pm\mathbf{8.37E\text{-}3}}$ | $\mathbf{9.66E\text{-}1}_{\pm\mathbf{2.61E\text{-}2}}$ |
| 7 Layers | Autoformer | $1.10E+0_{\pm1.42E-1}$ | $8.86E-1_{\pm7.45E-2}$ | $1.17E+0_{\pm7.99E-2}$ | $8.83E-1_{\pm2.75E-2}$ | $8.10E-1_{\pm2.20E-2}$ | $1.03E+0_{\pm3.71E-2}$ |
| | Informer | $2.71E-1_{\pm3.14E-2}$ | $4.39E-1_{\pm2.89E-2}$ | $1.13E+0_{\pm9.05E-2}$ | $2.32E-1_{\pm3.45E-2}$ | $3.85E-1_{\pm2.65E-2}$ | $1.02E+0_{\pm8.52E-2}$ |
| | MLP | $2.83E-1_{\pm1.40E-2}$ | $4.55E-1_{\pm1.52E-2}$ | $1.19E+0_{\pm1.06E-1}$ | $1.90E-1_{\pm1.16E-2}$ | $3.51E-1_{\pm7.41E-3}$ | $1.12E+0_{\pm3.85E-2}$ |
| | RES | $2.97E-1_{\pm5.89E-2}$ | $4.40E-1_{\pm4.78E-2}$ | $1.05E+0_{\pm1.86E-1}$ | $2.17E-1_{\pm4.55E-2}$ | $3.68E-1_{\pm3.78E-2}$ | $9.16E-1_{\pm7.42E-2}$ |
| | LSTM | $2.84E-1_{\pm3.15E-2}$ | $4.36E-1_{\pm1.90E-2}$ | $1.07E+0_{\pm1.04E-1}$ | $2.18E-1_{\pm1.54E-2}$ | $3.70E-1_{\pm5.56E-3}$ | $1.14E+0_{\pm1.17E-1}$ |
| | S4 | $4.01E-1_{\pm5.89E-2}$ | $5.15E-1_{\pm2.39E-2}$ | $9.77E-1_{\pm2.57E-1}$ | $2.89E-1_{\pm3.11E-2}$ | $4.33E-1_{\pm2.03E-2}$ | $1.25E+0_{\pm1.74E-1}$ |
| | **COPU** | $\mathbf{2.40E\text{-}1}_{\pm\mathbf{4.05E\text{-}3}}$ | $\mathbf{4.01E\text{-}1}_{\pm\mathbf{6.58E\text{-}3}}$ | $\mathbf{1.01E+0}_{\pm\mathbf{4.65E\text{-}2}}$ | $\mathbf{1.51E\text{-}1}_{\pm\mathbf{6.75E\text{-}3}}$ | $\mathbf{3.12E\text{-}1}_{\pm\mathbf{7.48E\text{-}3}}$ | $\mathbf{9.52E\text{-}1}_{\pm\mathbf{2.57E\text{-}2}}$ |

[†] MSE, MAE, MAPE stand for mean squared error, mean absolute error, and mean absolute percentage error.

To highlight the advantages of COPU, we have conducted additional experiments involving state-of-the-art time series forecasting models. We selected Pyraformer, SCINet, TimeMixer, and TimesNet, and their experimental results are presented in Tables 5 and 6. Please note that we replaced all temporal embeddings with positional embeddings and added an extra linear projection to ensure that the output is one-dimensional.

Table 5: Additional results of the SOTA prediction methods at different numbers of stacked layers on Debutanizer and SRU datasets.

| Model | | | Dataset & Metric | | | | |
|---|---|---|---|---|---|---|---|
| Stack num | Name | Debutanizer | | | SRU | | |
| | | MSE | MAE | MAPE | MSE | MAE | MAPE |
| 2 Layers | Pyraformer | $4.33\text{E-}2_{\pm1.40\text{E-}2}$ | $1.59\text{E-}1_{\pm2.55\text{E-}2}$ | $1.70\text{E+}2_{\pm4.53\text{E+}1}$ | $2.71\text{E-}3_{\pm3.38\text{E-}4}$ | $3.78\text{E-}2_{\pm3.09\text{E-}3}$ | $2.75\text{E-}1_{\pm3.30\text{E-}2}$ |
| | SCINet | $3.03\text{E-}2_{\pm9.25\text{E-}3}$ | $1.30\text{E-}1_{\pm1.98\text{E-}2}$ | $1.16\text{E+}2_{\pm4.89\text{E+}1}$ | $1.30\text{E-}3_{\pm4.05\text{E-}4}$ | $2.70\text{E-}2_{\pm4.88\text{E-}3}$ | $2.04\text{E-}1_{\pm3.56\text{E-}2}$ |
| | TimeMixer | $3.98\text{E-}2_{\pm1.15\text{E-}2}$ | $1.50\text{E-}1_{\pm2.27\text{E-}2}$ | $1.61\text{E+}2_{\pm5.88\text{E+}1}$ | $2.27\text{E-}3_{\pm6.52\text{E-}4}$ | $3.38\text{E-}2_{\pm5.64\text{E-}3}$ | $2.61\text{E-}1_{\pm5.53\text{E-}2}$ |
| | TimesNet | $3.69\text{E-}2_{\pm9.85\text{E-}3}$ | $1.47\text{E-}1_{\pm2.17\text{E-}2}$ | $1.47\text{E+}2_{\pm5.29\text{E+}1}$ | $2.31\text{E-}3_{\pm5.01\text{E-}4}$ | $3.41\text{E-}2_{\pm3.43\text{E-}3}$ | $2.51\text{E-}1_{\pm2.33\text{E-}2}$ |
| 3 Layers | Pyraformer | $4.53\text{E-}2_{\pm1.34\text{E-}2}$ | $1.62\text{E-}1_{\pm2.53\text{E-}2}$ | $1.70\text{E+}2_{\pm4.67\text{E+}1}$ | $2.62\text{E-}3_{\pm3.21\text{E-}4}$ | $3.68\text{E-}2_{\pm3.35\text{E-}3}$ | $2.71\text{E-}1_{\pm3.09\text{E-}2}$ |
| | SCINet | $3.30\text{E-}2_{\pm1.30\text{E-}2}$ | $1.36\text{E-}1_{\pm2.87\text{E-}2}$ | $1.22\text{E+}2_{\pm5.27\text{E+}1}$ | $1.24\text{E-}3_{\pm2.82\text{E-}4}$ | $2.62\text{E-}2_{\pm3.44\text{E-}3}$ | $1.95\text{E-}1_{\pm2.57\text{E-}2}$ |
| | TimeMixer | $4.42\text{E-}2_{\pm1.46\text{E-}2}$ | $1.57\text{E-}1_{\pm2.59\text{E-}2}$ | $1.63\text{E+}2_{\pm5.09\text{E+}1}$ | $2.01\text{E-}3_{\pm8.79\text{E-}4}$ | $3.23\text{E-}2_{\pm7.46\text{E-}3}$ | $2.52\text{E-}1_{\pm6.14\text{E-}2}$ |
| | TimesNet | $4.29\text{E-}2_{\pm1.34\text{E-}2}$ | $1.61\text{E-}1_{\pm2.40\text{E-}2}$ | $1.38\text{E+}2_{\pm5.13\text{E+}1}$ | $2.27\text{E-}3_{\pm3.14\text{E-}4}$ | $3.44\text{E-}2_{\pm3.07\text{E-}3}$ | $2.58\text{E-}1_{\pm2.60\text{E-}2}$ |
| 4 Layers | Pyraformer | $4.54\text{E-}2_{\pm1.02\text{E-}2}$ | $1.64\text{E-}1_{\pm1.95\text{E-}2}$ | $1.70\text{E+}2_{\pm5.19\text{E+}1}$ | $2.88\text{E-}3_{\pm5.06\text{E-}4}$ | $3.90\text{E-}2_{\pm4.50\text{E-}3}$ | $2.97\text{E-}1_{\pm4.63\text{E-}2}$ |
| | SCINet | $3.44\text{E-}2_{\pm1.13\text{E-}2}$ | $1.41\text{E-}1_{\pm2.48\text{E-}2}$ | $1.15\text{E+}2_{\pm5.83\text{E+}1}$ | $1.21\text{E-}3_{\pm3.14\text{E-}4}$ | $2.61\text{E-}2_{\pm3.76\text{E-}3}$ | $1.98\text{E-}1_{\pm2.69\text{E-}2}$ |
| | TimeMixer | $3.72\text{E-}2_{\pm1.50\text{E-}2}$ | $1.44\text{E-}1_{\pm3.28\text{E-}2}$ | $1.58\text{E+}2_{\pm6.82\text{E+}1}$ | $1.93\text{E-}3_{\pm5.01\text{E-}4}$ | $3.18\text{E-}2_{\pm4.42\text{E-}3}$ | $2.43\text{E-}1_{\pm4.07\text{E-}2}$ |
| | TimesNet | $4.22\text{E-}2_{\pm1.17\text{E-}2}$ | $1.57\text{E-}1_{\pm2.11\text{E-}2}$ | $1.62\text{E+}2_{\pm6.31\text{E+}1}$ | $2.36\text{E-}3_{\pm5.34\text{E-}4}$ | $3.52\text{E-}2_{\pm4.07\text{E-}3}$ | $2.61\text{E-}1_{\pm4.29\text{E-}2}$ |
| 5 Layers | Pyraformer | $4.35\text{E-}2_{\pm1.21\text{E-}2}$ | $1.58\text{E-}1_{\pm2.46\text{E-}2}$ | $1.73\text{E+}2_{\pm5.07\text{E+}1}$ | $3.03\text{E-}3_{\pm4.74\text{E-}4}$ | $4.02\text{E-}2_{\pm3.90\text{E-}3}$ | $3.11\text{E-}1_{\pm5.20\text{E-}2}$ |
| | SCINet | $3.30\text{E-}2_{\pm1.17\text{E-}2}$ | $1.41\text{E-}1_{\pm2.67\text{E-}2}$ | $1.13\text{E+}2_{\pm5.89\text{E+}1}$ | $1.31\text{E-}3_{\pm4.41\text{E-}4}$ | $2.71\text{E-}2_{\pm5.18\text{E-}3}$ | $2.04\text{E-}1_{\pm3.59\text{E-}2}$ |
| | TimeMixer | $4.62\text{E-}2_{\pm1.32\text{E-}2}$ | $1.62\text{E-}1_{\pm2.14\text{E-}2}$ | $1.74\text{E+}2_{\pm5.53\text{E+}1}$ | $1.77\text{E-}3_{\pm5.69\text{E-}4}$ | $3.04\text{E-}2_{\pm4.93\text{E-}3}$ | $2.37\text{E-}1_{\pm4.56\text{E-}2}$ |
| | TimesNet | $4.23\text{E-}2_{\pm1.25\text{E-}2}$ | $1.59\text{E-}1_{\pm2.48\text{E-}2}$ | $1.47\text{E+}2_{\pm6.00\text{E+}1}$ | $2.26\text{E-}3_{\pm4.98\text{E-}4}$ | $3.41\text{E-}2_{\pm4.63\text{E-}3}$ | $2.47\text{E-}1_{\pm3.10\text{E-}2}$ |
| 6 Layers | Pyraformer | $3.78\text{E-}2_{\pm1.03\text{E-}2}$ | $1.50\text{E-}1_{\pm2.13\text{E-}2}$ | $1.70\text{E+}2_{\pm5.57\text{E+}1}$ | $3.19\text{E-}3_{\pm5.07\text{E-}4}$ | $4.15\text{E-}2_{\pm3.92\text{E-}3}$ | $3.24\text{E-}1_{\pm4.35\text{E-}2}$ |
| | SCINet | $3.25\text{E-}2_{\pm1.23\text{E-}2}$ | $1.39\text{E-}1_{\pm2.51\text{E-}2}$ | $1.22\text{E+}2_{\pm5.48\text{E+}1}$ | $1.25\text{E-}3_{\pm3.76\text{E-}4}$ | $2.63\text{E-}2_{\pm4.33\text{E-}3}$ | $1.99\text{E-}1_{\pm3.41\text{E-}2}$ |
| | TimeMixer | $3.99\text{E-}2_{\pm1.40\text{E-}2}$ | $1.51\text{E-}1_{\pm2.81\text{E-}2}$ | $1.62\text{E+}2_{\pm5.40\text{E+}1}$ | $1.92\text{E-}3_{\pm6.21\text{E-}4}$ | $3.19\text{E-}2_{\pm5.86\text{E-}3}$ | $2.45\text{E-}1_{\pm4.31\text{E-}2}$ |
| | TimesNet | $4.49\text{E-}2_{\pm1.31\text{E-}2}$ | $1.63\text{E-}1_{\pm2.49\text{E-}2}$ | $1.72\text{E+}2_{\pm6.67\text{E+}1}$ | $2.16\text{E-}3_{\pm3.92\text{E-}4}$ | $3.34\text{E-}2_{\pm3.31\text{E-}3}$ | $2.48\text{E-}1_{\pm2.75\text{E-}2}$ |
| 7 Layers | Pyraformer | $4.55\text{E-}2_{\pm1.30\text{E-}2}$ | $1.62\text{E-}1_{\pm2.39\text{E-}2}$ | $1.70\text{E+}2_{\pm4.65\text{E+}1}$ | $3.29\text{E-}3_{\pm5.78\text{E-}4}$ | $4.18\text{E-}2_{\pm4.41\text{E-}3}$ | $3.27\text{E-}1_{\pm4.92\text{E-}2}$ |
| | SCINet | $3.39\text{E-}2_{\pm1.25\text{E-}2}$ | $1.41\text{E-}1_{\pm2.67\text{E-}2}$ | $1.02\text{E+}2_{\pm5.25\text{E+}1}$ | $1.18\text{E-}3_{\pm1.66\text{E-}4}$ | $2.58\text{E-}2_{\pm2.29\text{E-}3}$ | $1.94\text{E-}1_{\pm2.02\text{E-}2}$ |
| | TimeMixer | $3.85\text{E-}2_{\pm1.42\text{E-}2}$ | $1.49\text{E-}1_{\pm2.82\text{E-}2}$ | $1.51\text{E+}2_{\pm4.44\text{E+}1}$ | $1.61\text{E-}3_{\pm6.17\text{E-}4}$ | $2.95\text{E-}2_{\pm5.75\text{E-}3}$ | $2.20\text{E-}1_{\pm4.07\text{E-}2}$ |
| | TimesNet | $4.11\text{E-}2_{\pm1.18\text{E-}2}$ | $1.56\text{E-}1_{\pm2.36\text{E-}2}$ | $1.66\text{E+}2_{\pm3.84\text{E+}1}$ | $2.17\text{E-}3_{\pm5.40\text{E-}4}$ | $3.36\text{E-}2_{\pm4.50\text{E-}3}$ | $2.47\text{E-}1_{\pm3.54\text{E-}2}$ |

Table 6: Additional results of the SOTA prediction methods at different numbers of stacked layers on ETTm2 and ETTh2 datasets.

| Model | | | Dataset & Metric | | | | |
|---|---|---|---|---|---|---|---|
| Stack num | Name | ETTm2 | | | ETTh2 | | |
| | | MSE | MAE | MAPE | MSE | MAE | MAPE |
| 2 Layers | Pyraformer | $4.13\text{E-}1_{\pm1.08\text{E-}1}$ | $5.12\text{E-}1_{\pm7.21\text{E-}2}$ | $8.12\text{E-}1_{\pm6.90\text{E-}2}$ | $2.81\text{E-}1_{\pm1.11\text{E-}1}$ | $4.14\text{E-}1_{\pm8.98\text{E-}2}$ | $8.30\text{E-}1_{\pm1.25\text{E-}1}$ |
| | SCINet | $5.64\text{E-}1_{\pm2.31\text{E-}1}$ | $5.82\text{E-}1_{\pm1.28\text{E-}1}$ | $9.17\text{E-}1_{\pm9.57\text{E-}2}$ | $2.72\text{E-}1_{\pm7.72\text{E-}2}$ | $4.00\text{E-}1_{\pm5.50\text{E-}2}$ | $9.34\text{E-}1_{\pm8.35\text{E-}2}$ |
| | TimeMixer | $5.84\text{E-}1_{\pm2.81\text{E-}1}$ | $6.27\text{E-}1_{\pm1.73\text{E-}1}$ | $8.89\text{E-}1_{\pm1.26\text{E-}1}$ | $2.47\text{E-}1_{\pm1.07\text{E-}1}$ | $3.78\text{E-}1_{\pm8.22\text{E-}2}$ | $9.14\text{E-}1_{\pm1.52\text{E-}1}$ |
| | TimesNet | $3.94\text{E-}1_{\pm1.74\text{E-}1}$ | $5.00\text{E-}1_{\pm1.08\text{E-}1}$ | $8.97\text{E-}1_{\pm1.01\text{E-}1}$ | $1.62\text{E-}1_{\pm9.06\text{E-}3}$ | $3.18\text{E-}1_{\pm1.02\text{E-}2}$ | $9.27\text{E-}1_{\pm7.16\text{E-}2}$ |
| 3 Layers | Pyraformer | $6.63\text{E-}1_{\pm3.79\text{E-}1}$ | $6.64\text{E-}1_{\pm2.31\text{E-}1}$ | $9.33\text{E-}1_{\pm1.87\text{E-}1}$ | $1.84\text{E-}1_{\pm3.39\text{E-}2}$ | $3.33\text{E-}1_{\pm2.84\text{E-}2}$ | $9.47\text{E-}1_{\pm5.69\text{E-}2}$ |
| | SCINet | $4.89\text{E-}1_{\pm2.02\text{E-}1}$ | $5.43\text{E-}1_{\pm1.11\text{E-}1}$ | $9.57\text{E-}1_{\pm1.04\text{E-}1}$ | $3.03\text{E-}1_{\pm8.11\text{E-}2}$ | $4.22\text{E-}1_{\pm6.03\text{E-}2}$ | $9.49\text{E-}1_{\pm5.51\text{E-}2}$ |
| | TimeMixer | $6.03\text{E-}1_{\pm3.74\text{E-}1}$ | $6.26\text{E-}1_{\pm2.26\text{E-}1}$ | $9.30\text{E-}1_{\pm1.41\text{E-}1}$ | $3.22\text{E-}1_{\pm1.69\text{E-}1}$ | $4.42\text{E-}1_{\pm1.34\text{E-}1}$ | $8.34\text{E-}1_{\pm1.04\text{E-}1}$ |
| | TimesNet | $5.22\text{E-}1_{\pm2.41\text{E-}1}$ | $5.79\text{E-}1_{\pm1.44\text{E-}1}$ | $8.45\text{E-}1_{\pm8.99\text{E-}2}$ | $1.92\text{E-}1_{\pm3.34\text{E-}2}$ | $3.45\text{E-}1_{\pm3.05\text{E-}2}$ | $9.41\text{E-}1_{\pm1.38\text{E-}1}$ |
| 4 Layers | Pyraformer | $6.71\text{E-}1_{\pm2.99\text{E-}1}$ | $6.72\text{E-}1_{\pm1.82\text{E-}1}$ | $8.60\text{E-}1_{\pm1.67\text{E-}1}$ | $2.11\text{E-}1_{\pm7.46\text{E-}2}$ | $3.56\text{E-}1_{\pm6.99\text{E-}2}$ | $8.83\text{E-}1_{\pm9.32\text{E-}2}$ |
| | SCINet | $5.72\text{E-}1_{\pm2.71\text{E-}1}$ | $5.88\text{E-}1_{\pm1.45\text{E-}1}$ | $9.32\text{E-}1_{\pm1.14\text{E-}1}$ | $3.14\text{E-}1_{\pm7.92\text{E-}2}$ | $4.33\text{E-}1_{\pm5.41\text{E-}2}$ | $9.92\text{E-}1_{\pm8.72\text{E-}2}$ |
| | TimeMixer | $5.20\text{E-}1_{\pm3.11\text{E-}1}$ | $5.81\text{E-}1_{\pm1.84\text{E-}1}$ | $8.66\text{E-}1_{\pm1.83\text{E-}1}$ | $2.98\text{E-}1_{\pm1.82\text{E-}1}$ | $4.13\text{E-}1_{\pm1.24\text{E-}1}$ | $8.48\text{E-}1_{\pm6.60\text{E-}2}$ |
| | TimesNet | $4.58\text{E-}1_{\pm2.01\text{E-}1}$ | $5.41\text{E-}1_{\pm1.29\text{E-}1}$ | $8.35\text{E-}1_{\pm1.46\text{E-}1}$ | $1.82\text{E-}1_{\pm3.51\text{E-}2}$ | $3.34\text{E-}1_{\pm3.32\text{E-}2}$ | $8.99\text{E-}1_{\pm9.63\text{E-}2}$ |
| 5 Layers | Pyraformer | $5.47\text{E-}1_{\pm2.70\text{E-}1}$ | $5.99\text{E-}1_{\pm1.71\text{E-}1}$ | $8.79\text{E-}1_{\pm1.30\text{E-}1}$ | $2.60\text{E-}1_{\pm2.20\text{E-}1}$ | $3.92\text{E-}1_{\pm1.57\text{E-}1}$ | $9.49\text{E-}1_{\pm9.39\text{E-}2}$ |
| | SCINet | $4.59\text{E-}1_{\pm1.64\text{E-}1}$ | $5.27\text{E-}1_{\pm9.21\text{E-}2}$ | $9.13\text{E-}1_{\pm7.36\text{E-}2}$ | $2.93\text{E-}1_{\pm1.40\text{E-}1}$ | $4.11\text{E-}1_{\pm8.92\text{E-}2}$ | $1.01\text{E+}0_{\pm1.27\text{E-}1}$ |
| | TimeMixer | $4.88\text{E-}1_{\pm2.25\text{E-}1}$ | $5.65\text{E-}1_{\pm1.38\text{E-}1}$ | $8.74\text{E-}1_{\pm1.19\text{E-}1}$ | $3.48\text{E-}1_{\pm1.57\text{E-}1}$ | $4.49\text{E-}1_{\pm1.03\text{E-}1}$ | $8.78\text{E-}1_{\pm1.01\text{E-}1}$ |
| | TimesNet | $5.22\text{E-}1_{\pm2.13\text{E-}1}$ | $5.81\text{E-}1_{\pm1.25\text{E-}1}$ | $8.02\text{E-}1_{\pm1.34\text{E-}1}$ | $1.79\text{E-}1_{\pm2.24\text{E-}2}$ | $3.34\text{E-}1_{\pm2.38\text{E-}2}$ | $9.33\text{E-}1_{\pm1.32\text{E-}1}$ |
| 6 Layers | Pyraformer | $6.87\text{E-}1_{\pm2.88\text{E-}1}$ | $6.79\text{E-}1_{\pm1.75\text{E-}1}$ | $8.63\text{E-}1_{\pm1.14\text{E-}1}$ | $2.39\text{E-}1_{\pm1.59\text{E-}1}$ | $3.80\text{E-}1_{\pm1.25\text{E-}1}$ | $9.57\text{E-}1_{\pm1.05\text{E-}1}$ |
| | SCINet | $5.63\text{E-}1_{\pm2.28\text{E-}1}$ | $5.83\text{E-}1_{\pm1.24\text{E-}1}$ | $9.33\text{E-}1_{\pm1.05\text{E-}1}$ | $2.77\text{E-}1_{\pm6.78\text{E-}2}$ | $4.06\text{E-}1_{\pm4.99\text{E-}2}$ | $9.73\text{E-}1_{\pm8.45\text{E-}2}$ |
| | TimeMixer | $4.56\text{E-}1_{\pm1.42\text{E-}1}$ | $5.50\text{E-}1_{\pm9.76\text{E-}2}$ | $8.36\text{E-}1_{\pm4.99\text{E-}2}$ | $3.29\text{E-}1_{\pm2.18\text{E-}1}$ | $4.39\text{E-}1_{\pm1.60\text{E-}1}$ | $9.66\text{E-}1_{\pm1.69\text{E-}1}$ |
| | TimesNet | $3.83\text{E-}1_{\pm1.29\text{E-}1}$ | $4.94\text{E-}1_{\pm7.87\text{E-}2}$ | $8.47\text{E-}1_{\pm9.32\text{E-}2}$ | $2.41\text{E-}1_{\pm1.14\text{E-}1}$ | $3.80\text{E-}1_{\pm9.01\text{E-}2}$ | $9.03\text{E-}1_{\pm1.43\text{E-}1}$ |
| 7 Layers | Pyraformer | $5.48\text{E-}1_{\pm2.30\text{E-}1}$ | $5.99\text{E-}1_{\pm1.45\text{E-}1}$ | $8.25\text{E-}1_{\pm1.43\text{E-}1}$ | $2.01\text{E-}1_{\pm3.47\text{E-}2}$ | $3.50\text{E-}1_{\pm2.71\text{E-}2}$ | $9.29\text{E-}1_{\pm9.34\text{E-}2}$ |
| | SCINet | $6.13\text{E-}1_{\pm2.10\text{E-}1}$ | $6.09\text{E-}1_{\pm1.19\text{E-}1}$ | $9.03\text{E-}1_{\pm5.44\text{E-}2}$ | $3.02\text{E-}1_{\pm1.12\text{E-}1}$ | $4.19\text{E-}1_{\pm7.42\text{E-}2}$ | $1.00\text{E+}0_{\pm1.30\text{E-}1}$ |
| | TimeMixer | $6.66\text{E-}1_{\pm3.45\text{E-}1}$ | $6.65\text{E-}1_{\pm2.03\text{E-}1}$ | $8.88\text{E-}1_{\pm1.69\text{E-}1}$ | $3.38\text{E-}1_{\pm1.49\text{E-}1}$ | $4.37\text{E-}1_{\pm9.83\text{E-}2}$ | $8.73\text{E-}1_{\pm1.15\text{E-}1}$ |
| | TimesNet | $5.71\text{E-}1_{\pm2.26\text{E-}1}$ | $6.07\text{E-}1_{\pm1.41\text{E-}1}$ | $8.32\text{E-}1_{\pm9.86\text{E-}2}$ | $1.98\text{E-}1_{\pm4.76\text{E-}2}$ | $3.47\text{E-}1_{\pm3.98\text{E-}2}$ | $9.17\text{E-}1_{\pm8.38\text{E-}2}$ |

# E   THEORETICAL ANALYSIS ON COPU'S STRUCTURE

In this section, we will establish the effectiveness of the COPU structure. Specifically, we will prove that Equation 9 implicitly represents the minimum variance estimator of COPU from $\tilde{x}$ to $\tilde{x}y$. For

$\tilde{x}y$ to restore $y$ given $\tilde{x}$ and compute the MSE loss, $\tilde{x}$ must be of full column rank, i.e., the RR must equal 1. This proof collaborate with our previous analysis of RR, once again emphasizing the indispensability and significant importance of RR.

We begin the proof by defining the concept of unbiased estimation. Unbiasedness allows us to obtain precise approximations of the true value through repeated experiments, which has substantial practical value. Suppose $y$ is the regression variable of interest, following a probability density function $p(y)$. Let $\tilde{y}$ be any estimate of $y$, and let $\mathrm{E}[\cdot]$ denote the expectation operator. Unbiasedness is defined as:

$$\mathrm{E}[\tilde{y}] = \mathrm{E}[y], \quad \int \tilde{y}p(\tilde{y})d\tilde{y} = \int yp(y)dy \tag{10}$$

Next, we will prove that the minimum variance estimator is unbiased. Suppose $x$ is an observable variable related to $y$, and we aim to estimate $y$ through $x$ using the minimum variance estimator $\hat{y}$, which can be defined as $\hat{y} = \mathrm{E}[y|x] = \int yp(y|x)dy$. Note that the estimator $\hat{y}$ is a function of $x$ because the influence of $y$ has been eliminated through integration. It is easy to prove that $\hat{y}$ is an unbiased estimator of $y$,

$$\mathrm{E}[\hat{y}] = \int \mathrm{E}[y]p(y|x)dy = \mathrm{E}[y] \tag{11}$$

We proceed to prove that the minimum variance estimator is the optimal unbiased estimator under the MSE criterion. This means that any variation based on $\hat{y}$ will only increase the MSE. This can be proven from a functional perspective. Define the MSE loss between $y$ and $\hat{y}$ as $\mathcal{F}$,

$$\mathcal{F} = \int \int (y - \hat{y}(x))^2 p(x,y)dxdy \tag{12}$$

Let $\delta$ be an infinitesimal perturbation. Note that $\delta\hat{y}(x)$ is a function entirely independent of $\hat{y}(x)$. Performing a functional analysis of $\mathcal{F}$, we obtain,

$$\begin{aligned} \delta F &= \int \int -2(y - \hat{y}(x))\delta\hat{y}(x)p(x,y)dxdy \\ &= -2 \int \delta\hat{y}(x)p(x) \int p(y|x)\left(y - \int yp(y|x)dy\right)dydx \\ &= 0 \end{aligned} \tag{13}$$

Therefore, we can conclude that $\hat{y}$ is an extremum with respect to the estimation of $y$. Since $\mathcal{F}$ is convex, this extremum must be the global minimum, thereby proving that the minimum variance estimator is the optimal estimator under the MSE criterion.

In our previous discussion, we introduce the properties of the minimum variance estimator. Next, we will demonstrate that COPU is the augmented minimum variance estimator from $\tilde{x}$ to $\tilde{x}y$. Note in the derivation of COPU, the concepts of samples and features are interchanged. Shaoqi et al. (2024) has already provided a detailed proof that the linear minimum variance estimator from $x$ to $y$ has the form $\mathrm{E}[y] + \mathrm{R}_{yx}\mathrm{R}_{xx}^{-1}(x - \mathrm{E}[x])$. Since the data are normalized, we assume for convenience that all variables have a mean of zero, where $\mathrm{R}_{yx} = \mathrm{E}[y^T x]$. Therefore, $\mathrm{R}_{yx}\mathrm{R}_{xx}^{-1}x$ simplifies to the linear minimum variance estimator from $x$ to $y$, which is also known the orthogonal projection.

Consequently, the minimum variance estimator from $\tilde{x}$ to $\tilde{x}y$ is $\tilde{x}^T(\tilde{x}^T\tilde{x})^{-1}\tilde{x}^T\tilde{x}y_{copu}$, noting that we have performed a transposition here. In this context, $R_{\tilde{x}\tilde{x}}^{-1}$ corresponds to $(\tilde{x}^T\tilde{x})^{-1}$, and $R_{\tilde{x}y}$ corresponds to $\tilde{x}^T\tilde{x}y_{copu}$. Here, $y_{copu}$ represents the output of the forward process of COPU, i.e., $\tilde{x}^T\theta_r$. Since $\tilde{x}R_{\tilde{x}\tilde{x}}R_{\tilde{x}y}$ represents the minimum variance estimator from $\tilde{x}$ to $\tilde{x}y$, COPU uses $R_{\tilde{x}\tilde{x}}R_{\tilde{x}y}$, i.e., $(\tilde{x}^T\tilde{x})^{-1}\tilde{x}^T m$, as the minimum variance estimator of $y$ to compute the loss in Equation 9. For this method to be effective, we must be able to restore $y$ from $\tilde{x}y$ given $\tilde{x}$; this requires that $\tilde{x}$ must be of full column rank, i.e., a larger RR.

