# OpenReview forum: "COPU: Recognizing Time Series' Heterogeneity In Stacked Neural Network"
_ICLR.cc/2025/Conference — Submitted to ICLR 2025_

### Official Review · Reviewer_1q18 · 2024-10-17

**Soundness:** 2
**Presentation:** 1
**Contribution:** 2
**Rating:** 6
**Confidence:** 3

**Summary:**

In this paper, the authors present an approach called Cross-correlation Enhanced Approximated Orthogonal Projection Unit (COPU) to quantify and augment a neural network’s (NN’s) nonlinear modeling capacity. The authors motivate their approach based on the observed inability of NNs to learn diverse features and effectively perform nonlinear modeling. The proposed method amplifies heterogeneous components and reduces homogeneous ones, enabling the learning of unique and independent features, thereby enhancing the NN's nonlinear modeling capability.

**Strengths:**

- The authors motivate the proposed approach using realistic examples.

- They propose using rank ratio as a metric for measuring nonlinear modeling capability.

- Experiments conducted on real world datasets.

**Weaknesses:**

- The presentation of the paper is quite confusing. There are several mistakes, such as Eq. (6) not being referenced anywhere, and in line 251, the notation "x \in \mathbb{R}^{d,b}" is unclear. Additionally, there are several sentences that are difficult to follow.
For example:

- "This process enables the differentiation between redundant information and innovation, further suppressing homogeneous information among features while amplifying their differences"
what innovation means here?

- "while the augmentation modular focuses on diverse and informative feature extraction"
module?

- "We implement this approach by  employing dictionary lookup Ashish et al. (2017) to achieve feature mapping."
wrong reference type.

- Using scientific notation for MAPE (which is typically presented as percentage) is also confusing.

- S4 acronym not introduced.

These are just a few examples, but there are numerous similar presentation issues throughout the paper, making it very difficult to follow. This is a significant shortcoming in itself, bringing the paper below ICLR  standards.

- Results on standardized and well-known datasets are expected to support the authors' claims and facilitate the reproduction of the presented results.

- In many cases, an MLP performs similarly to (or better than) the proposed approach, raising questions about the actual difficulty of the datasets used in the evaluation and whether the results truly support the authors' claims. The authors should also provide the results of a linear model to highlight the necessity of modeling non-linear components.

**Questions:**

Overall, the paper is not yet ready for publication. Significant improvements are required in both the presentation and the experimental setup and evaluation in order to adequately support authors' claims. While the authors are welcome to respond to these shortcomings, I believe the issues are too fundamental to be addressed through a rebuttal alone and will likely require a major revision of the paper.

---

> ### Author Response · Authors · 2024-11-13
> **Response #1 to Reviewer 1q18**
>
> **Response to Reviewer 1q18**
>
> We are grateful for the time and effort you have invested in reviewing our manuscript. We take each of your concerns seriously and are confident that we can address all issues raised to your satisfaction.
>
> **Response to Weaknesses 1**
>
> We respectfully point out that this could be a misunderstanding.
> Eq. (6) is implicitly referenced by Eq. (7), Eq. (9), as well as in the text from line 245 to line 249. The $\mathcal{L}$ in Eq. (9) specifics the loss computation with respect to the dual parameter $m$ in Eq. (7), the $\nabla_m$ specifics taking the derivative with respect to $m$, and the following textual explanation,
> > Specifically, NGD has been applied to the trackable parameter $\theta_r$ as shown in Figure 4.
>
> All of these jointly describe how Eq. (6) has been applied to COPU.
>
> The paragraph from line 251 to line 257 should be under Eq. (8). This is a mistake in formatting. We apologize for the inconvenience caused by our carelessness.
>
>
> **Response to Weaknesses 2**
>
> We respectfully point out that this could be a misunderstanding.
> Innovation is an important concept in the field of stochastic/statistical signal processing. It represents the information of newly observed data that cannot be expressed through orthogonal projections (Orthogonal Projection Unit, OPU) from previously seen data. Innovation has significant applications in areas such as Kalman filtering, control, and estimation. In our work, we similarly use innovation to denote the part of the information that is not easily represented by orthogonal projection from other samples. This filtering process is realized through the analysis of cross-correlation, which is why the model is termed COPU.
>
> **Response to Weaknesses 3**
>
> Thank you for pointing out this typo, the accurate form is "module".
>
> **Response to Weaknesses 4**
>
> Thank you for pointing out this issue. The citation should have been used in parenthesis. This is indeed a wrong reference type. Thank you for your meticulous examination.
>
>
> **Response to Weaknesses 5**
>
> Thank you for pointing out this issue. We will include additional clarifications regarding this metric.
>
>
> **Response to Weaknesses 6**
>
> Thank you for pointing out this issue. We apologize for the carelessness. S4 refers to the Structured State Space model proposed by Albert Gu in his paper "EFFICIENTLY MODELING LONG SEQUENCES WITH STRUCTURED STATE SPACES".
>
>
> **Response to Weaknesses 7**
>
> Thank you for your valuable suggestion. Reviewer bobw and Reviewer VW2o have also suggested expanding the scope of our experiments by incorporating additional datasets and comparative models to further validate the generality and superiority of COPU. We have initiated the preparations and hope to complete the additional experimental work within the deadline of the rebuttal period.
>
> **Response to Weaknesses 8**
>
> We respectfully point out that this could be a misunderstanding.
> The MLP's final layer is constructed as a temporal-wise fully connected neural network to enable temporal dependencies modeling, as described in line 410. This configuration is equivalent to the DLinear or NLinear approaches discussed in "Are Transformers Effective for Time Series Forecasting?", with the distinction that we do not perform additional series decomposition or normalization operations. Therefore, the experimental results observed by the reviewer corroborate the issues raised in that paper, which is also one of the motivations behind our study. Rather than naively applying architectures from CV/NLP or designing overly intricate modules, it is more productive to deepen our understanding of the modality. Such insight into RR is more practically significant and can guide future research in time series analysis.
>
> **Conclusion**
>
> We are sincere and candid about our work. We believe in our innovativeness and contributions and are willing to take critical suggestions. COPU is not flashy or avant-garde research; it addresses a very down-to-earth issue: Why do complex models often underperform in time series forecasting problems? Unlike CV/NLP, this field lacks a dominant network architecture akin to the Transformer. We have proposed the hypothesis that the time-series modality is less informative compared to CV/NLP, making models from those fields less adaptable. Subsequently, we introduced CEP and COPU to address this issue. Table I demonstrates that COPU's performance indeed significantly surpasses the compared SOTA and basic models. Table II shows that COPU-CEP remains robust to changes in batch size, further validating the necessity of CEP.
>
> The reviewer can refer to ["Response to Weaknesses 1 / VW2o"]() and ["Response to Weaknesses 1 / bobw"]() for further explanation of our motivation if interested. We hope the reviewer can consider raising COPU's rating. We believe that recognizing these modality differences is of significant importance and aspire to contribute this discovery to the community.

---

> > ### Comment · Reviewer_1q18 · 2024-11-14
> >
> > Thank you for your detailed responses and for providing these clarifications. After reading the comments of the other reviewers and the authors' response, I decided that slightly raise my rating. The issue of lacking standarized evaluation still remains though, which is a critical limitation, as also highlighted by other reviewers.

---

> > > ### Author Response · Authors · 2024-11-14
> > > **Response #2 to Reviewer 1q18**
> > >
> > > Thank you sincerely for raising your rating of COPU. Your recognition is important to us. We are currently expediting our experiments; however, since we need to perform RR analysis on datasets such as ETT (as suggested by reviewer VW2o), this requires some more time. Additionally, we have found and noted that existing studies on datasets like ETT are conducting prediction tasks, i.e., using past labels to predict future labels. In contrast, our paper focuses on a regression task, utilizing related process data to regress the variable of interest, like the variable difficult to measure directly during actual production, such as the targeted variables (sulfur and butane content) in the SRU and Debutanizer datasets. This discrepancy may make it challenging to directly compare COPU with previous works.
> > >
> > > In summary, we are making every effort to expand our experiment by introducing more datasets to ensure the effectiveness and superiority of COPU. We are confident that we can provide additional experimental results before the deadline. Once again, thank you for your appreciation of our work.

---

> > > > ### Author Response · Authors · 2024-11-25
> > > > **Response #4 to Reviewer 1q18**
> > > >
> > > > Dear Reviewer 1q18,
> > > >
> > > > As the rebuttal-discussion period is coming to a close in approximately one day, we would like to know whether you have any remaining concerns regarding COPU.
> > > >
> > > > We would also like to recommend, in order, several responses that we believe precisely summarize the contributions of COPU. We hope these will enhance your appreciation of our work:
> > > >
> > > > 1. **Four reasons for raising the rating of COPU (Comment #2 on the second revision)**
> > > > 2. **Response #1 to three important issues**
> > > > 3. **Comment #1 on the second revision**
> > > > 4. **Response #8 to Reviewer VW2o**
> > > > 5. **Response #7 to Reviewer VW2o**
> > > >
> > > > Furthermore, we have expanded our experiments by introducing additional datasets and methods to ensure the effectiveness and superiority of COPU, as we have shown in **Comment (#1 & #2) on the (first & second) revision**.
> > > >
> > > > We hope that this revision will merit an increase in your rating. If you have any additional concerns, we will be more than happy to address them promptly.

---

> > > > > ### Comment · Reviewer_1q18 · 2024-11-26
> > > > >
> > > > > Thank you, after considering all of your responses I am further raising my score to acknolwedge the concerns that have been indeed addressed.

---

> > > > > > ### Author Response · Authors · 2024-11-26
> > > > > > **Response #5 to Reviewer 1q18**
> > > > > >
> > > > > > Dear Reviewer 1q18,
> > > > > >
> > > > > > Thank you so much for your recognition.  We are truly overwhelmed with excitement.  Your support really means a lot to us!  Thank you for your time. If you believe there is further room for improvement with COPU, we will promptly begin optimizing it.
> > > > > >
> > > > > > Sincerely,
> > > > > > All authors

---

> > > ### Author Response · Authors · 2024-11-20
> > > **Response #3 to Reviewer 1q18**
> > >
> > > Dear Reviewer 1q18,
> > >
> > > We are pleased to inform you that we have completed the revisions to our manuscript. We would like to extend our sincere gratitude for your patience and support. If you have any questions, we will be happy to address them promptly.
> > >
> > > After completing the additional experiments, we immediately set about preparing a **global response** in which we will elucidate three main issues:
> > >
> > > 1. Why do we start focusing on studying RR instead of other metrics.
> > > 2. Why the loss function is formulated as shown in Equation 8, and gradient is calculated as shown in Fig. 4.
> > > 3. The relationships among CEP, RR, and COPU.
> > >
> > > We believe that this explanation will resolve any confusion. We will release this response within two days. Hoping you will like it.
> > >
> > > Sincerely,
> > >
> > > All authors

---

### Official Review · Reviewer_VW2o · 2024-10-21

**Soundness:** 2
**Presentation:** 2
**Contribution:** 2
**Rating:** 3
**Confidence:** 5

**Summary:**

This paper think that conventional NN architectures overemphasize homogeneous feature extraction, impeding the learning of diverse features and diminishing their nonlinear modeling capability. To address this gap, they propose the Cross-correlation Enhanced Approximated Orthogonal Projection Unit (COPU) to quantify and augment the NN’s nonlinear modeling capacity.

**Strengths:**

They introduced RR as a metric to quantify the proportion of linear dependencies transformed into independence by the network. Performing a comprehensive exploration of NN’s nonlinear modeling capability.  This paper proposes a new NN component termed CEP that has strong nonlinear representation performance in time series analysis and serves as the foundation for developing the COPU framework.

**Weaknesses:**

1.  In page 1, the author said :  For these methods to be effectively applied in this field, it is
essential to recognize that time series has more ambiguous discriminative patterns than other forms
like images and text (Alec et al., 2021). Such ambiguity hinders the model’s ability to extract diverse
features from the input, obstructing its capacity for nonlinear modeling.

Then you give simple explanation, but the problem is you explanation is wrong. You classify original text and image data, while you do convoltuion on timeseries dat, and then you said "it is more challenging to discern the effect of two input sequences on the output
of a system". That is not resonable.


2. Some sentence is hard to understand like:
Page 7: This offers a novel perspective that elucidates the underlying efficacy of residual connections in NNs beyond the kernel (Duvenaud et al., 2014) and gradient (Kaiming et al., 2016) explanations.

3. In Figure 7, except for layer 7, it is hard for me to find difference CEP and other method, and I think with the iteration, the CEP curve will  close to other curve in the end.

4. You need to add explanation to different lines in Figure 8.

5. Last and most important Problem:
You propose a new method to solve the problem in time series prediction, but you did not conduct experiments on popular dataset and compare with some current SOTA models, like you refered in your paper : Are transformers effective for time series forecasting?

**Questions:**

The same to Weaknesses

---

> ### Author Response · Authors · 2024-11-13
> **Response #1 to Reviewer VW2o**
>
> **Response to Reviewer VW2o**
>
> We thank the reviewer for taking the time to assess our paper and giving the valuable feedback. We will meticulously address the reviewer's concerns in the following. Thank you for your comprehensive examination and critical suggestions.
>
> **Response to Weaknesses 1**
>
> We understand the reviewer's confusion. We have endeavored to use concise language to describe the phenomena we have observed.
>
> The term **"ambiguous discriminative patterns"** specifically refers to Figure 1, where two sequences that overlap by nearly two-thirds yield vastly different outputs after a simple convolution, while two entirely unrelated sequences produce similar outputs. Therefore, in the context of time-series analysis, it becomes challenging to discern the influence of input on the system's output. This contrasts with other data modalities. Even a simple convolution system exhibits such behavior in time-series data, let alone complex nonlinear dynamic systems. This is why we stated at the end of the first paragraph:
>
> >“Thus, from the perspective of input, the discriminative patterns among different time series are not only difficult to express mathematically but also inherently ambiguous. Figure 1 vividly illustrates this process using a simple kernel convolution.”
>
> **Time-series data, especially in industrial settings, are less informative compared to other modalities.** Images and sentences (through embedding techniques) can easily have variable dimensions reaching thousands or even tens of thousands, not to mention their rich semantic information. In contrast, time-series data are often recorded by sensors, with variable dimensions typically ranging from tens to hundreds (less than one hundred in industrial contexts). These sensors may even measure the same object at different time delays. Consequently, when the batch size is too large, the number of effective samples (i.e., samples that cannot be linearly represented by other samples) within a mini-batch may be smaller than the batch size. This characteristic is not present in other modalities such as computer vision and natural language processing.
>
> **Modeling time-series data is non-intuitive.** With images and sentences, we can intuitively accomplish downstream tasks through their semantic information, such as image classification, object detection, named entity recognition, machine translation, and so on—even though it is difficult to mathematically represent this process—hence the introduction of neural networks as modeling tools. In contrast, performing tasks like system identification, fault detection, and regression analysis through time-series data is extremely non-intuitive, but mathematical modeling is possible under specific structural assumptions (e.g., subspace identification, Gaussian process regression).
>
> Therefore, when the time-series modeling object is too complex to be represented using traditional mathematical tools, trivially applying networks designed for CV and NLP to extract features may be inefficient. This is because these architectures may struggle to effectively construct nonlinear relationships on less informative data, as shown in Figure 2. We should deepen our understanding of the time-series domain, recognizing that its data are less informative, as depicted in Figure 5. We should not let the scarce and valuable heterogeneous information be overwhelmed by abundant homogeneous information in stacked neural networks, i.e., to monitor and enhance RR, as illustrated in Figures 6 and 7.
>
> **Response to Weaknesses 2**
>
> The first referenced paper studies how repeated nesting of multiple kernel functions can lead to degradation in nonlinear modeling capabilities, ultimately causing the kernel to lose its nonlinear modeling ability. Introducing residual connections within the nested kernel pathways can effectively alleviate this phenomenon, enhancing nonlinear modeling capabilities. The second referenced paper indicates that, in the absence of residual connections, gradients tend to diminish during backpropagation, such that the shallowest layers of the network receive almost no updates, resulting in poor performance in deep networks.
>
> Figure 6 shows that as the number of layers increases, the RR of MLP decreases surprisingly rapidly, indicating that the effective information output by MLP diminishes (i.e., it can be linearly represented by other samples). Conversely, the RR of RES, which incorporates residual connections, remains robust, suggesting that the effective information output by RES hardly decreases with increasing depth. **This phenomenon provides an additional perspective on why residual connections are widely adopted.**
>
> In our effort to include as much information as possible within the limited page constraints, we may have sacrificed some readability. We apologize for any inconvenience this may have caused.

---

> > ### Author Response · Authors · 2024-11-13
> > **Response #2 to Reviewer VW2o**
> >
> > **Response to Weaknesses 3**
> >
> > We also find this phenomenon intriguing. Traditionally, one would expect that during the training process, networks continuously learn conducive features and construct nonlinear relationships, causing fluctuations in RR, i.e., RR might decrease or increase as the network converges to a few important features or learns numerous nonlinear characteristics. However, intriguingly, in all models except COPU, RR almost remains constant. We specifically conjectured about this phenomenon in our manuscript:
> >
> > > “Examining Figures 7 and 8 together uncovers intriguing characteristics. Notably, the interval during which the model's validation loss decreases rapidly closely coincides with the sharp rise in RR. The initial 20 iterations mark the rapid convergence phase for each model, during which most models exhibit a significant upward trend in RR. The observations suggest that in the early stages of training, NNs actively search across the wide parameter space, eagerly exploring various feature representation methods relevant to the task at hand. As training progresses and begins to stabilize, the RR settles down; however, the validation loss continues its steady decline. This indicates that the NNs are now fine-tuning the intricate mapping from latent variables to outputs, building upon the feature representation methods they previously discovered. Figure 7 and 8 imply that the window for effective learning, especially the acquisition of feature representation, is remarkably brief. Consequently, the ability of nonlinear models to extract diverse and rich information from inputs becomes critical.”
> >
> > In summary, we find this phenomenon novel and fascinating, and we believe that the community will also be eager to explore questions such as, **"Why does the network training process not align with our previous assumptions? How can we control the training process to improve performance?"**
> >
> > **Response to Weaknesses 4**
> >
> > Thank you for your reminder; we find there lacks a detailed explanation of the lines in Figure 8. Figure 8 serves as a direct evaluation of the performance of each base module in end-to-end experiments. The curves representing the different base modules are indicated in the legend of the figure. The output dimension of the network's final layer is 1.
> >
> > **Response to Weaknesses 5**
> >
> > Thank you for your suggestion. Since the author's team focuses on AI applications in industry, we habitually choose experiments from that field. We have conducted extensive experiments on hyperparameters, ablation studies, and significance properties, and we initially believe that the existing experiments are quite sufficient. We will strive to incorporate more datasets to provide a more comprehensive experimental analysis for COPU.
> >
> > We sincerely hope that our response has addressed the reviewer's concerns and cleared up any misunderstandings. We would greatly appreciate it if the reviewer could consider raising COPU's rating. If the reviewer has any further concerns, we are more than happy to help address them.

---

> > ### Comment · Reviewer_VW2o · 2024-11-18
> >
> > Weaknesses 2 & 5
> >
> > I am not a harsh reviewer, but this paper does indeed have significant issues. What I want to point out is that the poor performance of MLPs is already well-known. If you aim to theoretically study MLPs and ResNets, you should provide explanations based on specific theoretical frameworks rather than relying on observed phenomena, as there are too many phenomena to enumerate exhaustively.
> >
> > If this paper is about network design for addressing temporal modeling, you should demonstrate that your results are either SOTA or at least close to SOTA. However, the methods you compare against are outdated and perform significantly worse than the current SOTA.

---

> > > ### Author Response · Authors · 2024-11-18
> > > **Response #3 to Reviewer VW2o**
> > >
> > > We sincerely appreciate the reviewer's willingness to engage in a constructive discussion with us. We respectfully suggest that there may have been a misunderstanding. **In the following, we will provide a clear and detailed explanation of our research motivation, primary focus, and how our work differs from others in the field.** We hope that our explanation will dispel any misunderstandings.
> > >
> > > # Regarding the content
> > >
> > > Contrary to common perceptions, the performance of MLP-based models (e.g., DLinear, RLinear, NLinear) in time series is actually not poor. These models are frequently referenced and compared among SOTA papers, and their performance is often second only to the best. In contrast, the performance of MLP-based models in the CV/NLP domains is widely acknowledged to be poor. This contradictory phenomenon, where MLP networks outperform complex networks in the time series domain, has piqued our interest. **Rather than being content with simply proposing a solution (COPU) that surpasses MLPs or other complex SOTA, we aim to delve deeply into the underlying reasons.** To this end, we introduce RR as a tool to explain this phenomenon.
> > >
> > >
> > > **RR is not an arbitrarily listed indicator; it is originally employed as a metric to measure numerical errors and the degree of approximation, possessing a robust theoretical foundation.** In [AOPU](https://openreview.net/forum?id=xqrlhsbcwN&noteId=qZUAzeQEBS), RR is defined as the degree to which the objective function can restore the MSE loss (see Eq. 6 in AOPU), as it directly determines the precision loss in matrix inversion during the computation (inversion via SVD decomposition). The network structure of AOPU, featuring dual parameters and truncated gradients, represents an innovative breakthrough with strong theoretical support. Specifically, as RR approaches 1, the truncated gradient converges toward the natural gradient of the trackable parameter (i.e., manifold-aware gradient), and the network's output approaches the minimum variance estimation, i.e., the theoretical ceiling of an unbiased estimator. Building upon our understanding of the time series domain, we have expanded the original concept of RR—from a measure of approximation degree—to the concept proposed in this paper: information density. Given the outstanding achievements of AOPU, we also developed COPU based on this framework.
> > >
> > >
> > > In this context, variations in RR to batch size and sequence length can reflect the intrinsic information density of the data itself, while variations in RR to stacked layers can reflect the network's nonlinear modeling capabilities. Our experience in the industrial sector informs us that data collected by sensors often contains a substantial amount of repetitive and redundant information, which reduces its information density. We have conducted extensive experiments (including additional forthcoming experiments), and the results indicate that RR is significantly less than 1 across various application scenarios in the time series domain, such as chemical engineering, power systems, and weather forecasting. **This phenomenon reveals the significant difference between time series and CV/NLP modalities, i.e., that time series are less informative.** We believe it is this characteristic of time series that makes the straightforward application of complex models designed for CV/NLP modalities inappropriate. These models, such as Attention mechanisms, are intended to identify features of interest from massive and diverse information; however, in the context of time series, they might intend to extract multiple copies of a certain pattern from vast amounts of homogeneous information, lacking the ability for regularization—that is, the capacity to recognize multiple different patterns.
> > >
> > > This phenomenon has actually been implicitly recognized within the field of industrial system identification. For example, when applying ordinary least squares (OLS), the requirement that the number of samples significantly exceeds the feature dimension arises precisely because of the difficulty in ensuring that $xx^T$ is invertible. Similarly, partial least squares (PLS) is proposed as an identification tool to extract heterogeneous information within samples while suppressing homogeneous ones.
> > >
> > > Our intention is not to conduct a theoretical study of MLPs or ResNets. The statement, "This offers a novel perspective that elucidates the underlying efficacy of residual connections in NNs beyond the kernel (Duvenaud et al., 2014) and gradient (Kaiming et al., 2016) explanations," is merely an ancillary proposition stemming from our analysis of RR, an interesting way we found to explain the residual connection.
> > >
> > > RR represents a fresh perspective that enables the observation of numerous interesting and engaging phenomena. **We believe that this approach offers a more significant contribution than exclusive model-wise research.**

---

> > > > ### Author Response · Authors · 2024-11-18
> > > > **Response #4 to Reviewer VW2o**
> > > >
> > > > # Regarding the experiments
> > > >
> > > > To validate the phenomena observed based on RR, we proposed CEP and COPU, conducting extensive repeated experiments and hyperparameter sensitivity analyses to ensure the reliability of our results. The experiments demonstrate that an understanding of RR enables COPU to achieve superior performance and robustness. **While all model-wise innovative papers claim the best performance, COPU not only achieves this but also explains the underlying reasons, even guiding directions for improvement in other methods. Such accomplishments would not have been possible without a deep understanding of the time series domain.**
> > > >
> > > > Regarding the selection of comparative models in our experiments, this has been carefully thought through and screened. As is customary in time series research, we compare against MLP-based methods. We flatten the final layer of the MLP along the temporal dimension into a fully connected layer, essentially implementing methods like DLinear, except without additional series decomposition or normalization. We include LSTM because it is the most widely recognized benchmark. We incorporate S4 as it represents the SOTA in RNN. We include Autoformer because it is representative of attention-based SOTA methods and likewise accounts for the sequential correlation characteristics of time series data; the difference is that it utilizes autocorrelation features, whereas our paper leverages cross-correlation features. We also include Informer because it is a widely used and recognized SOTA model. We believe that these comparative models sufficiently support our claim that more complex network structures are not necessarily superior. However, we acknowledge that we should conduct experiments and validations on a broader range of datasets to reinforce our conclusions. We appreciate the reviewer's suggestion in this regard and are confident that we can submit the revised manuscript within four days.
> > > >
> > > >
> > > > The mainstream in time series NN is predominantly model-wise and method-wise, whereas our study on COPU adopts a modality-wise perspective. This shift signifies that our conjectures regarding RR and deep understanding of the time series domain are the prominent contributions of this paper. COPU is not flashy or avant-garde research; it addresses a very down-to-earth issue: Why do complex models often underperform in time series forecasting problems? **While we, like other model-wise studies, are confident in COPU, it is the understanding of RR that has the potential to inspire all researchers in time series NN. We believe that this broad applicability and impact make it major contribution of our work.**
> > > >
> > > > We sincerely appreciate the reviewer's patience in thoroughly reading our response. We hope the reviewer can consider raising COPU's rating. We believe that recognizing these modality differences is of significant importance and aspire to contribute this discovery to the community.

---

> > > ### Author Response · Authors · 2024-11-20
> > > **Response #5 to Reviewer VW2o**
> > >
> > > Dear Reviewer VW2o,
> > >
> > > We are pleased to inform you that we have completed the revisions to our manuscript. You can access the latest changes through our comprehensive response and by downloading the updated PDF. We believe that these revisions, along with our prior discussions, have fully clarified any misunderstandings.
> > >
> > > We kindly invite you to re-examine our paper. **We want to do more than merely propose a model; we aim to make substantial progress in this field** by deepening our understanding of time series data. RR is not introduced arbitrarily; it is central to both AOPU and COPU, directly reflecting the numerical stability of the model. It also encapsulates the model's gradient perception of the manifold and represents an approximation to the minimum variance estimation. Moreover, it serves as a metric for information density. These attributes collectively necessitate a thorough analysis of RR.
> > >
> > > If the reviewer has any additional concerns or questions, please feel free to contact us. We would be more than happy to help address them.
> > >
> > > Sincerely,
> > >
> > > All authors

---

> ### Comment · Reviewer_VW2o · 2024-11-21
>
> Neither LSTM nor Autoformer represents the SOTA methods, and although Linear has been open-sourced, you did not use the official code. Moreover, many current models perform better than Linear. I suggest that the author avoid taking shortcuts and conduct the necessary experiments, making proper comparisons where required. The premise of introducing new theories should be their validity or the completeness of proof, meaning thorough mathematical substantiation. Clearly, you have not achieved either. The author aims to prove the theory experimentally but refuses to compare it with SOTA methods on public datasets, a practice I find unconvincing.

---

> > ### Author Response · Authors · 2024-11-21
> > **Response #6 to Reviewer VW2o**
> >
> > We have some clarifications to make.
> >
> > - We never said LSTM is SOTA. We include it because it is widely recognized. We need to compare with at least one well-accepted and sound method to prove that the improvement is solid, **as we described above**.
> >
> > - Autoformer remains a SOTA, as it is widely used as a benchmark in cutting-edge research. So as Informer. Besides, the primary reason we chose Autoformer is that it likewise accounts for the sequential correlation characteristics of time series data; the difference is that it utilizes autocorrelation features, whereas our paper leverages cross-correlation features, **as we described above**.
> >
> > - Because we want to prove that base modules like MLP are not well-suited for time series, so we stacked MLP and performed Linear at the final layer, **as we described in the manuscript**.
> >
> > - Though many current models claim they perform better than Linear, in almost all these studies, Linear is only second to the best, **as we described above, and as we observed in the manuscript**.
> >
> > We respect the reviewer's statement.
> > We will try to include 2-3 SOTA methods in our comparisons. We will also include mathematical proof of the validity of COPU’s structure. The characteristic that RR represents the sample information density is too intuitive to require proof. The fact that changes in RR with respect to the number of layers reflect the model's nonlinear modeling capability cannot be theoretically proven; therefore, we have conducted a detailed analysis of the network's internal structure in this paper. The above revisions may take us **3 days** to complete. If you still have concerns about the adjustments we are about to make, please contact us promptly so that we can make timely modifications.
> >
> > We have shown the utmost respect to the reviewer, and we expect the reviewer to respect us as well. We sincerely hope the reviewer can check our responses and revisions carefully before making judgments.

---

> > ### Author Response · Authors · 2024-11-22
> > **Response #7 to Reviewer VW2o**
> >
> > To Reviewer VW2o,
> >
> > We have completed the proof of the effectiveness of the COPU structure in our revised manuscript. Specifically, we have demonstrated that the objective function of COPU utilizes the orthogonal projection from $\tilde{x}$ to $\tilde{x}y$, thereby validating Equation 9, with its computational accuracy measured by RR. Regarding SOTA comparisons, we have included additional experiments with SCINet, Pyraformer, TimesNet, and TimeMixer on the SRU, Debutanizer, ETTh2, and ETTm2 datasets. Although the experiments are not yet complete, we are willing to update the theoretical proof section first and would appreciate your feedback.
> >
> > For the detailed proof process, you may refer to our global response ** Comment #1 on the second revision** or directly review the revision.
> >
> > **We wish to emphasize that the innovation in the field of AI for time series is not confined to a single way**. Proposing sophisticated models to improve benchmark scores is just one way. Our work uncovers a previously unnoticed phenomenon: time series data are less informative. This finding may directly explain why simple linear models perform so well—perhaps there aren't many complex features can be extracted using attention mechanisms. This discovery further encourages research into the Math For AI. In a domain with lower data information density, we need more structural priors, such as COPU.
> >
> > **We are hesitant to conduct experiments on these overly complex SOTA models because**: 1) Based on our research findings, we do not have confidence in these novel, cutting-edge, and complex models. We believe in the achievements within the realm of "Math For AI" in time series, such as S4, or those results that have stood the test of time. 2) Our research focuses on regression tasks rather than prediction tasks, as evidenced by our emphasis on orthogonal projection. We selected Autoformer and Informer to represent transformer-based models, choosing ones with robust mathematical foundations and similarities to our work. We have selected two representative models for each base module.
> >
> > Our research offers a novel perspective, is methodologically sound, contributes significantly, introduces substantial innovation, and achieves excellent results. Once again, we respectfully request that the reviewer consider raising the rating for COPU, as a rating of 3 severely underestimates our work.
> >
> > **We must admit that your lack of recognition has left us feeling anxious and exhausted**. Yet we were not persuaded by the reasons you have provided, as we often found ourselves clarifying many of your misunderstandings. If the reviewer insists on rejecting our work, we would appreciate substantial reasons. We will continue to improve our manuscript to make it better, but we will not give up on pursuing publication.

---

> > ### Author Response · Authors · 2024-11-24
> > **Response #8 to Reviewer VW2o**
> >
> > Dear Reviewer VW2o,
> >
> > We think we might understand why our work hasn't resonated with you; we think this stems from a difference in our philosophies.
> >
> > **We surmise that you may view AI research as a competition, where the highest-scoring results get published. In contrast, we consider AI research to be a science that invites exploration and explaination.** As we have repeatedly emphasized, our goal is more than merely proposing a model; we aim to uncover the fundamental reasons why models adopted in CV/NLP cannot be trivially applied to time series data.
> >
> > In our view, the current advancements of AI in the field of time series may be exhibiting a few signs of stagnation. On one hand, linear methods are outperforming the vast majority of complex structural designs. On the other hand, models are unable to eliminate their dependence on past labels and perform poorly on challenging regression problems, as we will demonstrate in our revised manuscript by presenting the regression results of SOTAs on the Debutanizer and SRU. That is why we believe research that delves beneath the surface to examine underlying principles is crucial, and this is a significant contribution that distinguishes COPU from other studies.
> >
> > Our experimental setup is reasonable because our research focuses on different base modules. The manuscript introduces three base modules: MLP, RNN, and Attention. Accordingly, we selected six models: MLP, RES, LSTM, S4, Informer, and Autoformer, with each base module represented by two models. To enable the MLP to model temporal dependencies while showcasing its feature extraction capabilities, we added a Linear layer at its final stage; the same approach was applied to RES. The results in Table I demonstrate that COPU surpasses these comparative frameworks. Could it be that COPU is strong but CEP is ineffective? The ablation study in Table II indicates otherwise. When the batch size increases, only CEP exhibits excellent stability for RR, while all other base modules fail. These achievements are all based on our understanding of RR.
> >
> > **Our beliefs may differ, but such differences precisely drive the advancement of disciplines. As CEP suggests, should not let the tide of homogenization drown out individual uniqueness**. Let AI be diverse and full of vitality.
> >
> > We would also like to extend our apologies to you for our earlier words and expressions; we were overly nervous and anxious. Regardless, we respect your perspective and will still believe in ourselves.

---

> > > ### Comment · Reviewer_VW2o · 2024-11-26
> > >
> > > I don't understand why it is so difficult to simply provide a direct comparison of the final results. Could you please present the comparison results in the same format as Table 3 in this paper: https://arxiv.org/pdf/2211.14730.pdf? Almost all current time series prediction papers follow this approach, and the experiment is quite straightforward, taking no more than 2 hours to complete. If you find my request unreasonable, then I don't think it's fair for you to ask me to improve your scores. I believe this is also unreasonable.

---

> > > > ### Author Response · Authors · 2024-11-26
> > > > **Response #9 to Reviewer VW2o**
> > > >
> > > > First, we would like to thank the reviewer for being responsive.
> > > >
> > > > Then we want to emphasize **four** important points,
> > > >
> > > > 1. We have to run experiments 10-20 times to present **mean** and **std**. For each method, each dataset, and each configuration (layers).
> > > >
> > > > 2. We are focusing on regression task, so no prediction length or label length settings. As we have emphasized many times before.
> > > >
> > > > 3. We have already presented the results in the revised manuscript. Reviewer can go to our **Comment (#1 & #2) on the (first & second) revision** to check the update.
> > > >
> > > > 4. We are very reasonable, as we provided detailed explanations for nearly everything we did. We have followed the reviewer’s suggestions to provide both additional theoretical and experimental results, for that we have put a great deal of time. We think it is very reasonable for us to require a better rating.
> > > >
> > > > We sincerely hope the reviewer can go through our responses or manuscript.

---

> > > > ### Author Response · Authors · 2024-11-26
> > > > **Response #10 to Reviewer VW2o**
> > > >
> > > > Our **Response #8 to Reviewer VW2o** is we want to discuss further with you ***after*** we have completed your request!
> > > >
> > > > To enhance the efficiency, we will stop the red tape.
> > > >
> > > > Following your request, we have made ***Comment (#1 & #2) on the (first & second) revision*** in the global response, where we have added additional experimental results and theoretical proof.
> > > >
> > > > After that, we want to discuss further with you, to persuade you that our study is valuable in a different way, that is why we write **Response #8 to Reviewer VW2o**.
> > > >
> > > > Before that, we tried to discuss with you through our **Response #7 to Reviewer VW2o**, and you did not respond.
> > > >
> > > > We are very reasonable. We replied to you very promptly. We hope that can start a real discussion.

---

> > > > ### Author Response · Authors · 2024-11-26
> > > > **(Response #11 to Reviewer VW2o)**
> > > >
> > > > We want you to know we are waiting for your responses.
> > > >
> > > > We also hope that at least you are now aware of the following two very basic things,
> > > >
> > > > 1.  We have completed what you required (additional experimental results and theoretical proof).
> > > >
> > > > 2.  We focus on regression task.
> > > >
> > > > To be honest, we are almost driven crazy because you simply don't check our response. This is very irresponsible as a reviewer. We feel so disrespected.
> > > >
> > > > We think you might doing this deliberately.
> > > >
> > > > We hope you can answer and clarify this to us.

---

> > > > > ### Comment · Reviewer_VW2o · 2024-11-27
> > > > >
> > > > > You claim that you proposed a new theory, but you assert its strength and usefulness without verifying it through actual experimental tasks.
> > > > >
> > > > > Do you know KAN? His theory is more rigorous, and his mathematical proofs are more complete than yours. Why was KAN's work rejected? You propose a so-called theory without providing any practical validation or solid mathematical derivation. How can reviewers judge whether your theory is useful?
> > > > >
> > > > > I am driven crazy, too. You should look at how many reviewers have commented. Do you think I'm deliberately picking on you? If that were the case, I wouldn't have responded to you at all.
> > > > >
> > > > > I cannot see the validity or the usefulness of the so called theory you claim in your paper.

---

> > > > > > ### Comment · Reviewer_VW2o · 2024-11-28
> > > > > >
> > > > > > The review process is nearly complete, and if I have caused any discomfort or inconvenience, I sincerely apologize. Please know that my intentions were not directed at you personally.
> > > > > >
> > > > > > Despite my own experience of having my manuscript unfairly rated low and receiving no response from the reviewers, I still managed to provide you with timely feedback. I believe this demonstrates that I am a responsible and qualified reviewer.
> > > > > >
> > > > > > Moreover, I am not a competitor in your field, so there was no motive for me to give a low score maliciously. I think my review comments were intended to be clear, and I hope they will be helpful to you.
> > > > > >
> > > > > > Thank you for your understanding.

---

> > > > > > > ### Author Response · Authors · 2024-11-28
> > > > > > > **Response #12 to Reviewer VW2o**
> > > > > > >
> > > > > > > Thank you for your response.
> > > > > > >
> > > > > > >
> > > > > > > Please do not take offense; we do not know how to express this in a better way. However, it is difficult for us to maintain confidence in you because your responses have been brief and vague, and they repeatedly contain conflicts and misunderstandings, which we have clarified before. We understand that misunderstandings can occur, and we are willing to resolve them. We also welcome critical suggestions. If reviewers respect us and carefully examine our manuscript and responses, we will also accord them the highest respect, regardless of their decision or rating. When you decided to lower the rating, we respected you the same way we did to other reviewers. At that time, we supplemented experimental results and theoretical proofs in the global response while also clarifying misunderstandings in our private response to you. However, we will not accept malicious rejection, nor will we fear confronting any injustice. If you have ever experienced a similar sense of unfairness, you would understand how we feel.
> > > > > > >
> > > > > > > Thank you for your understanding too.

---

### Official Review · Reviewer_bobw · 2024-11-03

**Soundness:** 3
**Presentation:** 3
**Contribution:** 2
**Rating:** 6
**Confidence:** 3

**Summary:**

This paper proposes a new neural network component, called the cross-correlation Enhanced Perceptron (CEP), to solve the difficult problem of nonlinear modeling in time series analysis. CEP facilitates the learning of unique features by performing alignment and cross-correlation calculations on input features in a single step to distinguish and amplify different features while suppressing the influence of similar features. To further improve the model performance, the authors integrate CEP into the approximate orthogonal projection unit (AOPU) to form the COPU framework. COPU uses CEP to enhance nonlinear modeling capabilities and solves the limitations of AOPU in computational accuracy and expressiveness. The experimental results show that COPU is significantly superior to the existing methods in several real regression tasks. The contribution of this paper is to develop a CEP component specially used in time series analysis, which provides a new idea for feature modeling in a specific domain. At the same time, the authors also focus on an evaluation metric Rank Ratio (RR), because it can be interpreted as the proportion of linear dependencies that are translated into independence by neural networks.

**Strengths:**

1. The author clearly demonstrates the problem that this paper aims to solve: that complex networks can effectively solve problems related to computer vision and natural language processing; But its ability to solve the problem of time series analysis is insufficient.
2. The author proposes a CEP scheme, which quantifies the similarity between features through Cross-correlation to ensure alignment without directly ignoring differences in time, which is quite innovative.
3. The author summarized relevant literature, explained the characteristics of time series analysis problems, and proposed that RR index has a good evaluation significance in measuring time series analysis problems, and this index can be relatively convincing in the subsequent research
4. The author cited sufficient literature to demonstrate the point of view, and designed ablation experiments to enhance feasibility.

**Weaknesses:**

1. The advantages of indicator RR compared with other common indicators are not fully elaborated.
2. The relationship between CEP and COPU is not clearly stated, and it should be explained in detail how to achieve it.
3. The datasets used in the experimental design are insufficient, and more datasets are recommended to be added for verification.
4. In terms of experimental Settings, the author should provide setting parameters of different networks to compare the results more reasonably.

**Questions:**

1. In the design process of CEP, why may the c value be negative? Why is it feasible to use sigmoid to force a negative number to be positive?
2. Does the CEP module have to be fixed with your network structure or can it be embedded in other models as well?
3. Will the paper provide code? I want to know about the implementation of CEP and COPU.

---

> ### Author Response · Authors · 2024-11-13
> **Response #1 to Reviewer bobw**
>
> **Response to Reviewer mbUm**
>
> We thank the reviewer for taking the time to assess our paper and giving the positive feedback. Your recognition is a great encouragement to us. We have addressed each concern raised by the reviewer. Thank you for your careful review and valuable suggestions.
>
> **Response to Weaknesses 1**
>
> The structured design of COPU necessitates our study of RR, which distinguishes it from other indicators. In the objective function presented in Figure 4 (i.e., Equation 9), the matrix must be invertible to correctly compute the loss, which naturally leads to an analysis of RR.
>
> Neural networks such as LSTM, GRU, and Attention are widely used as base modules in many complex network structures to extract abstract features, owing to their outstanding performance in end-to-end tasks under MSE or entropy indicators. Even though these neural networks perform exceptionally well in end-to-end tasks, **it is challenging to quantify and analyze whether they are qualified as feature extractors**, and whether the features they extract are conducive to and consistent with the overall model. This issue, where the optimization objectives of sub-modules are inconsistent with that of the overall model, is referred to as mesa-optimization. A classic example is in CNNs, where the overall model's optimization goal is classification, while the shallow kernels aim to capture local textures and directional vectors. Although we cannot quantify and analyze the texture capture of CNNs in the field of computer vision, we can quantify and analyze the impact of COPU on RR in the time-series domain.
>
> As RR approaches one, the precision loss in Equation 9 decreases, and the gradient computation of COPU becomes more accurate. When RR equals one, the network's output is designed as the minimum variance estimator, and the gradients of the trackable parameters are conditional natural gradients, which incorporate manifold information and ensure that the network converges faster and more stably. These characteristics are not possessed by other common indicators, such as MSE, T$^2$, MAE, and so forth. Therefore, COPU places emphasis on analyzing RR.
>
> **Response to Weaknesses 2**
>
> Figure 4 illustrates the schematic relationship between CEP and COPU. Section 3, titled COPU: Cross-Correlation Enhanced Approximated Orthogonal Projection Unit on line 237, describes how to achieve COPU from a textual perspective. Specifically, CEP is introduced as the augmentation block in Equation 8. Subsequently, through computations in the dual space and the objective function, we obtain the loss, update the parameters via truncated gradient propagation as shown in Figure 4, and obtain the predicted output through the trackable parameters. The construction methods of the dual parameters and the objective function are shown in Equations 7 and 9, respectively.
>
> This structured design has a solid mathematical foundation and theoretical assurance, ensuring that when RR equals one, the network possesses the aforementioned excellent characteristics (natural gradients and minimum variance estimation). However, this design is highly sensitive to batch size; when RR is too small, the model may even fail to converge. For this reason, COPU is proposed to enhance model robustness and improve performance.
>
> **Response to Weaknesses 3**
>
> We greatly appreciate the reviewer’s valuable suggestions. Since the authors' team focuses on AI in industries, we habitually chose experiments from the field. The authors' team will endeavor to incorporate more datasets to provide a more comprehensive experimental analysis for COPU.
>
> **Response to Weaknesses 4**
>
> Figures 5 to 14 provide a detailed presentation of the static and dynamic changes in RR under different hyperparameter configurations in neural networks. We particularly emphasize the impact of different network depths on RR, as well as the influences of batch size and sequence length. These experiments offer valuable references for us to determine the hyperparameters of COPU. Except for AOPU, which follows the settings in its original paper, the hyperparameter settings of other comparative models are consistent with those of COPU, as indicated in Section 4.3, line 404.

---

> > ### Author Response · Authors · 2024-11-13
> > **Response #2 to Reviewer bobw**
> >
> > **Response to Questions 1**
> >
> > Due to the possibility of negative correlations between sequences, the value of $c$ may be negative.
> >
> > Ensuring that $c$ is positive through the sigmoid function has two advantages. First, it avoids numerical errors caused by $c$ being zero. Second, it assigns weights more accurately in cases of negative correlation.
> >
> > CEP enhances RR by suppressing homogeneous information and amplifying heterogeneous information. As shown in Equation 1, when performing the dictionary lookup, it is necessary to strengthen the weights of sequences that are dissimilar to the current sequence. The similarity between sequences is measured from two perspectives: the strength $c$ and the distance $ind$. The reason for considering the distance $ind$ is that, in industrial processes, many sensors measure the same data with different time delays, resulting in a large amount of redundant homogeneous information. Such copies of the same feature at different time delays should also be considered homogeneous information and thus be suppressed. Therefore, the expression $\frac{ind}{c}$ represents our requirement. When $ind$ is very small and $c$ is very large, it indicates that there is almost no time delay between the two sequences and high correlation; such homogeneous information needs to be suppressed. When $ind$ is large and $c$ is small, it indicates a significant time delay between the two sequences and low correlation, which can be considered independent information sources; such heterogeneous information needs to be extracted. Generally, $c$ is not negative because we calculate $c$ and $ind$ through argmax. Therefore, when $c$ is negative, it means a very rare situation has occurred where there are no positively correlated segments between the two sequences. This special case needs to be noted and given higher attention. Therefore, it is necessary to make $c$ positive while preserving the magnitude relationships; otherwise, softmax would assign extremely low weights. Hence, we choose to use the sigmoid function to impose this constraint.
> >
> > **Response to Questions 2**
> >
> >
> > While CEP can be embedded in other models as well, only COPU can explicitly leverage the various advantages brought by high RR. Figure 8 shows that the improvement of CEP itself over other base models isn't very outstanding. That is because CEP is proposed to address the issue of AOPU's robustness issue to batch size, rather than as an independent end-to-end solution. We believe that once the time-series deep learning community becomes aware of RR, much more outstanding results will emerge.
> >
> >
> > **Response to Questions 3**
> >
> > The code for CEP and COPU will be made publicly available after acceptance.
> >
> >
> > We sincerely believe that the research on COPU and RR is of significant value. We would greatly appreciate it if the reviewer could consider raising COPU's rating. If the reviewer has any further concerns, we are more than willing to address them.

---

> ### Author Response · Authors · 2024-11-21
> **Response #3 to Reviewer bobw**
>
> Dear Reviewer bobw,
>
> We are pleased to inform you that we have completed the revisions to our manuscript. We would like to extend our sincere gratitude for your patience and support. In the **global response to three important issues**, we have provided detailed explanations for the following three questions:
>
> 1. Why do we start focusing on studying RR instead of other metrics?
> 2. Why the loss function is formulated as shown in Equation 9, and the gradient is calculated as shown in Fig. 4?
> 3. What are the relationships among CEP, RR, and COPU?
>
> We hope that these explanations, along with the revisions to our manuscript and our prior discussions, will enhance your appreciation of COPU. If you have any additional questions, we would be happy to address them promptly.
>
> Sincerely,
>
> All authors

---

> ### Author Response · Authors · 2024-11-25
> **Response #4 to Reviewer bobw**
>
> Dear Reviewer bobw,
>
> As the rebuttal-discussion period is coming to a close in approximately one day, we would like to know whether you have any remaining concerns regarding COPU.
>
> We would also like to recommend, in order, several responses that we believe summarize the COPU. We hope these will enhance your appreciation of our work:
>
> 1. **Four reasons for raising the rating of COPU (Comment #2 on the second revision)**
> 2. **Response #1 to three important issues**
> 3. **Comment #1 on the second revision**
> 4. **Response #8 to Reviewer VW2o**
> 5. **Response #7 to Reviewer VW2o**
>
> Furthermore, we have expanded our experiments by introducing additional datasets and methods to ensure the effectiveness and superiority of COPU, and providing additional theoretical proof, as we have shown in **Comment (#1 & #2) on the (first & second) revision**.  In our **Comment (#1 & #2) on the first revision** we also listed some interesting analyses.
>
> We hope that this revision will merit an increase in your rating. If you have any additional concerns, we will be more than happy to address them promptly.

---

### Official Review · Reviewer_mbUm · 2024-11-04

**Soundness:** 3
**Presentation:** 3
**Contribution:** 3
**Rating:** 6
**Confidence:** 4

**Summary:**

This paper, titled COPU: Recognizing Time Series' Heterogeneity in Stacked Neural Network, addresses limitations in traditional neural network (NN) architectures when applied to time series data. The authors propose the COPU (Cross-correlation Enhanced Approximated Orthogonal Projection Unit), a new framework designed to enhance the NN's nonlinear modeling capabilities by emphasizing heterogeneous features. Traditional NN structures often focus on homogeneous feature extraction, which limits their performance on time series data where diverse feature extraction is essential. The COPU framework uses the Rank Ratio (RR) metric to measure the model's ability to capture unique information, enhancing heterogeneous components while compressing homogeneous ones. Experimental results demonstrate that COPU significantly outperforms existing NN structures in capturing nonlinear patterns, especially in time series datasets like SRU.

**Strengths:**

Timely and Novel Research Direction: In time series forecasting research, most studies have traditionally focused on increasing model capacity. This paper takes a novel approach by addressing time series analysis from a fresh perspective that emphasizes data-specific structural improvements. This unique angle offers high novelty by shifting from merely increasing the capacity of NNs to a more data-driven design that aligns closely with the characteristics of time series data.

Innovative Methodology: COPU stands out for emphasizing heterogeneous over homogeneous elements in the data, which is well-suited to the nature of time series data. Unlike previous studies that focus on extracting homogeneous features, this paper proposes the Cross-correlation Enhanced Perceptron (CEP) to align features based on their correlations, suppressing redundant homogeneous information while amplifying unique features. This approach enhances the NN's ability to capture nonlinear aspects effectively, making it highly suitable for time series analysis.

Performance and Stability: COPU demonstrates superior performance compared to traditional NN structures, especially in terms of capturing diverse features without overfitting. This model excels in real-world scenarios by effectively learning complex and varied representations in time series data, showcasing both stability and robustness.

**Weaknesses:**

Limited Applicability: COPU is specifically designed for time series data, which could restrict its application across different data types, such as image or text data. Although the focus on time series data is intentional, this may limit its adaptability to more general purposes.

Model Complexity: The architecture of COPU, particularly with CEP integration, is relatively complex, potentially requiring more computational resources than conventional NN models. This complexity could present challenges for implementation, especially for large-scale datasets or real-time applications.

**Questions:**

Generalizability of the RR Metric: The paper shows that RR is effective for assessing nonlinear modeling capabilities in time series data, but it is unclear if RR can be reliably applied to other data types (e.g., continuous vs. discrete time series, multimodal data).

Balancing Homogeneous and Heterogeneous Elements: While the approach of suppressing homogeneous features and amplifying heterogeneous ones is innovative, the paper does not detail how the balance between these elements impacts performance. More insight into how this balance can be adjusted and its effect on model performance would be valuable.

Effect of Stack Depth on COPU’s Performance: Although increasing the depth of COPU stacks improves performance, the model exhibits diminishing returns at higher depths. It would be helpful to have guidelines on optimizing stack depth for this model.

---

> ### Author Response · Authors · 2024-11-13
> **Response #1 to Reviewer mbUm**
>
> **Response to Reviewer mbUm**
>
> We are deeply grateful for the time and effort the reviewer has invested in reviewing our manuscript. Your recognition and support are crucial to us. We hold each of your concerns in the highest regard and will address them thoughtfully in the following.
>
> **Response to Strengths**
>
> Thank you for your gracious recognition and commendation. Through our practical deployment of deep learning methodologies, we have found that complex models do not invariably outperform simpler ones, particularly within industrial environments. This phenomenon starkly contrasts observations in other domains such as computer vision. We introduce the research of COPU to unravel the underlying reasons for this phenomenon. We believe that COPU will illuminate new directions for future endeavors in time series analysis within the deep learning community and make good contributions to the field.
>
> **Response to Weaknesses**
>
> The reviewer's observation is accurate. COPU focuses on analyzing how deep learning can properly adapt to time-series modal data through the examination of RR, making it challenging to generalize to other data types. Due to the involvement of time-frequency conversions in CEP, COPU incurs additional computational time overhead compared to traditional network architectures.
>
> Current research on neural networks in the time-series domain remains heuristic, relying on empirical conjecture and lacking in-depth domain knowledge. COPU proposes a universal monitoring metric in time-series domain that significantly enhances the interpretability of NNs, providing guidance for the design of deep network structures, which is of great significance. Therefore, we believe that some extra computational overhead is acceptable.
>
> **Response to Questions**
> 1.   Essentially, RR is a metric that evaluates the degree to which a model approximates the minimum variance estimator. The minimum variance estimator is the optimal unbiased estimator and is considered the ceiling for regression tasks. Therefore, the analysis of RR does not involve considerations of discrete or continuous properties. For other data modalities, such as computer vision, the minimum variance estimator may not be their task of interest; hence, we assert that COPU is not suitable for other modalities.
>
> 2.   As RR approaches one, the gradient backpropagation of the objective function in Figure 4 becomes more precise, and the neural network acquires more favorable properties. When RR equals one, the gradient of the trackable parameters is equivalent to the conditional natural gradient, which incorporates manifold information and ensures that the network converges faster and more stable. Simultaneously, the network's output becomes equivalent to the minimum variance estimator, ensuring that the neural network can reach the performance ceiling. Conversely, as RR approaches zero, the error in gradient backpropagation increases (due to the matrix inversion in Equation 9), leading to less stable model performance.
> Therefore, COPU's emphasis on suppressing homogeneous features and amplifying heterogeneous ones aims to increase RR. The series of experiments from Figures 5 to 8 were conducted to investigate how these elements impact RR, while Tables I and II illustrate how the RR influences performance. We hope this explanation resolves your confusion.
>
> 3.   We also find this topic exceptionally interesting and engaging. Through Figures 5 to 14, we meticulously present the inherent RR of the data under different batch sizes and sequence length settings, the RR of different layers during initialization, the dynamic changes of RR in various layers within networks of differing depths during training, and the performance monitoring results of networks with different depths throughout training. This approach enables us to deeply understand the internal changes occurring during the network training process and to determine how to choose the stack depth. We have discovered that depths capable of significantly enhancing RR during training tend to exhibit the best performance, rather than those depths and structures where RR remains constant over time. Our experimental results partially corroborate our findings: when COPU is deeper, RR is higher, and the model becomes more robust; when RR increases substantially, the model's performance also improves.
>
>
>
> We believe that COPU's thorough and meticulous analysis of the neural network training process highlights a highly valuable research direction in the time-series domain. We would be grateful if the reviewer could consider raising COPU's rating. If the reviewer has any further concerns, we are more than happy to help address them.

---

> ### Author Response · Authors · 2024-11-21
> **Response #2 to Reviewer mbUm**
>
> Dear Reviewer mbUm,
>
> We are pleased to inform you that we have completed the revisions to our manuscript. We would like to extend our sincere gratitude for your recognition. In the **global response to three important issues**, we have provided detailed explanations for the following three issues:
>
> 1. Why do we start focusing on studying RR instead of other metrics?
> 2. Why the loss function is formulated as shown in Equation 9, and gradient is calculated as shown in Fig. 4?
> 3. What are the relationships among CEP, RR, and COPU?
>
> We hope that these explanations, along with the revisions to our manuscript and our prior discussions, will enhance your appreciation of COPU. If you have any additional questions, we would be happy to address them promptly.
>
> Sincerely,
>
> All authors

---

> ### Author Response · Authors · 2024-11-25
> **Response #3 to Reviewer mbUm**
>
> Dear Reviewer mbUm,
>
> As the rebuttal-discussion period is coming to a close in approximately one day, we would like to know whether you have any remaining concerns regarding COPU.
>
> We would also like to recommend, in order, several responses that we believe summarize the COPU. We hope these will enhance your appreciation of our work:
>
> 1. **Four reasons for raising the rating of COPU (Comment #2 on the second revision)**
> 2. **Response #1 to three important issues**
> 3. **Comment #1 on the second revision**
> 4. **Response #8 to Reviewer VW2o**
> 5. **Response #7 to Reviewer VW2o**
>
> Furthermore, we have expanded our experiments by introducing additional datasets and methods to ensure the effectiveness and superiority of COPU, and providing additional theoretical proof, as we have shown in **Comment (#1 & #2) on the (first & second) revision**.  In our **Comment (#1 & #2) on the first revision** we also listed some interesting analyses.
>
> We hope that this revision will merit an increase in your rating. If you have any additional concerns, we will be more than happy to address them promptly.

---

> > ### Comment · Reviewer_mbUm · 2024-11-26
> >
> > Thank you, after considering all of your responses I will maintain my score.

---

> > > ### Author Response · Authors · 2024-11-26
> > > **Response #4 to Reviewer mbUm**
> > >
> > > Dear Reviewer mbUm,
> > >
> > > We would like to extend our sincere gratitude for your support of all time. It is your recognition that encourages us throughout the rebuttal-discussion period. We hope this review process has been a delightful journey for you.
> > >
> > > Sincerely,
> > > All authors

---

### Author Response · Authors · 2024-11-20
**Comment #1 on the first revision**

Dear all Reviewers,

We are delighted to inform you that we have completed the revision of our manuscript. We would also like to thank all reviewers for the careful reading, helpful comments, and constructive suggestions. All these efforts have significantly improved the presentation of the manuscript. We paid specific attention to the following aspects:

- Experiment: Conducted experiments on additional datasets to strengthen the general applicability of COPU and RR.
- Presentation: Emphasized the research focus and contributions, refined certain expressions, and corrected some typos.

In the revised manuscript, **the content highlighted in red denotes additional statements, experimental results, or analytical conclusions, while the content highlighted in orange represents refinements to the original material.** The specific changes are as follows:

1. Added a new section in Appendix D to analyze additional experiments.
2. Adjusted the presentation order of datasets in Tables I and II to emphasize COPU's leading advantages on the more challenging dataset (Debutanizer).
3. Emphasized the differences between the COPU study and other research on line 88.
4. Provided additional explanation of the sigmoid function on line 204.
5. Adjusted the order of equations on line 253 and corrected typos.
6. Corrected two reference type errors.
7. Added more explanations to the manuscript.
8. Deleted some sentences and shrunk the figure size to make room for the adjustments.

For the reviewers’ convenience, we have listed the important content added in the revision below.



>**(Line 88)** The mainstream of conventional stacked network research involves empirically explore the feasibility of extracting features using base modules that exhibit outstanding performance under mean squared error or entropy loss in end-to-end tasks. However, it remains challenging to quantify and analyze whether these modules are qualified as feature extractors, and whether the features they extract are conducive to and consistent with the overall model. Furthermore, they struggle to monitor whether such extracted features are beneficial in optimization, much less guide directions for model improvement.

---

> ### Author Response · Authors · 2024-11-20
> **Comment #2 on the first revision**
>
> > **(Line 1150)** The issue of RR highlights the significant difference between time-series data and CV/NLP modalities, that the time-series data are less informative. The pronounced decrease in RR as batch size increases indicates that the unique information contained within samples diminishes, making them more easily represented linearly by other samples. **This phenomenon has actually been implicitly recognized in the field of industrial system identification.** For example, when applying Ordinary Least Squares (OLS), there is a stringent requirement for the number of samples to vastly exceed the dimensionality of features, precisely because it is challenging to ensure the invertibility of $xx^T$. Similarly, Partial Least Squares (PLS) is proposed as an identification tool to refine heterogeneous information within samples while suppressing homogeneous ones.
> >
> >$\quad$
> >
> > It is this characteristic of time-series data that renders the trivial application of models designed for CV/NLP modalities inappropriate. Modules such as Attention aims to focus on features of interest from vast and diverse information. However, in the context of time series, they might extract multiple copies of the same pattern from large amounts of homogeneous information. Such modules lack the ability to regularize the diversity of the extracted features. Figures 26 and 27 and Tables 1 to 4 illustrate this point both qualitatively and quantitatively. We believe that a deeper understanding of RR can help transcend the limitations of existing models and ultimately lead to the design of network architectures specifically tailored for time-series analysis.
> >
> >$\quad$
> >
> >${Remark}$: The issue of RR further reveals a potential question.	In the domains of CV/NLP, increasing the batch size can reduce gradient variance, thereby aiding the model in optimizing toward a global optimum. However, in the field of time series, expanding the batch size may not change the number of effective samples, which may provoke a series of questions and considerations.
> >
> >$\quad$
> >
> >The impact of label consistency between redundant and effective samples presents an intriguing question that remains unclear and merits further investigation.
> Redundant samples refer to the samples constituting the difference between the batch size and the number of effective samples. **Essentially, these are linear combinations of effective samples, prompting us to ponder whether their labels satisfy such linear relationships.**
> If their labels do satisfy these linear relationships, which essentially implies embodying a linear model prior, will this cause networks to tend toward degenerating into linear models?
> Conversely, if their labels do not satisfy such linear relationships, will this encourage networks to learn the nonlinear dynamic characteristics inherent in the data, or will it lead to conflicts resulting in unstable updates?
> We believe that these reflections based on the RR issue can deepen our understanding of applying NNs in the realm of time-series analysis.
>
>
>
> We feel obliged to explain to all reviewers why we do not present experimental results on ETTh1 and ETTm1. The performance of all regression models on these two datasets is exceedingly poor, to the extent that the results compare not who excels but who is less inadequate. According to the $R^2$ metric, all methods exhibit significantly negative values on these datasets, indicating models' outputs offer no positive contribution. From an industrial and practical perspective, such comparisons hold little value. The two datasets might not be suitable for the regression task. COPU does not demonstrate outstanding performance on ETTh1 and ETTm1. Conversely, on ETTm2 and ETTh2, the $R^2$ results of all methods fluctuate between 0.1 and 0.3, rendering the comparisons meaningful. On ETTh2 and ETTm2, COPU continues to maintain its leading advantage.
>
> We are honest and sincere about our work. We have been meticulous in our choice of wording throughout the manuscript and are highly confident in the innovation and contributions of COPU. We believe that, with our clarifications and additional experiments, the focus and impact of COPU are now clear. We would greatly appreciate it if reviewers could consider raising COPU's rating. If reviewers have any further concerns, we are more than happy to help address them.

---

### Author Response · Authors · 2024-11-21
**Response #1 to three important issues**

In this global response, we aim to provide explanations for the following three issues to offer a clear interpretation of our manuscript:

We sincerely hope that these explanations can assist the reviewers in better understanding and appreciating our manuscript.


- **Why did we choose to focus on studying RR instead of other metrics?**

- Investigating the RR is an unavoidable aspect of studying COPU. The loss function of COPU (Equation 9) is not always well-defined. When the matrix $\tilde{x}$ is not of full column rank, the matrix $\tilde{x}^T\tilde{x}$ becomes non-invertible. To compute Equation 9, numerical approximation is necessary; we utilize an inversion computation based on SVD. In this context, RR represents the proportion of non-zero singular values, which reflects the precision of the inversion. A higher RR leads to a more accurate computation of COPU’s loss, enabling a better restoration of the MSE. Conversely, a lower RR results in greater errors in COPU’s loss, tending towards a quadratic form. **This is the initial reason for studying RR; therefore, RR is not an arbitrarily chosen metric.** During our subsequent research on RR, we observe many thought-provoking phenomena, which motivates the study presented in this paper.

$\quad$

- **Why is the loss function formulated as shown in Equation 9, and why is the gradient calculated as depicted in Figure 4?**

- The objective of Equation 9 is to restore the MSE. Then, why do we not compute the MSE directly? Because COPU aims to restrict the backpropagation path, ensuring it passes through the dual parameter $m$. Why does COPU require that the gradient is calculated through the dual parameter $m$ as illustrated in Figure 4? Because the truncated gradient at the dual parameter is approximately the natural gradient of the trackable parameter. The trackable parameter is crucial since it is used in the forward computation. The degree of approximation to the natural gradient is also measured by RR. **Overall, these considerations are intended to enable the use of the natural gradient to update the trackable parameter.** The advantage of the natural gradient lies in its ability to recognize manifold structures; it is a gradient that considers the second-order derivatives of the parameter space, achieving more stable optimization and better performance compared to first-order gradient optimizers such as Adam **(see [1](https://openreview.net/forum?id=xqrlhsbcwN&referrer=%5Bthe%20profile%20of%20Shaoqi%20Wang%5D(%2Fprofile%3Fid%3D~Shaoqi_Wang3)),[2](https://openreview.net/forum?id=vuMD71R20q&referrer=%5Bthe%20profile%20of%20Wu%20Lin%5D(%2Fprofile%3Fid%3D~Wu_Lin2)),[3](https://openreview.net/forum?id=Y2wRKE0Qor&referrer=%5Bthe%20profile%20of%20Wu%20Lin%5D(%2Fprofile%3Fid%3D~Wu_Lin2)))**. It is important to note that COPU essentially employs a conditional natural gradient, as CEP is updated through standard optimizers.

$\quad$

- **What are the relationships among RR, CEP, and COPU?**

- RR is a general metric that can be applied to any model and modality. The variation of RR with batch size and sequence length reflects the intrinsic information density of the dataset (modality). The change of RR with the number of stacked layers reflects the network's nonlinear modeling capability. Specifically, in COPU, RR additionally measures the computational precision of the objective function, the degree to which the truncated gradient approximates the conditional natural gradient, and the extent to which the network output approximates the minimum variance estimation. CEP is an important stackable module within COPU, designed to address the issue that RR tends to be too small in time series data. A too-small RR has a devastating impact on COPU, as it causes the chaotic error information in the loss to overwhelm the effective information. CEP conducts an in-depth analysis of the cross-correlation characteristics of time series data, suppressing homogeneous information while accessing heterogeneous information to enhance RR. COPU is an integration of CEP, RR, dual parameters, and trackable parameters; it is a comprehensive time series regression model whose structure is illustrated in Figure 4.

---

### Author Response · Authors · 2024-11-22
**Comment #1 on the second revision**

Dear Reviewers,

We would like to report our latest revisions. In response to reviewer VW2o's request, we have added Section E in the appendix, where we provide a detailed analysis of the rationality of the COPU loss function (Equation 9). We have demonstrated that this loss function employs the orthogonal projection from $\tilde{x}$ to $\tilde{x}y$, which effectively restores the MSE, with computational accuracy measured by the RR metric. We also plan to include additional experimental results for SCINet, Pyraformer, TimesNet, and TimeMixer on the SRU, Debutanizer, ETTh2, and ETTm2 datasets. We will make every effort to complete these experiments within the next three days.

For your convenience, we have excerpted the key portions of Section E below. Reviewers may refer to these in conjunction with our **"Response #1 to Three Important Issues”**. We hope that these explanations can assist the reviewers in better understanding and appreciating our manuscript.

> In our previous discussion, we introduce the properties of the minimum variance estimator. Next, we will demonstrate that COPU is the augmented minimum variance estimator from $\tilde{x}$ to $\tilde{x}y$. Note in the derivation of COPU, the concepts of samples and features are interchanged.
AOPU has already provided a detailed proof that the linear minimum variance estimator from $x$ to $y$ has the form $\text{E}[y]$+$\text{R}\_{yx}\text{R}\_{xx}^{-1}(x-\text{E}[x])$.
Since the data are normalized, we assume for convenience that all variables have a mean of zero, where $\text{R}\_{yx}=\text{E}[y^Tx]$. Therefore, $\text{R}\_{yx}\text{R}\_{xx}^{-1}x$ simplifies to the linear minimum variance estimator from $x$ to $y$, which is also known the orthogonal projection.
>
> $\quad$
>
> Consequently, the minimum variance estimator from $\tilde{x}$ to $\tilde{x}y$ is $\tilde{x}^T(\tilde{x}^T\tilde{x})^{-1}\tilde{x}^T\tilde{x}{y\_{copu}}$, noting that we have performed a transposition here. In this context, $R_{\tilde{x}\tilde{x}}^{-1}$ corresponds to $(\tilde{x}^T \tilde{x})^{-1}$, and $R\_{\tilde{x}y}$ corresponds to $\tilde{x}^T \tilde{x} y\_{\text{copu}}$. Here, $y\_{\text{copu}}$ represents the output of the forward process of COPU, i.e., $\tilde{x}^T \theta\_r$. Since $\tilde{x} R\_{\tilde{x}\tilde{x}} R_{\tilde{x}y}$ represents the minimum variance estimator from $\tilde{x}$ to $\tilde{x}y$, COPU uses $R\_{\tilde{x}\tilde{x}} R_{\tilde{x}y}$, i.e., $(\tilde{x}^T \tilde{x})^{-1} \tilde{x}^T m$, as the minimum variance estimator of $y$ to compute the loss in Equation 9. For this method to be effective, we must be able to restore $y$ from $\tilde{x}y$ given $\tilde{x}$; this requires that $\tilde{x}$ must be of full column rank, i.e., a larger RR.

---

> ### Author Response · Authors · 2024-11-25
> **Four reasons for raising the rating of COPU (Comment #2 on the second revision)**
>
> Dear reviewers,
>
> With less than two days remaining in the discussion period, we would like to present four reasons for raising the rating of COPU. We sincerely invite the reviewers to share any concerns they may have, and we will make every effort to address them promptly.
>
> 1. **Novelty**: Unlike mainstream model-wise studies, COPU distinguishes itself by pioneering an investigation into the characteristics of the time-series modality and proposing the RR metric. This work explains why complex models perform poorly in the time-series domain, specifically, because time-series data are often less informative. We introduce the CEP module to address this issue, which enhances heterogeneous information while suppressing homogeneous information. CEP helps us focus on diverse data patterns. By truncating the gradient backpropagation path, COPU achieves better optimization, explicitly leveraging the advantages of high RR, and resulting in significant improvements in both performance and robustness.
>
> 2. **Contribution**: The greatest contribution of COPU lies in the universality of its findings. Unlike mainstream model-wise studies, we did more than merely propose a model. We revealed a phenomenon in the time-series, namely, that time-series data are often less informative. The observation that RR is significantly less than 1 indicates an insufficient number of effective samples within a mini-batch, which may provoke a series of questions and considerations ***(see our Comment #2 on the first revision)***. Conventional studies have overlooked this issue, which explains why SOTA models perform poorly on other tasks. COPU's findings can be referenced by all researchers in time-series analysis, advancing progress in the field.
>
> 3. **Theoretical Foundation**: COPU is an extension of work based on AOPU, and its structure has solid theoretical support. AOPU explains the truncated gradient, demonstrating that RR is a metric for the degree of approximation of the truncated gradient to the natural gradient (a superior second-order, manifold-aware gradient). COPU explains the loss, proving that RR is a metric for the degree of approximation of the objective function to the minimum variance estimation (the ceiling performance of an unbiased regression model).
>
> 4. **Experimental Results**: The experiments conducted for COPU are comprehensive, and the results are significant. We tested three of the most commonly used NN-base modules, selecting two representative methods for each. The results in Table 1 demonstrate that COPU significantly outperforms all comparison methods. To verify the superiority of CEP, we tested COPU's performance using different base modules as the augmentation block. Table 2 shows that CEP is the only module capable of maintaining robust performance. At the request of reviewer VW2o, we have additionally included experiments on the ETTh2 and ETTm2 datasets, as well as experimental results of multiple SOTA time-series prediction methods in Tables 4,5, and 6. These results further substantiate the advantages of COPU and CEP.
>
>
> We are also pleased to announce that, per reviewer VW2o's request, we have completed the second revision of the manuscript. Reviewers can refer to Section D, Table 5 and 6 in the appendix to see the newly added experimental results of SOTA methods on the regression task.
>
> *Note: We inadvertently calculated the MSE, MAE, and MAPE of the outputs of Debutanizer and SRU on a normalized scale (which should have been calculated on their original scale), causing the results in Table 5 to be not directly comparable with those in Table 1. We are working diligently to correct this issue. We have highlighted it in red to draw your attention. We are presenting the results as they are for reviewers to examine and discuss.*
>
> We hope that our efforts will enhance your appreciation of our manuscript. We would be grateful if the reviewers would consider raising COPU's rating.

---

> > ### Author Response · Authors · 2024-11-26
> > **Comment #3 on the second revision**
> >
> > Dear reviewers,
> >
> > We are also pleased to announce that we have fixed the scale problems mentioned above. Table 1 and Table 5 are now comparable. We found that COPU outperforms these SOTAs significantly.
> >
> > **I will not tediously repeat COPU's contribution. But we should wonder why are these SOTAs not sound, and can't be generalized to other tasks (e.g., regression) like SOTAs in CV/NLP did. We believe COPU offers great insight into this.**
> >
> > If not required by the reviewers, we shall not update the manuscript anymore.

---

### Author Response · Authors · 2024-12-04
**Summary**

Dear PCs, ACs, and all reviewers,

As the rebuttal-discussion period draws to a close, We would like to provide a summary of our work and the rebuttal-discussion process to offer clarity.

We proposed COPU and RR to advance our understanding of NNs in time series analysis.
We proved that RR is a unique and indispensable core metric.
An RR significantly less than 1 implies a high degree of redundant information in the time series field, suggesting that conventional feature extraction modules may be unsuitable.
We introduced CEP to enhance RR.
CEP increases RR by amplifying heterogeneous information while compressing homogeneous ones.
We proposed COPU as a regression network model.
We conducted extensive and detailed experiments to demonstrate COPU's advancements, specifically the improvements over AOPU in robustness and over SOTAs in performance.



During the rebuttal, we primarily:
1. Clarified certain misunderstandings.
2. Corrected typos, incorrect reference types, and errors in paragraph sequencing.
3. Added experiments on additional datasets and with additional SOTAs.
4. Provided theoretical proofs affirming the unique and indispensable nature of RR.

After the rebuttal, reviewers mbUm and bobw maintained their ratings (6$\rightarrow$6,6$\rightarrow$6). Reviewer 1q18 increased the rating (3$\rightarrow$5$\rightarrow$6). Reviewer VW2o decided to reject our work (5$\rightarrow$3).

We are not convinced by reviewer VW2o's decision.
Reviewer VW2o did not provide sufficient reasoning and did not engage in effective discussion with us.
Our work introduces unique innovations, uncovers significant issues within the domain, and offers valuable insights for all researchers in this field.
We also met reviewer VW2o's requirements on additional experiments and proof.
However, reviewer VW2o did not mention these critical aspects at all in the rationale for rejection.
PCs, ACs, and other reviewers can check our discussions with reviewer VW2o along with our global responses if interested.


We hope this summary will assist PCs, ACs, and all reviewers in making the final decision.

Sincerely,

All authors

---

### Meta-Review · Area_Chair_4vAS · 2024-12-20

**Metareview:**

The paper presents a data-specific structural improvements that is an uniquely novel approach. Authors propose a Cross-correlation Enhanced Perceptron (CEP) to achieve deep nonlinear modeling for time series data Furthermore, the random gaussian matrix with CEP
is introduced as the augmentation interface to develop a COPU, ensuring the accuracy of natural gradient calculations. The CEP, smart utilisation of RR and COPU are solid contributions.
Strengths:
1.Timely and Novel Research Direction, innovative, superior and stable performance .[Reviewer mBUm, bobw, VW2o]
2. Good motivation, good metric and good choice of dataset demonstration [Reviewer 1q18]
3. Novel introduction of utilizing RR as a metric to quantify the proportion of linear dependencies transformed into independence by the network, thereby reflecting the nonlinear capacity

Weaknesses:
1. While we agree with the authors claim that " When the number of layers equals 2, CEP’s performance is on par with other comparative methods; when the number of layers reaches 7, CEP surpasses all other NNs, both in stability and convergence accuracy", From Table 1, the performance of COPU seems quite similar to the baselines across the two data sets and various stacks - esp the improvement in performance in comparison to MLP is not very substantial

2. Evaluations done only on two data sets, and the recent foundation models for time-series forecasting has been ignored - although these models perform forecasting, they can also be used for regression based on their representations [PatchTST etc: Would like to bring this paper to the author's attention: https://arxiv.org/abs/2405.02358]

3. Authors make this claim: We wish to emphasize that CEP makes COPU much more robust to the change of RR while other NN structures do not. This can be attributed to CEP’s ability to maintain high RR during training

While the method is interesting, the definition of metrics are valuable contributions, the paper could benefit from more evaluations on more time-series data sets and more baselines. Furthermore, the impact of the method is also not very substantial (Although there are performance gains in regression problems)

**Additional Comments On Reviewer Discussion:**

Authors insights during rebuttal:
1.  Therefore, COPU's emphasis on suppressing homogeneous features and amplifying heterogeneous ones aims to increase RR.
2. RR is most suitable for time-series and does not serve image data sets

The authors have tried to address comments by the reviewers, and have largely helped to address.

---

### Decision · Program_Chairs · 2025-01-22

Reject